# A real-world multi-center RNA-seq benchmarking study using the Quartet and MAQC reference materials

Duo Wang[1,2,3,7], Yaqing Liu [4,7], Yuanfeng Zhang [1,2,3], Qingwang Chen[4], Yanxi Han[1,2,3], Wanwan Hou[4], Cong Liu[1,2,3], Ying Yu [4], Ziyang Li[5], Ziqiang Li[1,2,3], Jiaxin Zhao[1,2,3], Leming Shi [4,6] ✉, Yuanting Zheng [4,6] ✉, Jinming Li [1,2,3] ✉ & Rui Zhang [1,2,3] ✉

Translating RNA-seq into clinical diagnostics requires ensuring the reliability and cross-laboratory consistency of detecting clinically relevant subtle differential expressions, such as those between different disease subtypes or stages. As part of the Quartet project, we present an RNA-seq benchmarking study across 45 laboratories using the Quartet and MAQC reference samples spiked with ERCC controls. Based on multiple types of 'ground truth', we systematically assess the real-world RNA-seq performance and investigate the influencing factors involved in 26 experimental processes and 140 bioinformatics pipelines. Here we show greater inter-laboratory variations in detecting subtle differential expressions among the Quartet samples. Experimental factors including mRNA enrichment and strandedness, and each bioinformatics step, emerge as primary sources of variations in gene expression. We underscore the profound influence of experimental execution, and provide best practice recommendations for experimental designs, strategies for filtering low-expression genes, and the optimal gene annotation and analysis pipelines. In summary, this study lays the foundation for developing and quality control of RNA-seq for clinical diagnostic purposes.

Transcriptome sequencing (RNA-seq) has expanded new avenues for exploring global expression patterns as well as identifying alternative splicing events[1]. Differential expression analysis of transcriptomic data enables genome-wide identification of gene or isoform expression changes associated with biological conditions of interest. This contributes significantly to the discovery of biomarkers for disease diagnosis[2], prognosis[3], and therapeutic selection[4]. These evidences facilitate the application of RNA-seq in clinical routine. Noticeably,

clinically relevant biological differences among study groups are often small, especially between certain diseases and normal tissues[5,6], or between different disease subtypes or stages[7–11], implying minor changes in gene expression profiles between sample types. We refer to such minor expression differences between sample groups with similar transcriptome profiles, as subtle differential expression, which typically manifests in the detection of fewer differentially expressed genes (DEGs). Subtle differential expression is typically challenging to

[1]National Center for Clinical Laboratories, Institute of Geriatric Medicine, Chinese Academy of Medical Sciences, Beijing Hospital/National Center of Gerontology, Beijing, PR China. [2]National Center for Clinical Laboratories, Chinese Academy of Medical Sciences & Peking Union Medical College, Beijing, PR China. [3]Beijing Engineering Research Center of Laboratory Medicine, Beijing Hospital, Beijing, PR China. [4]State Key Laboratory of Genetic Engineering, School of Life Sciences, Human Phenome Institute, and Shanghai Cancer Center, Fudan University, Shanghai, China. [5]Department of Laboratory Medicine, The Second Xiangya Hospital, Central South University, Changsha, Hunan, PR China. [6]International Human Phenome Institutes, Shanghai, China. [7]These authors contributed equally: Duo Wang, Yaqing Liu. ✉e-mail: lemingshi@fudan.edu.cn; zhengyuanting@fudan.edu.cn; jmli@nccl.org.cn; ruizhang@nccl.org.cn

distinguish from the technical noises of RNA-seq. Therefore, translating RNA-seq into clinical diagnostics poses requirements for more sensitive differential expression analysis, emphasizing the necessity for quality assessment at subtle differential expression levels.

However, over the past decade, quality assessment of RNA-seq in the community has predominantly relied on the milestone MAQC reference materials, characterized by significantly large biological differences between samples, which were developed by the Micro-Array/Sequencing Quality Control (SEQC/MAQC) Consortium from ten cancer cell lines (MAQC A) and brain tissues of 23 donors (MAQC B)[12]. The MAQC Consortium utilized these samples with spike-ins of 92 synthetic RNA from the External RNA Control Consortium (ERCC) to assess RNA-seq performance and demonstrated a high accuracy and reproducibility of relative expression measurements across different sites and platforms under appropriate data processing and analysis conditions[13,14]. More large-scale studies have also employed these two RNA reference materials to compare different library preparation protocols and sequencing platforms[15–17], and have utilized the MAQC datasets for benchmarking bioinformatics pipelines[18–21]. Moreover, the Genetic European Variation in Disease, a European Medical Sequencing (GEUVADIS) Consortium sequenced RNA samples from lymphoblastoid cell lines of 465 individuals across seven sites to assess reproducibility across laboratories and examined the sources of inter-laboratory variation under an identical experimental and bioinformatics process[22].

Noticeably, quality control based on the MAQC reference materials may not fully ensure the accurate identification of clinically relevant subtle differential expression[23]. Moreover, in contrast to the rigorously controlled RNA-seq workflows of previous study designs, the real-world scenarios present significant differences in sample processing, experimental protocols, sequencing platforms, and analysis pipelines across laboratories, where confounding factors may compromise the accuracy and reproducibility of RNA-seq[14,15,22]. In the context of such diverse experimental and bioinformatics processes, understanding of the sources of inter-laboratory variation remains limited. Therefore, a detailed quality assessment of the overall performance of real-world RNA-seq in detecting subtle differential expression for clinical diagnostic purposes and of the technical factors affecting diagnostic performance is necessary.

Recently, the Quartet project for quality control and data integration of multi-omics profiling introduced multi-omics reference materials derived from immortalized B-lymphoblastoid cell lines from a Chinese quartet family of parents and monozygotic twin daughters[24]. These well-characterized, homogenous, and stable Quartet RNA reference materials have small inter-sample biological differences, exhibiting a comparable number of DEGs to clinically relevant sample groups and significantly fewer DEGs than the MAQC samples[23]. Furthermore, the Quartet RNA reference materials also provided large-scale ratio-based reference datasets[23]. Thus, the Quartet samples could reflect subtle differential expression, providing a unique opportunity for the assessment and benchmarking of transcriptome profiling at subtle differential expression levels in a reference-based manner.

Within the scope of the Quartet project, this study utilized the Quartet RNA samples with spike-ins of ERCC controls, and MAQC RNA samples to generate RNA-seq data across 45 independent laboratories, each using its own in-house experimental protocol and analysis pipeline. Overall, approximately 120 billion reads of RNA-seq data from 1080 libraries were generated and analyzed, representing the most extensive effort to conduct an in-depth exploration of transcriptome data to date. Through the quality assessment based on the Quartet and MAQC samples in parallel, this study thoroughly elucidated the performance of real-world RNA-seq, particularly when detecting subtle differential expression. Subsequently, we leveraged gene expression data from 26 different experimental processes and 140 differential analysis pipelines to investigate sources of variation in the

experimental and bioinformatic aspects, respectively. This study provides best practice recommendations for the experimental and bioinformatics designs of the RNA-seq toward the scientific question addressed, and underscores the necessity of quality controls at subtle differential expression levels through the comparisons of the Quartet and MAQC reference materials.

## Results
### Study design
Our multi-center study involved four well-characterized Quartet RNA samples (M8, F7, D5, and D6) with ERCC RNA controls spiked into M8 and D6 samples, T1 and T2 samples constructed by mixing M8 and D6 at the defined ratios of 3:1 and 1:3, respectively, and MAQC RNA samples A and B (Fig. 1a). The sample panel design introduces various types of ground truth, encompassing three reference datasets: the Quartet reference datasets and the TaqMan datasets for Quartet and MAQC samples, and 'built-in truth' involving ERCC spike-in ratios and known mixing ratios for the T1 and T2 samples (Methods). Each sample was provided with three technical replicates, resulting in a total of 24 RNA samples, which were sequenced and analyzed by 45 independent laboratories. Each laboratory employed distinct RNA-seq workflows, involving different RNA processing methods, library preparation protocols, sequencing platforms, and bioinformatics pipelines (Supplementary Data 1). Overall, RNA-seq data from 45 different laboratories reflected the inter-laboratory variations. Sixteen laboratories assigned all libraries to different flowcells or lanes for sequencing, introducing batch effects in RNA-seq data, while other laboratories sequenced them within the same lane, thereby without batch effects. This approach accurately mirrored the actual research practices in real-world scenarios.

In total, 1080 RNA-seq libraries were prepared, yielding a dataset of over 120 billion reads (15.63 Tb) for the Quartet and MAQC samples. Here, these two sets of reference materials represent different experimental conditions of small and large biological differences. In both conditions, our aim is to provide real-world evidence on the performance of RNA-seq in terms of data quality, and the accuracy and reproducibility of gene expression and DEGs (Fig. 1b). Moreover, after excluding low quality data, the fixed analysis pipelines were applied to exclusively investigate the sources of inter-laboratory variation from the experimental processes (Fig. 1c). A total of 140 different analysis pipelines consisting of two gene annotations, three genome alignment tools, eight quantification tools following six normalization methods, and five differential analysis tools were applied to high-quality benchmark datasets selected from 13 laboratories to investigate the sources of variation from the bioinformatics processes (Fig. 1d). Based on multiple types of ground truth, the influences of factors involved in experimental and bioinformatics processes on RNA-seq performance were evaluated.

### Significant variations in detecting subtle differential expression
We combined multiple metrics for a robust characterization of RNA-seq performance in real-world scenarios: (i) quality of gene expression data using signal-to-noise ratio (SNR) based on principal component analysis (PCA)[23], (ii) the accuracy and reproducibility of absolute and relative gene expression measurements based on several ground truths, and (iii) the accuracy of DEGs based on the reference datasets (Fig. 1b). These metrics constitute a comprehensive performance assessment framework that captures different aspects of gene-level transcriptome profiling (Supplementary Notes 3.1).

PCA-based SNR values using both the Quartet and MAQC samples discriminated the quality of all gene expression data into a wide range, reflecting the varying ability to distinguish biological signals in different sample groups from technical noises in replicates (Fig. 2a). However, smaller intrinsic biological differences appeared to be more challenging to distinguish from noises, as indicated by lower average

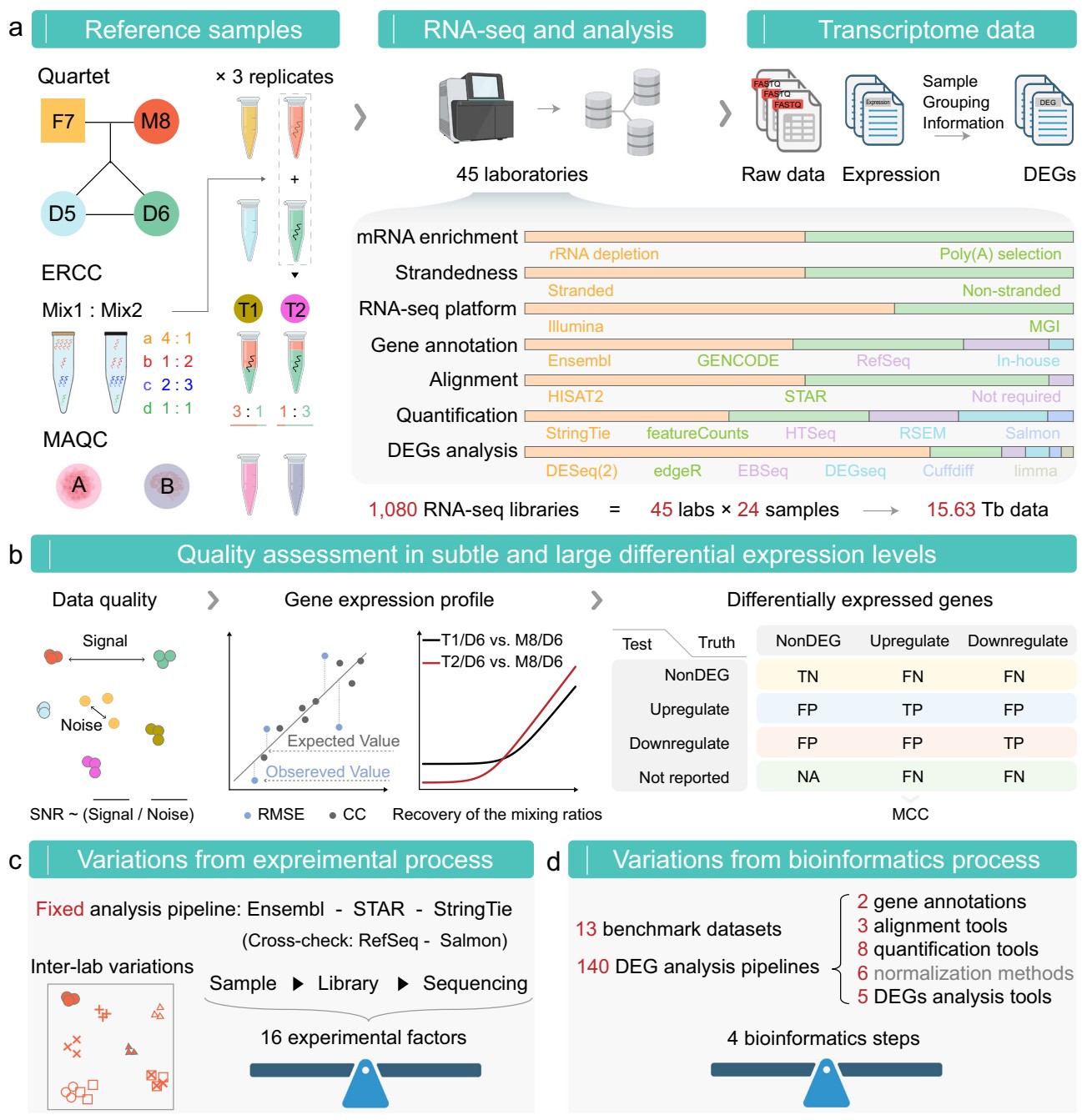

**Fig. 1 | Overview of study design. a** Two MAQC samples (A = Universal Human Reference RNA and B = Human Brain Reference RNA), two ERCC synthetic RNA mixtures, and Quartet RNA reference materials were utilized to prepare a set of samples. The M8 and D6 samples were combined with ERCC controls at manufacturer recommended amounts, and then mixed at 3:1 and 1:3 ratios to create samples T1 and T2, respectively. Each sample was prepared with three replicates, and tested by 45 laboratories with distinct protocols and analysis pipelines, resulting in a total of 1,080 libraries and 15.63 Tb of data generated. All 45 laboratories submitted expression data and differential expression calls at gene and transcript levels, while 42 laboratories submitted complete raw sequencing data. DEG, differentially expressed gene. **b** A comprehensive framework for assessment of real-world RNA-seq data, encompassing assessment of data quality using PCA-based SNR, as well as gene expression profiles and differentially expressed genes by comparing with various ground truths. **c** A fixed analysis pipeline was applied to all raw data to exclude the influence of the bioinformatic process. Then the relative contributions of experimental factors to inter-laboratory variations were investigated. **d** High-quality data from 13 laboratories were selected for the benchmarking study, and the performance of 140 differential analysis pipelines composed of two gene annotations, three alignment tools, eight quantification tools following six normalization methods, and five differential analysis tools was compared to explore the sources of variations from the bioinformatics process. SNR Signal-to-Noise Ratio, RMSE Root Mean Square Error, CC Correlation Coefficient, MCC Matthews Correlation Coefficient, TN True Negative, TP True Positive, FN False Negative, FP False Positive, DEG Differentially Expressed Gene. **a**, **c**, **d** included icons created with BioRender.com released under a Creative Commons Attribution-NonCommercial-NoDerivs 4.0 International license.

SNR values for the Quartet samples among laboratories compared to those for the MAQC samples, at 19.8 (0.3–37.6) and 33.0 (11.2–45.2), respectively (Supplementary Fig. 1). The reduced biological differences among the mixed samples led to a further decrease in the average SNR values to 18.2 (0.2–36.4). Particularly, for different laboratories, the gap between two sets of SNR values, one based on the Quartet and mixed samples and the other based on the MAQC samples, differed from 4.7 to 29.3, suggesting that diagnosing quality

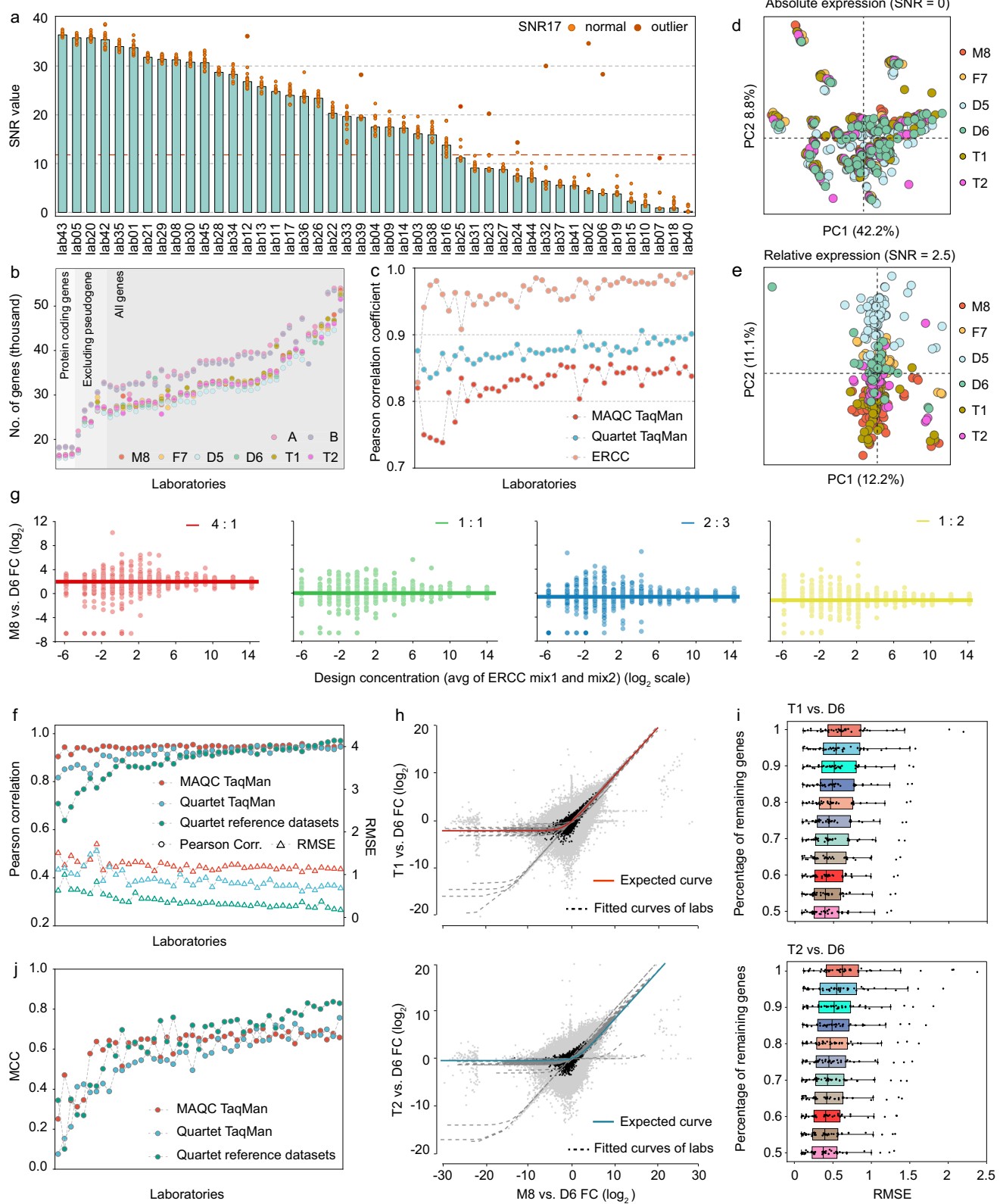

issues at subtle differential expression levels was sensitive. A total of 17 laboratories had SNR values based on the Quartet and mixed samples less than 12, which were considered as low quality. Moreover, SNR examinations allowed for identifying random library or sequencing failures in the individual replicates by calculating SNR17 from any 17 out of the 18 samples (12 Quartet and 6 mixed samples). We observed that the SNR17 values increased by six decibels compared to the

corresponding SNR18 values after excluding nine samples in nine laboratories, indicating that these nine samples were low-quality outliers (Fig. 2a).

Gene expression measurements were assessed based on the Quartet reference datasets, the TaqMan datasets, and the built-in truths including the ERCC spike-in ratios and mixed ratios of samples T1 and T2. Considering the varying gene types of interest among

**Fig. 2 | RNA-seq performance metrics for real-world laboratories. a** SNR values to measure the expression data quality submitted from 45 laboratories. Dots represented SNR values based on any 17 of the 18 samples (12 Quartet and 6 mixed samples). A dot in dark red represented an SNR17 outlier that increased over six decibels compared to its standard SNR (18-sample SNR) after excluding one low-quality sample in a laboratory, while orange dots represented acceptable SNR17 values that decreased or increased less six decibels. The red dashed line represents the SNR cutoff of 12. **b** The gene types of interest for all laboratories and the number of genes supported by at least one read in any one technical replicate (Supplementary Notes 3.3). **c** Assessment of absolute expression based on the TaqMan datasets and ERCC concentrations on the $\log_2$ scale. **d** PCA plots on expression data from 45 laboratories in absolute and (**e**) relative expression levels. The circles with same color represent all laboratories for each sample. **f** Assessment of relative expression using Pearson correlation coefficient and the root mean square error (RMSE) based on the Quartet reference datasets and the TaqMan datasets on the $\log_2$ scale. **g** ERCC ratios can be recovered increasingly well at higher expression levels. **h** A consistency test for recovering the expected sample mixing ratio in T1 and T2. The red and cyan solid line traces the expected curve after mRNA/total-RNA shift correction. The gray dashed lines indicate the fitted curves from 45 laboratories. The ERCC genes are shown in black, and the other human genes are shown in gray. **i** The ability to recover expected mixing ratios was measured using RMSE between the observed and expected relative expression values. Box plots present RMSE values for all 45 laboratories, and data are presented as median values (center lines) and the upper and lower quartiles (box limits). Colors representing the different percentages of remaining genes after progressively filtering out genes with low fold changes. **j** Assessment of differentially expressed genes using Matthews Correlation Coefficients (MCC) based on the Quartet reference datasets and the TaqMan datasets. FC fold change, QC quality control.

laboratories (Fig. 2b), only protein-coding genes were included to facilitate comparisons between laboratories. In absolute gene expression levels, all laboratories exhibited lower Pearson correlation coefficients at 0.825 (0.738–0.856) with the MAQC TaqMan datasets (830 results for protein-coding genes), compared to those at 0.876 (0.835–0.906) with the Quartet TaqMan datasets (143 results for protein-coding genes) (Fig. 2c). Correlations with the nominal concentrations of the 92 ERCC spike-in RNAs were consistently high for all laboratories with an average correlation coefficient of 0.964 (0.828–0.963). More ERCC based assessments are shown in Supplementary Notes 3.2. These results indicate that accurate quantification of a broader set of genes is more challenging, highlighting the importance of large-scale reference datasets for performance assessment. We also focused on the absolute expression for other gene types, and observed that small non-coding RNAs (sncRNA) exhibited the largest inter-laboratory variations, followed by pseudogenes, long non-coding RNAs (lncRNA), and immunoglobulin/T cell receptor segments (Supplementary Fig. 2), which appeared to be associated with gene features specific to each type, such as gene lengths and gene expression levels (Supplementary Figs. 3–4).

Relative expression measurements are more reliable than absolute expression measurements, but they still present challenges when identifying subtle differential expression. The variations in relative expression across laboratories decreased compared to those in absolute expression, as indicated by the fact that samples tended to cluster based on the source sample rather than the laboratory in PCA plots (Fig. 2d–e). Noticeably, laboratories still exhibited considerable variations in relative expression exceeding the small biological difference among the Quartet samples (Supplementary Fig. 5). When compared with multiple reference datasets, relative expression exhibited higher correlation coefficients than absolute expression (Supplementary Fig. 6), whereas the accuracy of relative expression in detecting the Quartet samples was lower than that of the MAQC samples. Laboratories exhibited lower average correlation coefficients of 0.865 (0.288–0.978) and 0.860 (0.488–0.944) with the Quartet reference datasets (23790 results for protein-coding genes) and the Quartet TaqMan datasets, respectively. In contrast, the average correlation coefficient was 0.927 (0.778–0.949) with the MAQC TaqMan datasets (Fig. 2f). It is noteworthy that the root mean square error (RMSE) values between laboratories and the Quartet reference datasets were consistently lower, reflecting the systematic deviations between RNA-seq and TaqMan RT-qPCR assays but not between RNA-seq and the Quartet reference datasets (Fig. 2f). In addition, the two built-in truths allowed for a complementary examination of the accuracy and reproducibility of the relative expression across 92 ERCC RNAs and all detected human genes, respectively. Our results revealed the impact of low gene expression and subtle differential expression on relative expression measurements. The expected ERCC spike-in ratios were more accurately recovered for high-concentration ERCC genes compared to low-concentration genes (Fig. 2g). The mixing ratios in the mixed samples were recovered well in most laboratories (Fig. 2h). Laboratories that failed to recover the mixing ratio demonstrated the presence of outliers (Supplementary Fig. 7), which are typically caused by the erroneous detection or calculation of low-expressed genes (Supplementary Fig. 8). By stepwise filtering genes with low fold changes, the RMSE values between the observed and expected fold changes decreased, indicating challenges in detecting genes with small expression differences (Fig. 2i).

The accuracy of DEG identification, as assessed based on the Quartet reference datasets and the TaqMan datasets for Quartet and MAQC samples, also exhibited variations across laboratories. Due to the varied number of genes inputted for differential expression analysis, true positives ranging from 0.03% to 78.6%, from 1.2% to 82.0%, and from 0.2% to 52.9% of the three reference datasets, respectively, were not reported across laboratories. Consequently, we categorized these instances as false negatives, and employed a penalized Matthews Correlation Coefficient (MCC) to assess the accuracy of DEG identification (Fig. 1b and Supplementary Notes 3.1). The MCC values based on the Quartet reference datasets and the Quartet TaqMan datasets were more dispersed among laboratories, ranging from 0.100 to 0.837 and from 0.075 to 0.756, respectively (Fig. 2j). In contrast, the MCC values based on the MAQC TaqMan datasets ranged from 0.251 to 0.702. Importantly, the relatively low MCC values in certain laboratories could be explained by several factors (Supplementary Data 2). For example, in the case of lab18, the expression data exhibited a SNR of 0.9, indicating that the low-quality library preparation or sequencing processes resulted in uninformative RNA-seq data for differential analysis. The lab03 and lab04 demonstrated low accuracy of fold change determination, impacting the reliability of the DEG detection. Additionally, due to the diversity in their experimental designs and methodologies, different thresholds for filtering low-expression genes and DEG identification were chosen by laboratories, which led to variations in the number of DEGs and impacted the accuracy. For example, lab45 filtered genes with FPKM < 1 in all replicates before differential expression analysis, leading to true-positive genes present in the reference datasets being filtered out. Lab26 employed a stringent threshold of $p < 0.001$ while lab07 only utilized $Q \leq 0.05$ without incorporating $\log_2 FC$ for DEG identification, which resulted in either few or excessive DEGs and consequently lower accuracy.

**Quality control check for filtering of low-quality RNA-seq data**
Complete RNA-seq raw data were available in 42 laboratories. We first assessed the sequencing quality properties (pre-alignment) for the Quartet and MAQC samples, including sequencing depth, base quality, GC content, and duplication rate (Supplementary Data 3). The number of reads ranged from 39.4 Mb to 418.8 Mb for the Quartet samples and from 40.9 Mb to 424.2 Mb for the MAQC samples across laboratories (Supplementary Fig. 9). Within the same laboratory, different samples exhibited variations in sequencing depth, particularly noticeable for laboratories with higher average sequencing depths. Given that

different flowcells or lanes can lead to variations in total reads counts, we compared 16 laboratories that assigned 24 libraries to two or more lanes to other laboratories that assigned libraries to a single lane, and observed no increased variations (Supplementary Fig. 10). Therefore, inter-sample variations were considered to be due to difficulties of equimolar pooling[22].

Both Quartet and MAQC samples exhibited high Q30 scores, ranging from 88.4% to 96.6% and from 88.3% to 96.7%, respectively, reflecting the high quality of base calling (Supplementary Fig. 11). Most laboratories showed a biased quality score distribution in the first 1–10 bases for the Quartet and MAQC samples (Supplementary Figs. 12–13), which was attributed to lower signal intensities for the first sequencing cycle and insufficient correction for factors such as phasing and cross-talk in the initial bases during base calling[25,26]. We also observed that the quality scores of forward reads (the first sequenced reads) were generally higher than those of reverse reads (the later sequenced reads), particularly in laboratories using the Illumina sequencing platform. This was attributed to the decreased cluster size and higher number of errors due to more amplification steps, and reduced DNA polymerase activity before sequencing the reverse reads[27].

GC content bias was found across laboratories, with the average GC content ranging from 42.3% to 54.2% for the Quartet samples and from 42.4% to 52.9% for the MAQC samples. Such laboratory-specific GC content bias, primarily caused by different sites of library preparation[28], was more noticeable than the sample-specific GC content bias (Supplementary Figs. 14–15). Unusual GC content presents inherent challenges, as GC-poor genes (<35%) tended to exhibit more variable expression levels between laboratories than genes with medium or high (> 65%) GC content (Supplementary Fig. 16).

The average duplication rates of the sequencing reads varied significantly across laboratories, ranging from 4.2% to 73.4% for the Quartet samples and from 5.0% to 75.5% for the MAQC samples (Supplementary Fig. 17), with increased duplication rates correlating with higher sequencing depth (Supplementary Fig. 18a). Additionally, the mRNA enrichment method also influenced the duplication rates, with the Poly(A) selection method demonstrating higher duplication rates, similar to observations in other studies (Supplementary Fig. 18b)[15]. Noticeably, the rRNA depletion method, due to capturing a large amount of non-coding RNA, showed a significant decrease in duplication rates after removing reads aligned to non-coding regions, indicating difficulties in detecting non-coding RNAs (Supplementary Fig. 19). Seven laboratories exhibited an average duplication rate exceeding 30%, surpassing the typical duplication levels observed in prior research[15,29,30]. These data were characterized by high duplication rates at both low and high expression levels (Supplementary Fig. 20a). Excessively duplicated reads at low expression levels were recognized to originate from PCR amplification bias[31,32]. Within three of these seven laboratories, we also observed large dispersions in duplication rates for all samples, which exhibited a strong negative correlation with library concentration with Spearman correlation coefficients ranging from -0.886 to -0.986, highlighting the influence of the lack of library complexity (Supplementary Fig. 20b)[32].

We next performed the post-alignment quality control after mapping the raw reads using STAR. All laboratories exhibited a high overall alignment rate, ranging from 90.69% to 98.7% for the Quartet samples and from 92.1% to 98.9% for the MAQC samples (Supplementary Fig. 21). The slightly lower uniquely mapping rate was noticeable in the Quartet samples in comparison to the MAQC samples, with average mapping rates of 89.7% (80.9–95.4%) and 92.0% (84.1–96.0%), respectively. This was similar to the common characteristics observed when comparing clinical samples with the MAQC samples[20]. The multi-mapping rate seemed to be associated with the mRNA enrichment methods. The rRNA depletion method resulted in higher average multi-mapping rates than the Poly(A) selection method (Supplementary Fig. 22), possibly due to the capture of a greater

number of small non-coding RNAs with high sequence similarity[33]. Meanwhile, a high multi-mapping rate was consistently correlated with a higher mismatch rate. The percentage of aligned reads mapping to annotated exons is directly related to expression quantification, and is, therefore, a critical quality metric. The Poly(A) selection method consistently showed a higher median percentage of exonic reads at 84.5% and 80.9%, compared to the rRNA depletion method at 46.3% and 44.1% for the Quartet and MAQC samples, respectively (Supplementary Figs. 23–24).

Additionally, we performed the sample-level quality control to identify any problematic samples. First, we examined SNR after applying the fixed data analysis pipeline to eliminate the effects of different bioinformatics workflows used by the laboratories. Data from 14 laboratories were classified as low quality (Supplementary Fig. 25). Second, based on the fact that the single nucleotide polymorphisms (SNPs) among technical replicates are theoretically identical, we calculated the pair-wise correlation measures on the variant allele frequencies (VAF) of the SNPs for sample-identity checks. All laboratories exhibited higher Pearson correlations between any two technical replicates, than those between-sample group comparisons, indicating no sample swaps or mislabeling (Supplementary Fig. 26). Despite this, the correlation coefficients between replicates of the MAQC samples were relatively low, which were similar to values of 0.53 and 0.39 observed between replicates of MAQC A and B, respectively, in the RNA-seq data from the MAQC-III study[12]. This may be attributed to the increased genome complexity of the MAQC samples A and B, which were prepared by mixing total RNA samples from ten different cancer cell lines and from the brain of 23 donors, respectively[12]. Finally, based on percentage of reads mapped to ERCC reference sequences allowed for the identification of problematic samples or libraries. In four samples (MAQC A, B, and Quartet F7, D5) without ERCC spike-ins, we observed reads counts ranging from 1 to 213,467 mapped to ERCC genes across 38 laboratories (Supplementary Fig. 27). Four laboratories showed more than 0.005% of reads mapped to ERCC genes in a specific sample. Particularly, lab10 exhibited an exceptionally high fraction of ERCC reads in the two replicates of MAQC sample A, accounting for 0.8% and 0.06% of the exonic reads. This indicates potential cross-contamination, possibly due to sample, library, or barcode contamination during library preparation[34], or misallocation of barcodes and carry-over contamination from previous samples during the sequencing process[34–36].

Using the above multiple metrics, including pre-alignment, post-alignment, and sample-level quality metrics, RNA-seq data from 26 laboratories passed all criteria and were used for subsequent analysis, whereas the other data from remaining 16 laboratories were flagged as low quality and excluded from subsequent analysis to mitigate the impacts of poorer library or sequencing quality, as well as sample cross-contamination (Supplementary Data 3) (Fig. 3).

## Sources of variation from the experimental processes

The significant inter-laboratory variations necessitated the investigation of the sources. Here, we focused on the magnitude of variations across 26 laboratories in absolute or relative gene expression introduced by each RNA-seq step. To exclusively investigate variations from the experimental processes, we employed a uniform data analysis pipeline for all RNA-seq raw data, involving fastp for data pre-processing, Ensembl gene annotation, STAR for read alignment, and StringTie for gene quantification. When compared to the original expression data, the variations in SNR and the accuracy of gene expression measurements decreased across laboratories, with a significant improvement observed in some laboratories (Supplementary Figs. 28–29). Similar results were observed when an alternative gene quantification pipeline (for example, RefSeq and Salmon) was used for gene quantification (Supplementary Fig. 30). These findings indicated that the fixed pipeline effectively reduced variations introduced by various

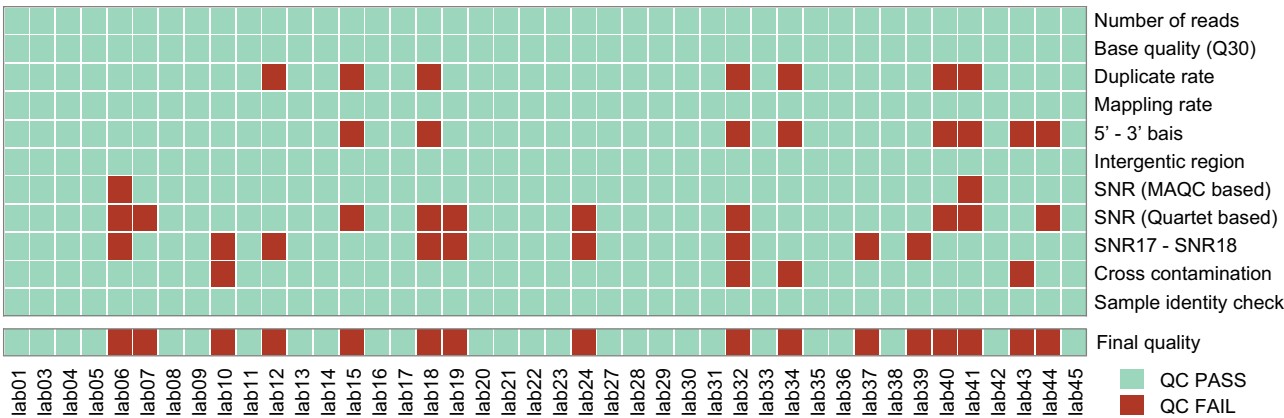

**Fig. 3 | Quality flags of RNA-seq data.** Multiple quality metrics were used, including the number of sequencing reads, base quality (Q30), the percentage of reads mapped to the human genome (Mapping rate), gene body bias (5'–3' bias), the percentage of mapped reads that were located in the intergenic region (Intergenic region), SNR based on the MAQC and Quartet samples, the difference between the SNR calculated from any 17 out of 18 samples and the SNR calculated from all 18 samples (SNR17-SNR18), the percentage of reads mapped to ERCC sequences in samples without ERCC mixtures (Cross contamination), SNP-based sample-identity check, and final quality flag.

bioinformatics processes employed by the laboratories. The variations arising from different RNA processing methods, library preparation protocols, and sequencing platforms among laboratories represent 'experimental noise'.

The inter-laboratory variations in absolute gene expression levels introduced by experimental processes were significant in both Quartet (Fig. 4a) and MAQC samples (Fig. 4b), especially impacting subtle differential expression measurements. We quantified the relative contribution of technical and biological factors to the total variations by principal variance component analysis (PVCA) based on absolute expression data from all laboratories for all samples. A total of 15 factors from the experimental process were considered (Supplementary Data 4), which introduced significantly greater variations than biological differences among the Quartet samples (85.1% vs. 5.8%), with mRNA enrichment methods and strandedness as the primary sources (Fig. 4c). Additionally, other factors, including library preparation kits, read lengths, the number of exonic reads, RNA inputs, and their interactions also contributed to more than 25% of the variations. These factors corresponded to clustering patterns of gene expression consistency across laboratories (Fig. 4f). In contrast, while the MAQC samples revealed similar sources, variations from all experimental factors were lower than biological differences between the MAQC samples (38.2% vs. 61.2%) (Fig. 4d). Employing the alternative analysis pipeline (RefSeq and Salmon) also revealed similar sources of variations, with these introduced variations representing approximately 15-fold and 0.4-fold of biological differences in the Quartet and MAQC samples, respectively (Supplementary Fig. 31).

In relative gene expression levels, the proportion of variation attributed to experimental factors decreased to below 20% for the Quartet and MAQC samples, respectively, which was observed when employing both fixed analysis pipelines (Supplementary Fig. 32). This indicated that relative expression could effectively correct for the influence of experimental factors. The increased consistency of relative gene expression between any two laboratories compared to absolute expression further confirmed this (Fig. 4e–f).

## Sources of variation from the bioinformatics processes
To assess the sources of variation from the bioinformatics process, high-quality data for the Quartet and MAQC samples from 13 laboratories served as benchmark datasets, encompassing 13 different library preparation protocols, seven sequencing platforms, and a wide range of sequencing depths spanning 42.6 Mb to 425.3 Mb to mitigate bias (Methods). Following commonly used transcriptomic profiling

pipelines in real-world settings, two gene annotations, three genome alignment tools, and eight expression quantification tools were incorporated into the analysis, resulting in 28 combined quantification pipelines. Subsequently, six representative normalization methods were systematically compared (Supplementary Fig. 33). Variations caused by different combinations of analysis tools represent 'bioinformatics noise'.

In absolute gene expression levels, each bioinformatics step introduced variations, with a greater impact on the detection of subtle differential expression (Fig. 5a–b). For the Quartet samples, all bioinformatics steps and their interactions collectively introduced significantly greater variations than the intrinsic biological differences (75.1% vs. 5.6%). Normalization methods were the primary source of variations, followed by quantification tools, alignment tools, and gene annotation types (Fig. 5a). However, the MAQC samples revealed smaller variations introduced from different bioinformatics steps than their biological differences (34.0% vs. 56.7%) (Fig. 5b).

The relative gene expression also helped reduce variations from bioinformatics processes, as indicated by the increased consistency of relative gene expression across different analysis pipelines compared to absolute expression (Fig. 5c–d). Furthermore, the contribution of bioinformatics factors to variations in relative expression levels decreased over 60% and 30% in the Quartet and MAQC samples, respectively (compare Supplementary Fig. 34 with Fig. 5), suggesting that the relative expression calculations could correct for the influence of different analysis tools. However, there were still 28.4% of the variations from the bioinformatics process for the Quartet samples, suggesting inherent performance differences among various analysis tools.

## Best practices for experimental designs
The low quality of experimental execution significantly influences the RNA-seq performance. RNA-seq data from 16 laboratories, failing multiple quality metrics, were considered low experimental quality in library preparation or sequencing processes (Fig. 3). Based on four types of ground truth, these laboratories exhibited lower Pearson correlation coefficients or higher RMSE for absolute and relative gene expression measurements, as well as lower MCC for DEG identification, compared to the other 26 laboratories (Supplementary Fig. 35). Noticeably, these instances of low quality were unrelated to the choice of experimental methods, as they were distributed across various experimental workflows. Therefore, our results highlighted the importance of multidimensional quality control in experimental design.

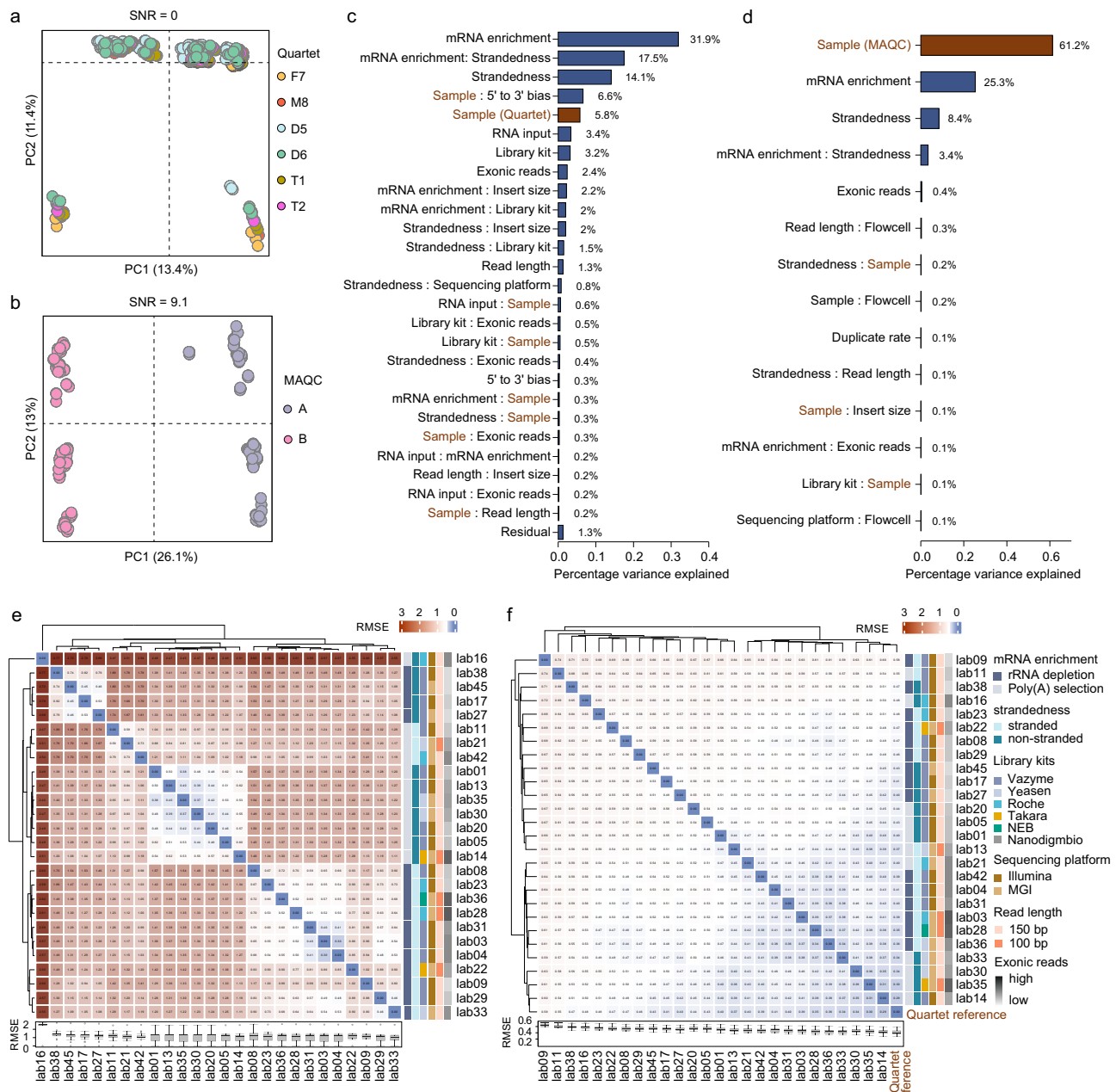

**Fig. 4 | Sources of variation from the experimental processes.** Scatterplots of PCA on RNA-seq data from 26 laboratories for the (**a**) Quartet samples, and (**b**) MAQC samples after applying the fixed analysis pipeline. The circles of the same color represent all replicates across all laboratories for each sample. Principal variance component analysis quantifies the proportion of variance explained by each experimental factor for the (**c**) Quartet samples, and (**d**) MAQC samples. Heatmap and hierarchical clustering of laboratories based on the root mean square error (RMSE) at (**e**) absolute expression levels, and (**f**) relative expression levels for the Quartet samples. The average gene expression values across three replicates of the four Quartet samples for the intersecting genes between each laboratory's data and the Quartet reference datasets ($n = 30,956$) were used to calculate the RMSE between any two laboratories or between a laboratory and the Quartet reference datasets. The box plots present the RMSE values for 26 laboratories, and data are presented as median values (center lines) and the upper and lower quartiles (box limits).

Different experimental protocols tended to influence RNA-seq performance, which should be considered during experimental design. Based on multiple types of ground truths, we assessed the influence of experimental factors using RNA-seq data from 26 laboratories in four aspects: data quality, absolute and relative gene expression, and DEG identification (Fig. 6). (i) For expression data quality, the Poly(A) selection method exhibited higher SNR values than the rRNA depletion method. Considering the differences in gene types captured by the two methods, we further examined SNR values for different gene types and observed that the Poly(A) selection method primarily exhibited higher SNR values for protein-coding genes.

Conversely, for other gene types, particularly sncRNA, the rRNA depletion method demonstrated significantly higher SNR, indicating a more accurate capture of biological differences in these RNAs (Supplementary Fig. 36). (ii) For absolute expression levels, the influences of mRNA enrichment method and strandedness were observed, with the rRNA depletion method and stranded-specific libraries exhibiting higher correlation coefficients with the reference datasets. (iii) For relative gene expression and DEG levels, a higher number of reads mapped to the exonic regions showed a strong or moderate correlation with improved accuracy, likely due to more reliable detection of lowly expressed genes[37]. To validate their impacts, we down-sampled

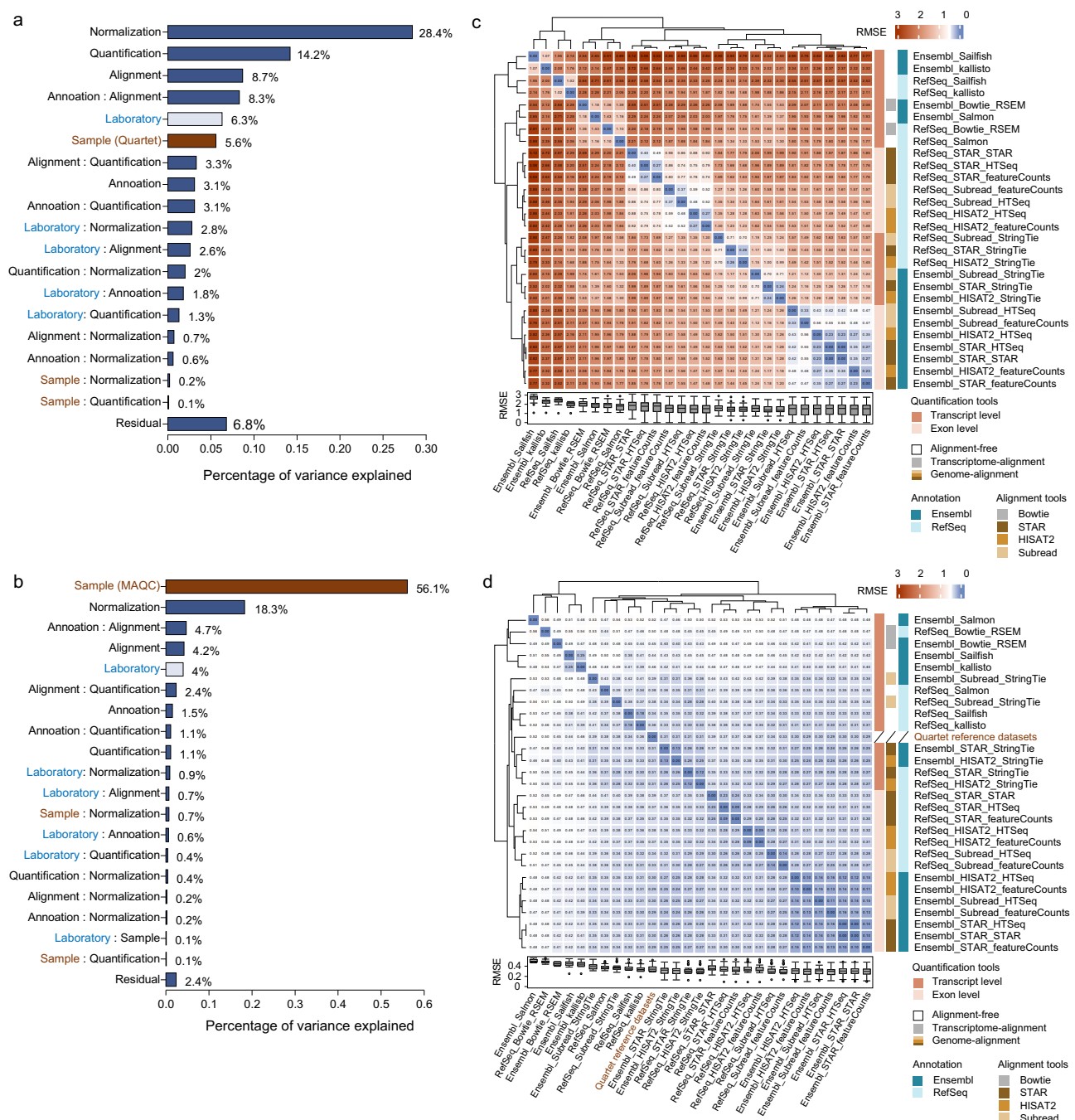

**Fig. 5 | Sources of variation from the bioinformatics process.** Principal variance component analysis quantifies the proportion of variance explained by each data analysis step for the (**a**) Quartet samples, and (**b**) MAQC samples. Heatmap and hierarchical clustering of 28 gene quantification pipelines based on the root mean square error (RMSE) at (**c**) absolute expression levels, and (**d**) relative expression levels. The average gene expression values across three replicates of the four

Quartet samples for the intersecting gene results between the 13 benchmark datasets and the Quartet reference dataset ($n = 359,842$) were used to calculate the RMSE between any two quantification pipelines or between a quantification pipeline and the Quartet reference dataset. The box plots present the RMSE values for all 28 quantification pipelines, and data are presented as median values (center lines) and the upper and lower quartiles (box limits).

RNA-seq data to different depths and drew similar conclusions. Specifically, lower read depths resulted in decreased accuracy of relative gene expression and DEG identification, while exerting a relatively minor overall impact on SNR and absolute expression levels. However, when sequencing depth decreased to extremely low levels, such as 10 Mb, SNR and accuracy of absolute expression detection also significantly declined (Supplementary Fig. 37). Furthermore, the MAQC TaqMan datasets revealed significant differences in the accuracy associated with read lengths. To further explore their impacts, we trimmed sequencing reads to different lengths ranging from 25 bp to

150 bp across four laboratories, revealing that longer sequencing reads tended to exhibit higher accuracy, especially in relative gene expression measurements, and either similar or higher accuracy in DEG identification (Supplementary Fig. 38).

Additionally, we also observed statistically significant differences in accuracy among laboratories using different RNA inputs, library kits, and sequencing platforms, based on a singular type of ground truth (Supplementary Fig. 39), which have been recognized as influencing factors previously[15,17,38]. In addition, for insert size, a factor considered to influence gene or isoform identification and quantification, our results

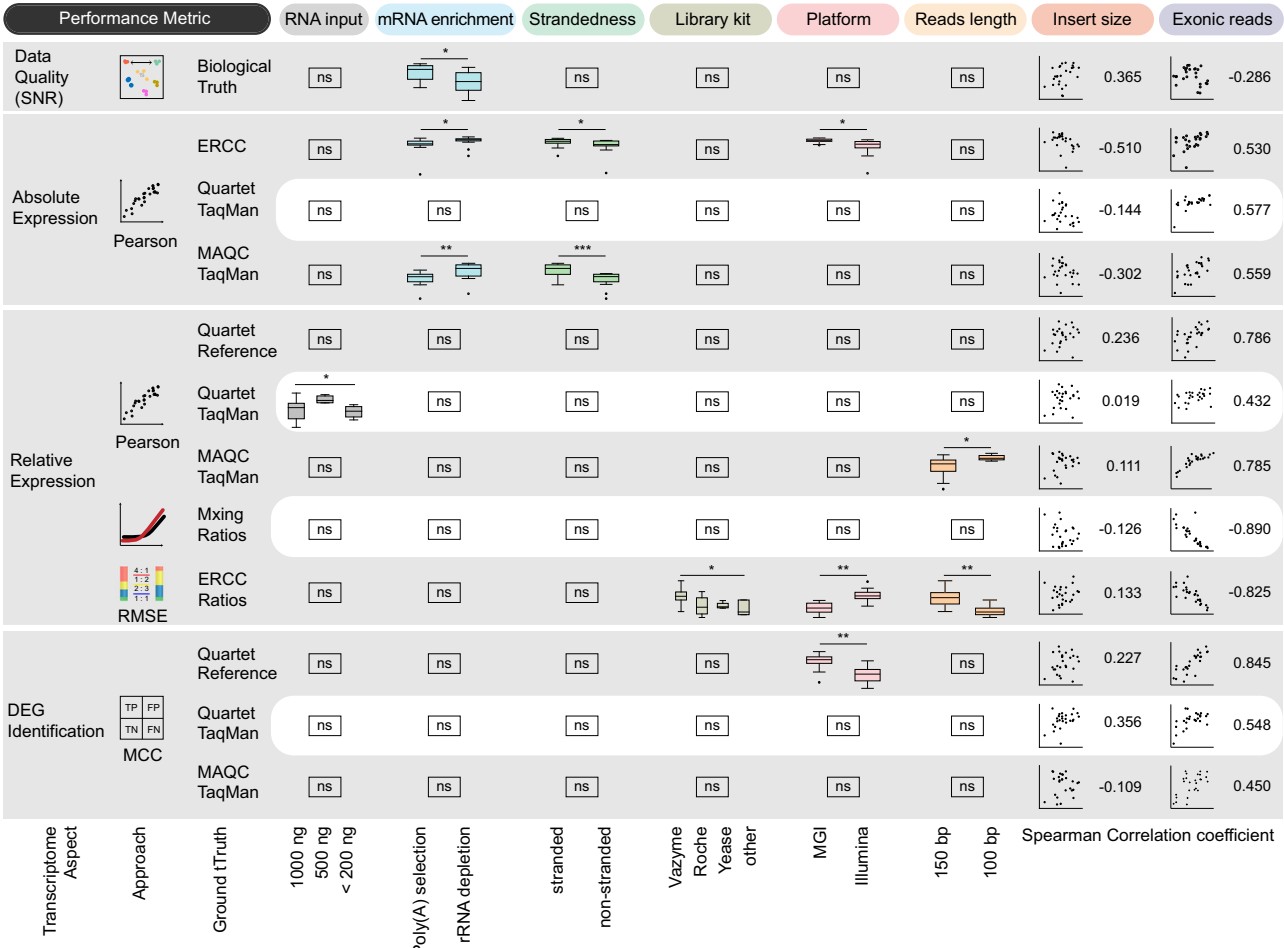

**Fig. 6 | The influence of experimental factors based on different performance metrics.** Performance metrics included SNR for data quality, Pearson correlation coefficient for accuracy of absolute and relative expression, the root mean square error (RMSE) for recovery of mixing ratios, and the Matthews correlation coefficient (MCC) for the accuracy of DEG identification. Exonic reads indicate the number of mapped reads located in the exonic regions. Exonic reads were considered as an experimental factor as they could be determined by the total sequencing depth and the mRNA enrichment method chosen. The impact of exonic reads on data quality and accuracy is evaluated using Spearman correlation analysis. The box plots present performance metric for 26 laboratories, which are divided into different groups based on the protocols used at each experimental step, and data are presented as median values (center lines) and the upper and lower quartiles (box limits). Significance testing among groups was conducted based on normal distribution assumptions using one-way analysis of variance (ANOVA) and paired *t*-tests; or, in cases where normal distribution was not observed, independent samples were subjected to Kruskal-Wallis test and Mann−Whitney *U* test. * indicates a two-sided $p < 0.05$, ** indicates a two-sided p-value < 0.01, and *** indicates a two-sided p-value < 0.001. The exact p-values are shown in Supplementary Fig. 39. The scatter plots represent the correlation between performance metrics and insert size or the exonic reads across the 26 laboratories. ns not significant, TN True Negative, TP True Positive, FN False Negative, FP False Positive.

demonstrated a poor correlation with gene quantification accuracy under the involvement of other factors[22]. The impact of these factors needs further validation during method development through focused experimental designs controlling for other influencing factors.

In addition to gene quantification aspects, certain experimental factors such as sequencing depth and read length, were considered to influence exon-exon junction detection, which implies an impact on isoform and alternative splicing identification[39,40]. We observed that even lower sequencing depth was sufficient to detect known junctions, and increasing the sequencing depth further facilitated the identification of novel junctions (Supplementary Fig. 40). The impact of read length was similar, as known junctions can still be detected when the read length was reduced to 25 bp, while novel junctions were almost undetectable (Supplementary Fig. 41).

### Best practices for bioinformatics designs

To obtain an optimal analysis pipeline for gene-level quantification and differential expression measurements, we sequentially evaluated the performance of 140 combined analysis pipelines with regard to alignment quality, quantification accuracy, normalization effectiveness, low-expression gene filtering efficacy, and accuracy of DEG identification.

We first evaluated the influence of six alignment approaches combined with two annotations and three genome alignment tools in terms of sequence alignment and splice junction discovery. In comparison to the RefSeq annotation, the Ensembl consistently resulted in higher uniquely mapping rates and lower multi-mapping rates (Supplementary Fig. 42a). STAR exhibited the highest overall mapping rate as well as uniquely mapping rate. STAR either mapped or discarded the paired reads, avoiding the alignment of unpaired single-end reads (Supplementary Fig. 42b). HISAT2 and Subread had comparable uniquely mapping rates, yet HISAT2 tended to have slightly higher multi-mapping rates in most samples, resulting in higher overall mapping rates. Subread displayed a higher tolerance of accepting mismatch, primarily concentrating in fewer mismatched bases (Supplementary Fig. 42b). Given that Subread did not detect exon-exon

junctions, we compared the junctions from STAR and HISAT2. The Ensembl annotation, being more complex, led to the validation of a greater number of junctions (Supplementary Fig. 43). For these known junctions, two alignment tools did not exhibit significant differences, whereas HISAT2 identified more completely novel junctions. Most of novel junctions were not reliable, indicated by significantly decreased number after applying a counts-based threshold (Supplementary Fig. 44).

We next assessed the performance of 28 gene quantification pipelines, consisting of six genome alignment approaches and eight quantification tools (Supplementary Fig. 33). Both absolute and relative gene expression from these pipelines exhibited clustering based on principles of quantification tools: genome-alignment quantification pipelines, and transcriptome-alignment or alignment-free quantification pipelines (Fig. 5c–d). Gene annotation and alignment tools also influenced the clustering patterns. In particular, pipelines employing the same annotation demonstrated notable clustering in the PCA plots (Supplementary Fig. 45). We further examined the accuracy of these pipelines for relative expression measurements by comparing them to three reference datasets and the two types of built-in truths, revealing similar clustering patterns based on the accuracy metric (RMSE) (Fig. 7a). Our results also demonstrated that the performance of each bioinformatics step was interdependent. Specifically, the performance of gene annotation varied depending on the combination with different quantification tools, with Ensembl annotation exhibiting higher or similar accuracy when combined with exon-level quantification tools, while RefSeq exhibited higher accuracy when combined with transcript-level quantification tools (Supplementary Fig. 46). The three different alignment tools demonstrated similar accuracy, especially when using Ensembl annotation and quantification tools such as featureCounts and HTSeq. When using StringTie for quantification, STAR and HISAT2 outperformed Subread (Supplementary Fig. 47). The influence of quantification tools was notable, with genome-alignment quantification tools consistently showing higher accuracy (Fig. 7c–e). Alignment-free quantification tools exhibited higher accuracy when selecting RefSeq gene annotation, particularly Salmon, showing performance similar to genome-alignment quantification pipelines. RSEM exhibits moderate accuracy, outperforming alignment-free quantification tools combined with Ensembl annotation. Overall, our performance ranking of all quantification pipelines for relative gene expression measurements based on four types of ground truth supported the selection of Ensembl gene annotation (or RefSeq annotation when using StringTie) and genome-alignment quantification strategy for gene quantification purposes (Supplementary Fig. 48).

We converted the raw counts from the 28 quantification pipelines using six normalization methods, followed by an assessment of expression data quality using PCA-based SNR. Trimmed mean of M values (TMM), and DESeq normalization methods appeared to improve the raw counts most effectively, while upper quartile (UQ) normalization exhibited the poorest improvement (Supplementary Fig. 49). Then we examined the gene expression distribution for all normalization methods, and found that the median gene expression from DESeq was the highest, followed by TMM, total counts (TC), and UQ, while fragments per kilobase million (FPKM) and transcripts per million (TPM) had similarly low levels (Supplementary Fig. 50).

The setting of low-expression gene filtering conditions may affect the interpretation of differential expression calls (Supplementary Data 2). To elucidate the impact of filtering conditions on the performance of differential analysis, we evaluated six filtering methods and various threshold values (the percent of filtered genes ranging from 0 to 70%) across five differential analysis tools, utilizing four RNA-seq datasets representing different sequencing depth levels (Supplementary Fig. 33) (Methods). Across all six filtering methods, elevating the threshold values resulted in an increase in both the DEG number and the true positive rate (TPR) until they reached their respective peak

values, accompanied by a slight yet acceptable decrease in precision (Supplementary Fig. 51). Such threshold effects were observed for four differential analysis tools, including edgeR, DESeq2, limma, and DEGseq, except for EBSeq, which employed stringent internal filtering criteria (Supplementary Fig. 52). Overall, the six filtering methods led to general consistency in terms of the maximum number of DEGs and the highest TPR across data from all laboratories (Supplementary Figs. 53–54). Thus, the key consideration shifts to the determination of optimal threshold value. Opting for a threshold value balancing the TPR and precision appears to be an effective approach, but the lack of benchmark datasets for assessing sensitivity or precision presents a challenge in practice. In contrast, calculating the maximum number of DEGs is practical. Although there were slight differences between the thresholds based on the maximum number of DEGs and the highest TPR, especially in the Quartet samples (Supplementary Fig. 55), the resulting TPR values corresponding to these two thresholds were highly consistent (Supplementary Fig. 56).

After applying a series of threshold values to filter low-expression genes, we compared the optimal performance of five differential analysis tools with different choices of quantification pipelines, which contributed to 140 differential analysis pipelines (Supplementary Fig. 33). We assessed the DEG accuracy based on three reference datasets and investigated the influences of each bioinformatics steps on the accuracy (Fig. 8a–e). edgeR and DESeq2 consistently outperformed other tools, with DEGSeq and limma slightly lower, and EBSeq being the lowest (Fig. 8d). As another accuracy measure, the area under the receiver operating characteristic curve (AUC) was compared across all differential expression analysis pipelines, which captured the statistical discrimination capability of the DEGs. Similarly, edgeR outperformed the other tools, and DESeq2 also exhibited relatively high AUC values (Fig. 8e and Supplementary Fig. 57). The influences of annotation, alignment, and quantification steps on the accuracy of DEG identification were particularly significant when detecting subtle differential expression among the Quartet samples, showing larger dispersions in MCC values (Fig. 8d). Ensembl annotation consistently demonstrates higher or comparable MCC compared to RefSeq (Fig. 8a), and three alignment tools showed similar MCC (Fig. 8b). Genome-alignment quantification tools consistently yielded higher accuracy of DEG identification, with Salmon being an exception among alignment-free quantification tools, displaying higher accuracy similar to exon-level quantification tools (Fig. 8c). RSEM exhibited lower accuracy for the Quartet samples but higher accuracy for the MAQC samples. The accuracy ranking of 140 pipelines was provided in Supplementary Data 5.

## Discussion

As part of the Quartet project, this study represents the most extensive cross-laboratory examination of real-world RNA-seq data and analysis outcomes to date, employing the Quartet and MAQC RNA reference materials. Through the systematic assessment of transcriptome data from 45 laboratories and the investigation of factors involved in 26 experimental processes and 140 bioinformatics pipelines based on several types of ground truth, we attempted to address several questions: (i) the performance of real-world RNA-seq in detecting subtle differential expression; (ii) the sources of inconsistency among laboratories; and (iii) the recommended practices to enhance the accuracy of RNA-seq in practical applications (Supplementary Table 1).

This study, for the first time, revealed noteworthy real-world interlaboratory variations in transcriptome profiling performance, especially when detecting subtle differential expression among the Quartet samples. This prompts a reconsideration of the actual performance, which may not be as robust as in previous studies conducted under rigorously controlled RNA-seq workflows[14,15,22]. First, we examined the data quality from multiple perspectives and identified a total of 16 batches of low-quality data, which seems to be more common than

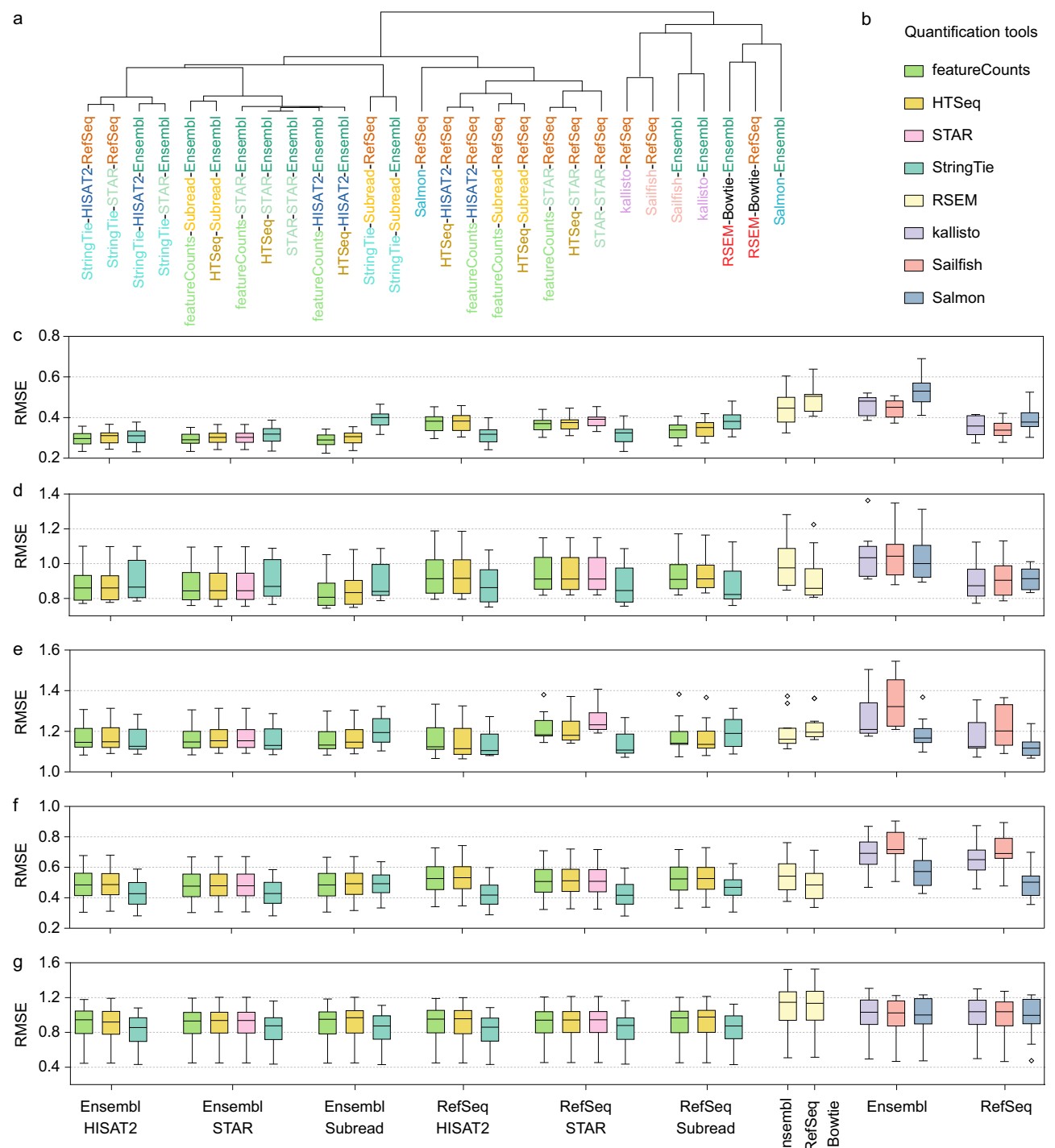

**Fig. 7 | Accuracy of gene quantification pipelines. a** Clustering of 28 quantification pipelines based on accuracy metrics (RMSE) derived from comparisons to four types of ground truth. **b** The color legend for c-g. The accuracy of all 28 quantification pipelines was assessed based on the (**c**) Quartet reference datasets, (**d**) Quartet TaqMan datasets, (**e**) MAQC TaqMan datasets, (**f**) built-in truth of mixing ratios in T1 and T2, and (**g**) built-in truth of ERCC spike-in ratios. The accuracy of quantification tools is compared under the same conditions of gene annotation and alignment tools. Box plots represent the RMSE values for 13 benchmark datasets, and data are presented as median values (center lines) and the upper and lower quartiles (box limits). Diamonds indicate outliers.

previously reported[15,23]. Second, absolute expression levels exhibited substantial inter-laboratory variations, while relative expression measurements were more reliable, which have become widely recognized[14]. However, this study also revealed greater variations in relative expression especially when detecting subtle differential expression. Certain laboratories exhibited low consistency with reference datasets and poor recovery of known mixing ratios between mixed samples, which were primarily due to inadequate restoration of

inter-sample biological differences in low-quality data or erroneous detection of low-expression genes. Third, the number of DEGs varied widely, and the accuracy metrics for DEG calls demonstrated a broad range across laboratories, even when focusing on protein-coding genes. Differences in data quality, filtering conditions for low-expression genes, differential analysis tools, and the cutoff setting for DEG identification collectively contribute to such variations, which appear to be more significant than the differences in DEG detection

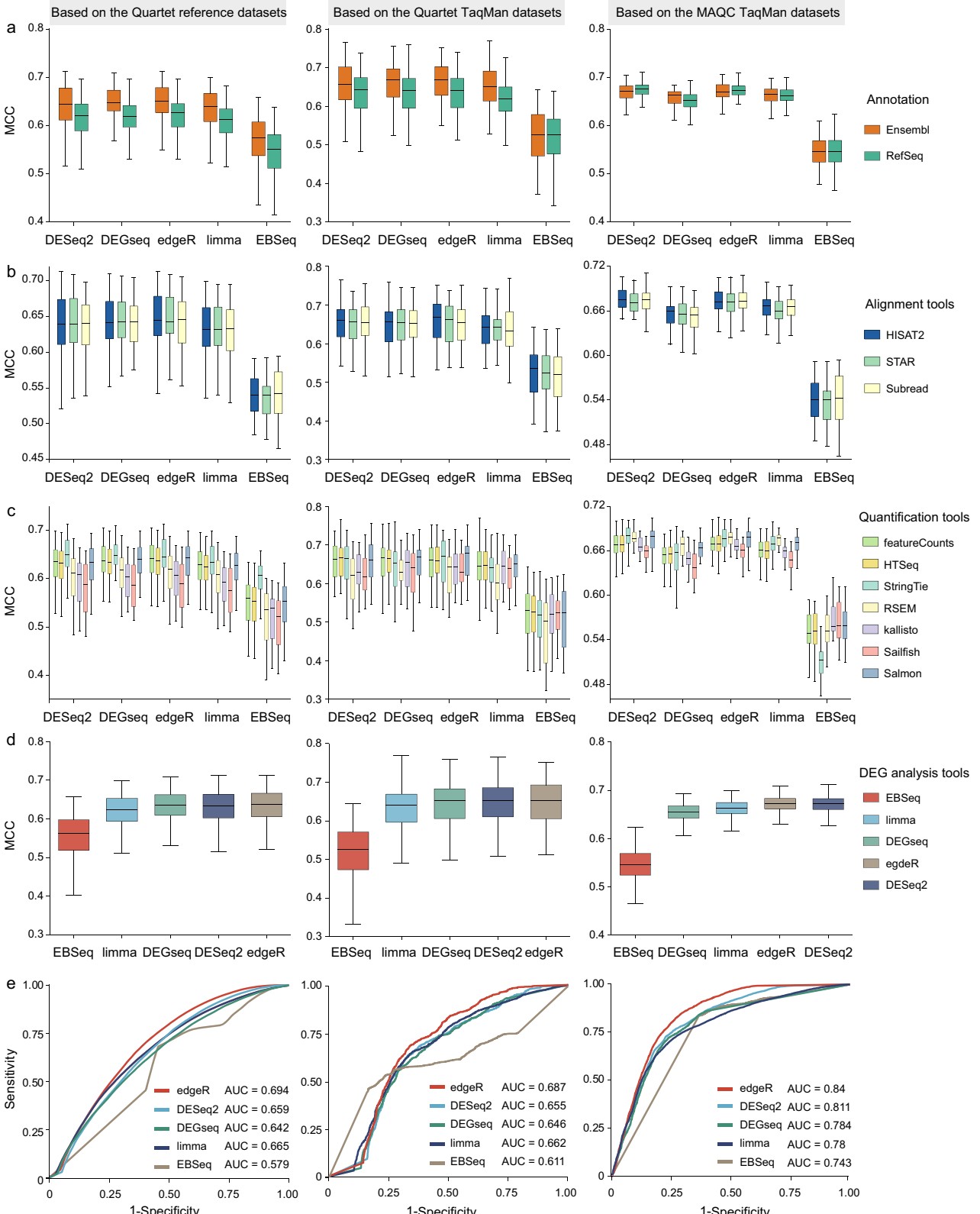

**Fig. 8 | Impact of each bioinformatics step on the accuracy of DEG identification.** Using the benchmark datasets from 13 laboratories, the Matthews correlation coefficients (MCC) for 140 bioinformatics pipelines were measured based on the Quartet reference datasets, the Quartet TaqMan datasets, and the MAQC TaqMan datasets. The impact of each bioinformatics step on the accuracy of DEGs was assessed, including (**a**) different annotations, (**b**) different alignment tools, (**c**) different gene quantification tools, and (**d**) different DEG identification tools. Box plots indicate MCC values for 13 benchmark datasets when different annotations, alignment tools, quantification tools, and DEG identification tools are used, and data are presented as median values (center lines) and the upper and lower quartiles (box limits). **e** The AUC was analyzed for five DEG identification tools using RNA-seq data from lab01, and the AUC values for other high-quality benchmark datasets are displayed in Supplementary Fig. 57. AUC Area Under the receiver operating characteristic Curve.

performance across platforms, sites, and analysis tools previously reported[14,15,21]. Therefore, our results underscore the fact that real-world RNA-seq performance may not fully meet the clinical diagnostic demands, requiring ongoing quality improvement specifically toward subtle differential expression.

The greater inter-laboratory variations in detecting subtle differential expression among the Quartet samples prompted to investigate the sources of variations from diverse RNA-seq workflows, which compensated for previous studies that exclusively focused on the sources of variation under identical protocols and analysis pipelines[16,22]. We observed that the technical factors in experimental and bioinformatics processes contributed to a higher proportion of variations in absolute gene expression levels for the Quartet samples (85.1% and 75.1%) compared to MAQC samples (38.2% and 34%). Relative expression reduced the variation caused by technical factors to below 20%, for both the Quartet and MAQC samples, indicating its significance of eliminating batch effects, as previously shown to facilitate multi-omics data integration[24]. To be specific, in the experimental process, we identified factors affecting absolute expression quantification, including mRNA enrichment methods, strandedness, library kits, read length, the number of exonic reads, and RNA input. In the bioinformatics process, the normalization step is the primary source of variations, followed by quantification, alignment, and annotation. These factors have been individually studied[15,17,41–44], and in contrast, our study revealed the magnitude of their impact in real-world laboratory settings, providing clarity on the priority of technical factors to consider when designing RNA-seq systems.

The experimental design centered on establishing strict quality control to avoid the influence of low-quality experimental execution, as well as choosing experimental protocols for specific research objectives, considering their impact on RNA-seq performance (Supplementary Table 1). First, the quality of experimental execution is a prerequisite. Low-quality execution contributed to poor RNA-seq performance, and its impact is more severe than the choice of different experimental protocols. A comprehensive quality control is necessary to promptly exclude low-quality data for subsequent analysis. Second, the impact of experimental factors should be considered, with the mRNA enrichment methods, strandedness, read length, and sequencing depth identified as key factors based on multiple types of ground truth. (i) The rRNA depletion method exhibited significantly higher accuracy for absolute gene expression measurements. In relative expression levels, our results revealed a generally higher accuracy for the Poly(A) selection method as previously reported[15,17], although differences between laboratories employing the two methods did not reach statistical significance. (ii) Stranded libraries showed enhanced accuracy in gene expression, which appears to be related to more reliable quantification of overlapping genes transcribed from the opposite strands[42]. (iii) Longer read lengths and higher sequencing depths not only benefit gene quantification but also significantly improve junction identification, and the latter is crucial for the application of RNA-seq for isoform and alternative splicing analysis[1]. Our results demonstrated an evident decrease in the accuracy of gene quantification and DEG identification when the sequencing depth was reduced to 10 Mb, which was consistent with the previously proposed minimum requirement for read depth[1]. For other experimental factors, including RNA input, library kits, sequencing platforms, and insert sizes, this study and previous studies have consistently revealed their influence on the RNA-seq performance[15,17,22,38]. However, thorough validation during method development is still necessary to make informed choices regarding these factors. Finally, the experimental designs also need to consider that different experimental methods capture distinct transcriptomic features, which concerns the research objectives. For example, the rRNA depletion method detects more non-coding RNAs and pseudogenes compared to the Poly(A) selection method[15,41]. Stranded and non-stranded libraries also contributed to

the differential expression of pseudogenes and antisense genes[42]. This underscores the importance of considering sample type and quality, such as the extent of RNA degradation[17], as well as research objectives, which may involve non-coding RNAs, pseudogenes, antisense genes, novel transcripts, and alternative splicing events[41,42,45].

The bioinformatics design, centered on the choice of optimal analysis tools, requires equal attention, as the variations from the bioinformatics processes are comparably significant as those from the experimental processes (Supplementary Table 1). This study assessed different normalization methods from the data quality aspects and found that TMM and DESeq significantly improved the quality of expression data, agreeing with conclusions drawn from previous studies[46]. For each step of the differential expression analysis, we found that the performance of any analysis tool is not constant but depends on the other tools used in combination with it. Nevertheless, this study provided the optimal bioinformatics design through an evaluation of arbitrary combinations of analysis tools. First, for gene annotation, choose Ensembl when using exon-level quantification tools such as featureCounts and HTSeq, and choose RefSeq when using transcript-level quantification tools for gene quantification purposes. For differential expression analysis purposes, Ensembl consistently demonstrated higher or similar DEG identification accuracy than RefSeq. Second, for alignment tools, STAR with Ensembl annotation resulted in the highest mapping rate. Similar to previous studies, the three alignment tools generally exhibited similar accuracy in gene quantification or DEG identification[20,47]. Specific cases have also been observed, such as lower accuracy with Subread when using the StringTie for gene quantification. The performance of alignment tools also varied depending on factors including the genomic complexity of different sample types and sequencing data characteristics, which should be considered when selecting alignment tools[20,48]. Specifically, differences in efficiency in aligning reads to ambiguous genes, such as pseudogenes, significantly impact the accuracy of DEGs[49,50]. Third, for gene quantification, choose tools using genome-alignment strategy, particularly featureCounts and HTSeq. Fourth, the threshold for filtering low-expression genes is not fixed but varies with different samples and analysis tools[51]. If benchmark datasets are available, the optimal thresholds could be chosen by balancing sensitivity and specificity after applying a series of thresholds ranging from low to high to filter low-expression genes. Otherwise, it is practical to achieve optimal sensitivity and acceptable specificity of DEG identification by choosing the threshold corresponding to the highest number of DEGs. Finally, edgeR or DESeq2 is preferred for conducting differential gene expression analysis.

This study significantly advances the understanding of the role of reference materials in quality control applications by utilizing the Quartet and MAQC reference materials in parallel (Supplementary Table 1). Overall, the assessment based on these two sets of reference materials demonstrated common patterns in multiple aspects of the transcriptome across laboratories. Notably, each of the two sets of reference materials has significantly enhanced the reliability and distinctiveness of the assessment and exploration of RNA-seq data. On the one hand, the Quartet samples enabled the assessment in subtle differential expression levels and demonstrated advantages in the performance assessment for different laboratories and various analysis pipelines, underscoring the need for a shift in RNA-seq benchmarking toward subtle differential expression levels. First, Quartet samples with large-scale reference datasets enabled a more precise and comprehensive assessment of the RNA-seq performance. The performance metrics exhibited a broader range than those from the MAQC samples in terms of SNR values for assessing data quality, Pearson correlation coefficients for assessing gene expression accuracy, and MCC coefficients for evaluating the accuracy of DEG calls. This implies a higher discriminative ability for discovering performance differences among different batches,

protocols, sites, and analysis tools. Second, Quartet samples allowed for a more sensitive uncovering of technical noise. In the context of subtle biological differences among the Quartet samples, the variations introduced by experimental and bioinformatics factors become more pronounced. Third, the Quartet reference datasets revealed no systemic differences with the RNA-seq data at both absolute and relative expression levels. Methodological differences between RNA-seq and TaqMan RT-qPCR have previously limited gene expression assessments concerning correlation analyzes[14], which are considered to have limitations in representing consistency[52]. The Quartet reference datasets showed a lower RMSE with RNA-seq data compared to the TaqMan datasets, allowing for a direct comparison of the quantitative values of gene expression. On the other hand, the MAQC samples established connections with previous milestone studies, contributing to a deeper understanding of real-world RNA-seq performance based on these traditional RNA reference materials in the community. Moreover, a large-scale TaqMan RT-qPCR dataset for the MAQC samples ensures an unbiased performance assessment, effectively complementing the Quartet reference datasets originated from the Ensembl-HISAT2-StringTie pipeline which may introduce biases especially when assessing diverse RNA-seq analysis pipelines[53].

Limitations should also be noted, considering the influence of the biological complexity of samples on RNA-seq performance. The interpretation of our findings should fully consider the characteristics of the samples. This study utilized two sets of distinctly different reference materials, representing conditions of small (Quartet) and large (MAQC) biological differences, which may not fully reflect more complex biological experiments, such as those involving genetic perturbations or chemical treatments[23,54]. Previous studies have reported that the RNA-seq performance may vary depending on the biological complexity of samples[23]. This concern is particularly relevant in DEG analysis, where each analysis tool uses a distinct data distribution model, inter-sample normalization strategy, and statistical test, tailored to the expected variances in read counts or expression values. Consequently, the performance of the tools could be potentially influenced by specific variances and characteristics of the samples[55]. Despite this, our findings also exhibited similarities with previous studies that used different samples, such as the assessment of different DEG analysis tools[56]. Additionally, the two sets of fundamental RNA reference materials allow for the establishment of a robust foundation of RNA-seq performance across laboratories before embarking on more complex biological experiments. This means that if technical noises cannot be distinguished even in analyzing these reference materials, detecting true biological differences in samples with higher biological complexity or small treatment effects will be even more challenging. We underscored the critical role of comprehensive quality control, meticulous experimental design, and the selection of bioinformatics tools, and these insights obtained from the Quartet and MAQC samples are expected to enhance the quality and interpretation of a variety of RNA-seq experiments.

In summary, this study unveils significant inter-laboratory variations in real-world transcriptome profiling especially when detecting subtle differential expression, with respect to data quality, absolute expression, and differential gene expression. The investigation of the sources of inter-laboratory variations in both experimental and bioinformatic aspects has highlighted key points for the development and optimization of RNA-seq methods. This study provided best practice recommendations regarding the experimental and bioinformatics design and quality control of RNA-seq (Supplementary Table 1). These will aid researchers in accurately identifying subtle changes in disease conditions, accelerating the transition of RNA-seq into a diagnostic tool. Furthermore, these data can also be used to address other aspects of transcriptome profiling, including alternative splicing, gene fusion, RNA editing, and RNA variations.

## Methods

### RNA Reference samples preparation

Four Quartet RNA reference materials (M8, F7, D6, D5) were used[53], and External RNA Control Consortium (ERCC) spike-in Mix 1 and Mix 2 were added to M8 and D6 samples at manufacturer recommended amounts, respectively (4456740, Thermo Fisher Scientific)[13]. Samples T1 and T2 represent mixtures of samples M8 and D6 at the defined ratios of 3:1 and 1:3, respectively, and thus hold 'built-in truths' of sample mixing ratios. Universal Human Reference RNA (740000, Agilent Technologies) and Human Brain Reference RNA (QS0611, Thermo Fisher Scientific) were used, which were labeled as MAQC samples A and B by MAQC Consortium[12]. MAQC B sample was paired with MAQC A sample as a control sample for differential analysis, while Quartet D6 sample served as a control sample for differential analysis of sample M8, F7, D5, T1, and T2. Based on these reference materials, three technical replicates were prepared for 8 RNA samples, resulting in a total of 24 RNA samples (Fig. 1a). All the samples dispensed as 8 μL aliquots into 200 μL thin-wall polypropylene PCR tubes with a concentration of 200 ng/μL and stored at −80 °C.

### RNA-seq workflow

The samples were transported to each laboratory on dry ice, and the ERCC reference sequences and gene annotation files were provided with the names of the 92 ERCC genes modified to 'SPIKEIN' followed by the corresponding identifier. Laboratories conducted the experiments and data analysis following their routine procedures. To accurately capture batch effects within the laboratories, the sample grouping information was provided to the laboratories after they submitted the sample quality results, raw FASTQ files, and quantification results at the gene and transcript levels. Subsequently, laboratories were required to submit differential analysis results at gene and transcript level, and alternative splicing results.

### TaqMan RT-qPCR

Primers and TaqMan probes were designed for 91 genes based on the RNA sequences. Among them, *C1ORF43* was selected as the reference gene for the PCR method. Primers and probes were synthesized by Sangon Biotech, and the sequences are shown in Supplementary Data 6. Before proceeding with the bulk qPCR experiments, we designed two sets of primers and probes for the reference gene and the target gene (*CD180*) to verify the acceptable impact of primer and probe selection on the results. Then the amplification efficiency of the primers and probes was confirmed to meet the requirements by performing gradient dilution experiments with the samples. The results of the *CD180* gene were used for inter-batch quality control for qPCR experiments.

Five μg of each Quartet RNA sample was reverse transcribed using the PrimeScript™ RT reagent Kit (RR037A, TaKaRa) in a 50 μl reaction. This reaction mixture was incubated at 37 °C for 15 min, then for 5 s at 85 °C and finally for termination at 4 °C. cDNA obtained in the previous step was used as template for qPCR. qPCR reactions were run in 96-well plates, the qPCR reactions were carried out using Premix Ex Taq™ (RR390A, TaKaRa) containing 2 μL of cDNA, 0.4 μL of each forward and reverse primers, 0.8 μL of TaqMan probes in a 20 μL final volume reaction. The qPCR was performed on an Applied Biosystems 7500 Real-Time PCR System using the following cycling conditions: 30 s at 95 °C followed by 45 cycles of 5 s at 95 °C and 34 s at 56 °C. Three replicates per sample per gene were conducted for eliminating random variations.

### The ground truth inherent in RNA reference samples

The sample design introduced four types of ground truth, comprising three reference datasets: the Quartet reference datasets and the TaqMan datasets for the Quartet and MAQC samples, and two built-in truths: ERCC spike-in ratios and the mixing ratios in T1 and T2 samples.

**Quartet reference datasets.** The Quartet reference datasets provided relative gene expression and DEG results. They were obtained from the Quartet data portal (https://chinese-quartet.org/#/dashboard)[57], consisting of 31,155 results regarding gene relative expression and DEGs in comparisons of M8/D6, F7/D6, and D5/D6[23]. After intersecting with the gene list of Ensembl release-109 annotation, the remaining 30,976 results (including 5,036 DEGs) comprised 76.8% of protein-coding genes, 13.7% of long non-coding RNAs, 1.1% of small non-coding RNAs, 6.4% of pseudogenes, and 1.9% of immunoglobulin/T-cell receptor gene segments.

**TaqMan datasets for the Quartet samples.** The TaqMan datasets for the Quartet samples included absolute, relative gene expression and DEG results (Supplementary Data 7). RT-qPCR assays were conducted for 90 genes selected from the Quartet reference datasets. Normalized gene expression was quantified using the Delta Ct method, with the *C1ORF43* gene serving as the endogenous control[58]. Fold changes were calculated using the Delta Delta Ct method for the three Quartet sample comparisons (M8/D6, F7/D6, and D5/D6). A gene is classified as a DEG when the student's *t*-test $p < 0.05$ and fold change $\geq 2$ or $\leq 0.5$[15]. The TaqMan datasets and the Quartet reference datasets have a high consistency in terms of fold change (Pearson correlation = 0.93), and DEG identification (88%, 162/184) (Supplementary Fig. 58).

**TaqMan datasets for the MAQC samples.** The TaqMan datasets for the MAQC samples also included absolute, relative gene expression and DEG results. Normalized expression values for 1044 genes were obtained from the Gene Expression Omnibus database (accession number GSE5350), and 830 genes remained after removing genes with undetectable CT values (CT > 35 or CT = 0). Fold changes between the MAQC samples A and B were calculated as the ratios between the mean expression values of the four technical replicates. DEGs were identified using the same method for the Quartet TaqMan datasets.

**The built-in truth of ERCC spike-in ratios.** ERCC spike-in ratios were used to assess both absolute and relative gene expression accuracy. ERCC RNA controls comprised 92 genes in Mix 1 and Mix 2 with known wide-ranging concentrations, which were used for assessing absolute expression levels. ERCC genes could be further divided into four subgroups labeled as a, b, c, and d with respective ratios of 4:1, 1:2, 2:3, and 1:1 between Mix 1 and Mix 2 for each group, which facilitated the assessment of relative expression levels.

**The built-in truth of mixing ratios in samples T1 and T2.** This built-in truth of mixing ratios represents expected ratios in relative expression between T1/D6 or T2/D6 and M8/D6, which was used to assess the accuracy and reproducibility of relative gene expression. Translating the known mixing ratios of 3:1 and 1:3 in RNA samples into gene expression ratios between samples followed methods described in previous studies[14], involving two steps.

First, the expected absolute gene expression of genes in T1 and T2 should be calculable from the absolute expression in M8 and D6 using Eq. (1):

$$\log_2 E_{T1/T2} = \log_2(k1 \times E_{M8} + k2 \times E_{D6}) \tag{1}$$

where $E_{T1/T2}$ represents the expected absolute gene expression in T1 or T2, and $E_{M8}$ and $E_{D6}$ respectively represent the absolute gene expression in M8 and D6. k1 and k2 represent mixing coefficients for calculating gene expression in T1 and T2, respectively.

Second, using gene expression in D6 as the common denominator, the expected relative gene expression in comparisons of T1/D6 or T2/D6 could be calculated from the relative expression observed in the comparison of M8/D6, utilizing Eq. (2):

$$\log_2 E_{\frac{T1}{D6}} = \log_2 \left( k1 + k2 \times 2^{\log_2 E_{\frac{M8}{D6}}} \right)$$
$$\log_2 E_{\frac{T2}{D6}} = \log_2 \left( k2 + k1 \times 2^{\log_2 E_{\frac{M8}{D6}}} \right) \tag{2}$$

where $E_{\frac{T1}{D6}}$ and $E_{\frac{T2}{D6}}$ represents the expected relative gene expression in comparisons of T1/D6 or T2/D6, and $E_{\frac{M8}{D6}}$ represents the relative gene expression in the comparison of M8/D6. Due to potentially different mRNA proportions in different samples, as stated by ref. 61, the mixing coefficients for M8 and D6 may not adhere strictly to the theoretical values of 1/4 and 3/4. A correction value, z, was introduced to correct the mixing coefficients: $k1 = z/(z + 3)$ and $k2 = 3z/(3z + 1)$, which was determined using RT-qPCR assays. In brief, ten genes with a broad range of expression levels were tested using RT-qPCR, and the corresponding *z*-values for each gene were calculated using Eq. (1) based on gene expression results from different samples. The obtained average z values were $0.974 \pm 0.06$ for T1 and $0.949 \pm 0.09$ for T2. To validate the *z*-values, RNA-seq data from the top ten laboratories with higher accuracy for recovering ERCC spike-in ratios were used, resulting in *z*-values of $0.965 \pm 0.024$ and $0.941 \pm 0.026$ for T1 and T2, respectively. Finally, the *z* values from RT-qPCR assays were used.

## RNA-seq performance metrics

The PCA-based SNR was used to assess the data quality at the gene expression level, and the calculation method of PCA-based SNR as shown in the previous study[23]. A gene was included for PCA analysis and subsequent SNR calculation if more than one read was mapped to it in any one of the samples included. SNR calculated using the absolute gene expression profiles from 24 samples including the MAQC samples (SNR24) and 18 samples excluding the MAQC samples (SNR18), represented the ability to distinguish large and subtle biological differences from technical noises among technical replicates, respectively. Additionally, SNR17 was calculated from any 17 out of the 18 samples (12 Quartet and 6 mixed samples). If the removal of a particular sample results in a significant increase in SNR17 compared to SNR18 (with a difference greater than 6), the quality of the RNA-seq data of the excluded sample is considered low and the sample is treated as an outlier. Such random occurrences of problematic library preparation or sequencing in individual samples are referred to as random failures.

The Pearson correlation coefficient was used to evaluate the accuracy of gene expression. This involved examining the correlation between $\log_2$FPKM and ERCC concentrations as well as TaqMan gene quantification results, for assessing absolute gene expression, and examining the correlation between sample-pair relative expression and gene fold change in the TaqMan datasets and the Quartet reference datasets, for assessing relative gene expression. The Root Mean Square Error (RMSE) was also used to measure the difference between RNA-seq data and Quartet reference datasets and TaqMan results.

The built-in truth of mixing ratios in T1 and T2 was employed to assess the accuracy of RNA-seq in capturing known gene expression ratios between samples and the reproducibility of gene quantification within the laboratory[14,59,60]. A nonlinear robust fit (nlrob) was performed for relative gene expression in comparisons of T1/D6 or T2/D6 and M8/D6 for each laboratory, and the fitting curves were compared to the expected curves defined by Eq. (2) (Fig. 1b). The closeness between the laboratory data and the expected curve indicates accuracy, while the clustering of laboratory data represents reproducibility. Additionally, the RMSE between the observed and the expected relative expression from Eq. (2) was calculated for comparisons of T1/D6 or T2/D6.

The MCC was used to evaluate the accuracy of DEG identification for a given pair of samples based on the Quartet reference datasets and TaqMan results. The true positives, true negatives, false positives, and false negatives were judged as shown in Fig. 1b. Then MCC was calculated using Eq. (3).

$$MCC = \frac{TP \times TN - FP \times FN}{\sqrt{(TP+FP)(TP+FN)(TN+FP)(TN+FN)}} \qquad (3)$$

### Fixed data analysis pipeline

RNA-seq data from all laboratories were analyzed using uniform analysis pipelines to excluded variations from different bioinformatics pipelines. Preliminary processing of raw reads was performed using fastp (v.0.23.2) to remove adapter sequences[61]. Sequences were aligned to the GRCh38 genome assembly[62] using STAR (v.2.7.10b)[63] with Ensembl annotation. Gene quantification was conducted using StringTie (v2.2.1)[64]. Additionally, we also utilized RefSeq gene annotation and Salmon (v.1.10.1)[65] for gene quantification to strengthen the validity of conclusions. The Ensembl release-109 annotation[66] and the recent RefSeq annotation (2023-03-21)[67] was obtained. STAR was run with the "--twopassMode Basic" option, and genes were quantified using StringTie -e function. Salmon was run in the mapping-based mode, using optional parameters: --gcBias, --seqBias, and --posBias. The $\log_2$ transformation was then performed based on Fragments Per Kilobase of transcript per Million mapped reads (FPKM) values. To avoid infinite values, a value of 0.01 was added to the FPKM value of each gene before $\log_2$ transformation.

Quality control analysis of sequencing data at pre-alignment, post-alignment, and sample level was conducted using FastQC (v.0.12.1), fastp (v.0.23.2)[61], Qualimap (v.2.2.2)[68], MultiQC (v.1.8)[69], and NGSCheckMate (v.1.0.1)[70].

### Relative expression calculation

Relative expression data were obtained within each laboratory on a gene-by-gene basis. Specifically, relative expressions were calculated based on $\log_2$FPKM values. For each gene, the mean of expression profiles of replicates of reference sample(s) (for example, D6) was first calculated and then were subtracted from the $\log_2$FPKM values of that gene in other samples.

### Filtering of low-quality data

To exclusively focus on the impact of methodological differences in experimental processes on RNA-seq performance and to mitigate the effects of low-quality library or sequencing and sample cross-contamination due to experimental execution, we employed pre-alignment, post-alignment, and sample-level quality metrics listed in Supplementary Data 3, based on FASTQ, BAM, and expression profiles, respectively, to filter out low-quality RNA-seq data from subsequent analysis. The main pre-alignment quality criteria included: (i) Q30 > 85%, (ii) the number of paired reads >20 Mb, and (iii) the duplication rate <30%. The post-alignment quality criteria included: (i) the total mapping rate >90%, (ii) 5'-3' bias ranging from 0.8 to 1.2, and (iii) the proportion of reads mapped to the intergenic region <10%. The sample-level quality criteria included: (i) the SNR value based on the MAQC samples > 20, (ii) the SNR value based on the Quartet samples > 12, (iii) the difference between SNR17 and SNR18 < 6, (iv) the proportion of reads mapped to ERCC sequences <0.005%, and (v) passing the SNP-based sample-identity check.

### Principal variance component analysis

Principal variance component analysis (PVCA) was performed to estimate the sources of variation in absolute or relative expression profiles from the experimental and bioinformatics processes using the pvca package (v.1.40.0)[71]. In brief, after filtering out low-quality data, the expression data for all samples from the remaining RNA-seq data were used as the input for PVCA. PVCA involved four basic steps: (i) perform PCA and select the first few principal components to retain the majority of the variations in the gene expression data, (ii) fit a mixed linear model separately to each principal component with all factors of interest as random effects, (iii) estimate the variance components of each factor through restricted maximum likelihood and average the estimated variance components with their corresponding eigenvalues as weights for each factor, and (iv) standardize the weighted average variance components by dividing by their sum, so that the proportion of variance components reflects the magnitude of sources of variation[72]. We examined a total of 15 experimental factors, including RNA input, mRNA enrichment method, strandedness, library kit, library concentration, read length, sequencing platform, flowcell, lane, base quality, GC content, duplication rate, the number of reads in exonic region, insert size, and 5' to 3' bias, and four bioinformatics factors, including annotation type, alignment tool, quantification tool, and normalization method. Sample grouping also served as a factor in PVCA, allowing for a comparison between biological differences among samples and variations introduced by various factors.

### Benchmark of bioinformatics pipelines

**Benchmark datasets.** High-quality data from 13 laboratories were further selected as the benchmark dataset. In brief, from all RNA-seq data that passed all QC metrics, data were further selected based on SNR values exceeding 20 (SNR based on the Quartet and mixed samples).

**Gene annotation.** Two human gene annotations were included as the guiding reference for alignment and quantification tasks in this study, including the Ensembl release-109 annotation and the recent RefSeq annotation (2023-03-21). All these annotations were generated based on the human reference genome GRCh38. The gene annotation files were used in conjunction with reference genome or transcriptome files from the corresponding database.

**RNA-seq analysis tools.** We included all analysis tools used by the 45 laboratories to reflect real-world scenarios. Additionally, we also included three popular tools, including Subread for alignment[73], and two alignment-free quantification tools (Sailfish and Kallisto)[74,75]. The details of RNA-seq analysis tools, versions, and the command line are listed in Supplementary Data 8.

Three alignment tools, STAR (v.2.7.10b)[63], HISAT2 (v.2.2.1)[76], and Subread (v.2.0.3)[73], were included. Eight gene or transcript quantification tools were included, consisting of genome-alignment quantification tools like featureCounts (v.2.0.3)[77], HTSeq (v.2.0.2)[78], StringTie (v.2.2.1)[64], and STAR, one transcriptome-alignment quantification tool, RSEM (v.1.3.1)[79] with Bowtie (v.1.3.1) for alignment[80], as well as three alignment-free quantification tools, including kallisto (v.0.48.0)[74], Salmon (v.1.10.1)[65], and Sailfish (v.0.9.0)[75]. According to the different quantification principles, these quantification tools can be divided into exon-level tools (featureCounts, HTSeq, and STAR) and transcript-level tools (Kallisto, Salmon, Sailfish, RSEM, and StringTie), which quantify gene expression by counting or estimating reads mapped to exons or transcripts, respectively. Five differential expression analysis tools, edgeR (v.3.42.4)[81], limma (v.3.56.2)[82], DESeq2 (v.1.40.2)[83], DEG-seq (v.1.54.0)[84], and EBSeq (v.1.40.0)[85], were included and compared.

Each tool was run with default parameters, and certain specific options were also selected considering their potential to improve alignment or gene quantification as stated in the Reference Manual. The specific parameter options are as follows:

- STAR was run with the "--twopassMode Basic" option for more accurate alignment, "--quantMode GeneCounts" option to obtain read counts, and "--outSAMunmapped Within" option to retain unmapped reads.

- Subread was run with –multimapping and -B to retain multi-mapped reads.
- featureCounts was run with optional arguments: -B to only count fragments that have both ends successfully aligned and -C to exclude the chimeric fragments.
- StringTie was run with parameter -e specifying to only estimate the expression levels of the reference transcripts.
- Salmon was run in the mapping-based mode, using optional parameters: --gcBias, –seqBias, and --posBias to correct sequence-specific biases, GC biases, and 5′ or 3′ positional bias.
- Kallisto was run using –bias to correct biases.

The mapping information of each mapping tool was evaluated using Samtools flagstat and stats function[86]. The number of mismatches was detected using the NM tag. The junctions were extracted from BAM files using 'junction_annotation.py' in RSeQC package (v.5.0.1)[87]. Transcript-level reads counts from Sailfish and Kallisto were transformed to gene-level counts using tximport package (v.1.28.0)[88]. Post alignment duplication rates in the BAM files were calculated using Picard (v.3.1.1) CollectInsertSizeMetrics function[89]. The outputs of all quantification tools were converted to FPKM for comparison with the ground truth or between tools.

**Normalization methods.** We consider six normalization methods: total counts (TC), fragments per kilobase million (FPKM), transcripts per million (TPM), trimmed mean of M values (TMM), upper quartile (UQ) normalizations, and normalization method used by DESeq2 (v.1.40.2). TC also known as CPM (Counts Per Million), corrects for library size (expressed in million counts) so that each count is expressed as a proportion of the total number reads in the sample. FPKM and TPM are similar methods that correct for both library size and gene length, but TPM divides counts by gene length first and then by total number of transcripts in the sample, resulting in each normalized sample having the same number of total counts. The TMM approach is to choose a sample as a reference sample and the others as test samples. Under the hypothesis that the majority of genes are not DEGs, a scaling factor is calculated to adjust for each test sample after excluding highly expressed genes and genes with high log ratios between the test and the reference sample[55]. The TMM normalization method is implemented in the edgeR package (v.3.42.4) by means of the calcNormFactors function[81]. UQ normalization first removes all zero-count genes and calculates a scaling factor for each sample to match the 75% quantile of the counts in all the samples[90]. UQ normalization was performed using the uqua function in package NOISeq (v.2.44.0)[91]. DESeq normalization method is also based on the hypothesis that most genes are not DEGs. The scaling factor for a given sample is computed as the median of the ratio of the read count and the geometric mean across all samples for each gene[92]. Raw counts were normalized using the estimateSizeFactors() and sizeFactors() functions in the DESeq package (v.1.40.2).

**Filtering conditions for low-expression genes.** Data from four different laboratories, with varying sequencing depth levels ranging from low to high, were utilized to validate the optimal filtering methods and thresholds. We calculate the maximum (max), median, and sum of raw read counts and CPM for each gene from the replicated samples, resulting in six different combined filtering methods. Using each filtering method, we applied a series of thresholds, ranging from low to high, to filter out up to 70% of lowly expressed genes. To facilitate comparison of different filtering methods, the real threshold values were transformed into percentile-based thresholds. We next examined the performance of different differential analysis tools after applying different filtering conditions. The true positive rate (TPR) measures the proportion of DEGs that are accurately detected as positive by the differential analysis tools. Precision measures the proportion of the detected DEGs made are correct (true positives).

## Statistical analysis

All statistical analyses were performed using R statistical packages (v.4.3.0) and python (v.3.10.10). PCA was conducted with the uni-variance scaling, using the prcomp (v.3.6.2) function. Principal variance component analysis (PVCA) was performed by pvca package (v.1.40.0)[72] to quantifies the proportion of variance explained by each influencing factor. Cutadapt (v.4.8)[93] was used to shorten read lengths into 100 bp, 75 bp, 50 bp, and 25 bp. The seqtk (v.1.4)[94] was employed for down-sampling RNA-seq data to various sequencing depths. The distribution of duplicated reads across expression levels were examined using dupRadar (v.3.18)[31].

## Reporting summary

Further information on research design is available in the Nature Portfolio Reporting Summary linked to this article.

## Data availability

The data supporting the findings of this study are available from the corresponding authors upon request. The high-quality raw sequence data reported in this paper have been deposited in the Genome Sequence Archive (Genomics, Proteomics & Bioinformatics 2021) in National Genomics Data Center (Nucleic Acids Res 2022), China National Center for Bioinformation / Beijing Institute of Genomics, Chinese Academy of Sciences under accession code of GSA-Human: HRA005937[95].

## Code availability

The 28 gene quantification pipelines were integrated using Snakemake and are available on GitHub at https://github.com/lyaqing/snakemake_rnaseq.git[96]. Custom scripts for the assessment of data quality and accuracy of gene expression and DEGs are available on GitHub at https://github.com/wangduo-ux/Asssessment-of-RNA-seq-performance.git[97].

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

## Acknowledgements

This study was supported in part by the National Key R&D Project of China (grant 2023YFC3402503 to J.L., 2023YFC3402501 to L.S., and 2018YFE0201600 to L.S.). We thank the 45 laboratories for performing RNA-seq and for returning the raw data and analysis results on time. Some of the illustrations in the Fig. 1 were created with BioRender.com.

## Author contributions

R.Z., J.L., Y.T.Z., and L.S. conceived the study, provided overall supervision, and revised the manuscript. Y.Z., L.S., W.H., and Y.Y. provided the RNA reference materials and reagents. D.W. performed the experiments, conducted data analysis, generated the figures, and wrote the original manuscript. Y.L. developed the Snakemake data analysis pipelines, provided helpful discussions, reviewed, and revised the manuscript. Y.F.Z. managed the datasets. Q.C., Y.H., C.L., Y.Y., Z.Q.L., Z.Y.L., and J.Z. participated in the study design, provided constructive discussions and invaluable advice for this study. All authors reviewed and approved the manuscripts.

## Competing interests

The authors declare no competing interests.
