## [Peer Review File · Nature Communications]

A Real-World Multi-Center RNA-seq Benchmarking Study Using the Quartet and MAQC Reference MaterialsREVIEWER COMMENTS

Reviewer #1 (Remarks to the Author):

This study makes a significant contribution by identifying and recommending best practices for RNA-seq data analysis, particularly for detecting subtle differences in differentially expressed genes (DEGs) that have crucial clinical implications. Utilizing the established MAQC datasets along with a unique dataset from Quartet for benchmarking, the research aims to pinpoint factors contributing to inter-laboratory variation. The authors have meticulously designed their evaluation strategies, assessing data quality, accuracy, and reproducibility of gene expression analyses, including differential gene expression. They have compared more than 40 experimental processes and 140 differential analysis pipelines against multiple 'ground truths'. The evaluation is thorough, offering various considerations for effective RNA-seq data analysis and proposing optimal bioinformatics pipelines. The findings are expected to be highly beneficial to the scientific community.

Considering the worth of this investigation, there are a number of points that I believe will benefit from further clarification. I've outlined below several specific questions regarding the experimental details and evaluation methods (line number corresponds to the merged pdf file):

L145: "Basic quality control for raw reads and read alignment" Is there any criterion for filtering low quality samples? such as those with low base quality, low mapping rate, high duplication rates or high contamination. Did you perform any sample level quality control? such as clustering and sample identity check (e.g., SNP based).

Given that multiple QC have been done, it would be great to have a table to summarize each sample with PASS or Fail for each QC.

L177: The result shows that some laboratories/samples got very high duplication rates. Did you check the source of it, e.g., by looking at the distribution of duplicated reads of those samples? Did you filter those samples as outliers?

L196: In Supplementary Figure 12, a subset of Quartet samples subjected to poly(A) selection exhibited a comparably low percentage of exonic fragments. Have you investigated potential reasons for this trend within this subset of samples?

L203: Have the RNA samples or libraries identified with potential contamination been excluded from subsequent analysis? Additionally, could you provide information on the number of samples that failed quality control and how many were used in the downstream analysis?

L227: How many samples were classified as poor quality based on signal-to-noise ratio (SNR)? Are these samples the outliers depicted in Figure 2a?

L370: "Best practices for experimental designs", you have shown (from supplementary figure 29, 30 and Figure 5) that not a specific factor was identified that can specifically contribute to inter-laboratory variations. To better echo the section title, I think you should make it clearer in the main context to readers about your conclusion.

L390: Sequencing was performed with only 100 and 150 base pairs of read length. The data does not conclusively demonstrate that 100bp constitutes best practice. Notably, 50 bp read lengths have been utilized in many projects, including TCGA, indicating the potential value of investigating shorter reads. Additionally, RNA fragment length represents a critical design factor worth examining in conjunction with read length to comprehensively assess performance.

L391: In Figure 5, how is exonic coverage defined? It sounds like a quality control (QC) outcome rather than an aspect of the experimental design.

L930: Given that ground truth expression values and DEGs are central to the paper's evaluations, it is essential to include a dedicated section in the methods part to describe in detail how the ground truths were generated.

L1047: Bioinformatics pipelines should be both reproducible and publicly available. Workflow languages, such as CWL, WDL, Nextflow, or Snakemake, are recommended to construct the pipelines, as they can help simplify and standardize workflows, thus facilitating easier sharing and reproduction. Furthermore, In addition to the focus on data availability (L1121), I would like to suggest that the authors make the reproducible code/workflows publicly accessible alongside the manuscript.

Reviewer #2 (Remarks to the Author):

Toward Best Practice in Identifying Subtle Differential Expression with RNA-seq: A Real-World Multi-Center Benchmarking Study Using Quartet and MAQC Reference Materials

This study thoroughly evaluated real-world RNA-seq by analysing Quartet and MAQC samples, particularly when detecting subtle differential expression levels. They also investigated sources of variation in gene expression data at experimental and bioinformatics aspects. In several studies, it has been already discovered that different sample processing protocols, platforms, and analysis pipelines used across laboratories compromise the accuracy and reproducibility of RNA-seq. In this study, they are also trying to find the sources of inter-laboratory variations. They generated RNA-seq data across 45 independent laboratories using Quartet RNA samples with spike-ins of ERCC controls, and MAQC RNA samples, where each laboratory used their in-house experimental protocol and analysis pipeline. The insights they provide in this paper are interesting and informative however certain aspects mentioned are vague and require further elaboration. Specifically, the method section lacks explanations and details about the evaluation of variations and reproducibility. For example, in evaluating the variation introduced by each bioinformatics tool, the methodology to mask the effect of other steps of analysis and their contribution in causing variations needs to be explained.

Please address the following comments:

1- How do you define subtle differential expressions?

2- How built-in truth is known based on mixed-ratios of T1 and T2 (lines 231 to 233)? I am not convinced how fold change can be used for T1 and T2 in measuring the accuracy and reproducibility (Supplementary notes section 2.1).

3- What does SNR17 and SNR18 mean and how the difference between SNR17 and SNR18 are related to the identification of random failures in technical replicates? What do you mean by random failures (lines 227 to 230)?

4- What is DEG classification here, does it mean the same genes are reported as DEGs? The number of DEGs says nothing about the consistency based on my understanding (lines 280 to 282).

5- It would make reading the paper easier if you just mentioned the 17 factors, rather than referring to the supplementary table (lines 319 and 320).

6- How the percentage of variation is calculated in line 324 and how do you know they were specifically coming from these factors? I did not find any explanation about this part in the method section.

Also what are the variations based on, is it DEG or absolute differences? The same applies to the percentages of variations calculated for bioinformatics tools/steps. How do you know they are exactly coming from a specific tool? How do you exclude the influence of other factors?

7- in line 445 it is stated that "the performance ranking of all quantification pipelines supported the superior performance of opting for Ensembl gene annotation, any alignment tool, and either featureCounts or HTSeq for quantification". Was RMSE the lowest over all reference datasets or did you take the average over all three to make such a conclusion? Is RMSE calculated based on

absolute expression levels? What are the colors in Figure 38 in supplementary material? Some quantification tools return TPM and some return gene counts, did you convert them all into one gene count before comparing?

8- RSEM uses either Bowtie, Bowtie2, or Star for alignment. Which one did you use? Please clarify that in your figures and methods section.

9- It was concluded that "the impact of different alignment tools on differential expression measurement is minimal". Please add more details on this. Alignment is one of the important steps in the quantification of RNA-seq and most of the quantification tools rely on the input alignment for the quantification. Different alignment tools handle ambiguity in different ways and therefore that will impact the results of the quantification tools. I am not convinced that alignment tools can be picked arbitrarily in quantification pipelines.

10- How table 1 is related to this statement "The experimental design is generally considered to be centered around addressing the biological questions of interest (Table 1) lines 555 and 556?"

11- In line 591 it is mentioned that a threshold chosen based on the maximum number of DEGs is practical. How this threshold is chosen?

12- How correlation coefficients for assessing gene expression accuracy is calculated in line 607?

13- in the supplementary material and notes section 2.4 it was mentioned that only the genes that have at least one read in all replicates are considered. What alignment tool did you use for this part? what is the logic/ justification behind ignoring the genes that don't have a read mapped to that gene in all replicates?

14- Figure 2a- labs are not labeled.

15- In Supplementary Figure 17 gene types should be labeled

In a few other plots, the labs are not labeled. Please double-check.

Typo: TPE -> TPR

Supplementary Figure 45. Comparison of the optimal threshold values determined by maximum number of DEGs and highest TPR. For all RNA-seq from laboratories, the threshold values corresponding to the highest TPE and the maximum number of DEGs were calculated and compared for Quartet (up) and MAQC (down) samples. TPR, true positive rate.

Typo: accuracy -> accurate

Supplementary Figure 20. Relative expression measurements were more accuracy than absolute expression measurements. Based on TaqMan datasets for both Quartet and MAQC samples, relative expression consistently exhibited Pearson higher correlation coefficients when compared to absolute expression.

Reviewer #3 (Remarks to the Author):

Wang et al. undertook a comprehensive analysis of over 1000 RNA-seq datasets from 45 independent laboratories to explore the variability and noise inherent in transcriptome analysis via RNA-seq. Their study aimed to establish best practices for experimental design and analysis pipelines, mainly focusing on identifying differentially expressed genes and quantifying the absolute and relative expression levels of transcripts. Unlike the MAQC dataset, which is characterized by significant gene expression variances among samples, Wang et al. additionally utilized the Quartet dataset, notable for its minimal gene expression differences. This addition allows their methodological evaluations to mirror typical biological research conditions better. By leveraging the Quartet and MAQC datasets, the authors thoroughly evaluated various quantitative indicators across multiple methods and under various conditions. Their work provides essential insights for designing and analyzing future transcriptome studies, reflecting a more realistic and applicable approach to the challenges of RNA-seq analysis.

Despite the significant insights offered by the analyses, the study scheme and the composition of datasets have inherent limitations that necessitate careful interpretation of the findings. These limitations must be clearly communicated to ensure the readers can accurately grasp the insights provided before the study's publication. In particular, the following major points will require careful consideration.

The manuscript extensively uses SNR calculations through PCA as a primary quantitative metric for library quality assessment. However, an issue arises, as depicted in Figure 2A, where SNR values are notably higher for low-quality outliers, considered random failures. Although the utilization of PCA-based SNR is discussed in the original paper on the Quartet dataset, I believe that relying on PCA-based SNR as a universal quality metric is still premature without accounting for additional factors that could affect its monotonicity with actual library quality.

- The study leverages the MAQC and Quartet datasets throughout the manuscript, complicating these findings' generalization. The MAQC dataset demonstrates substantial between-sample variation, whereas the Quartet dataset exhibits minimal variation. Applying the study's conclusions to conventional RNA-seq analyses may be limited, especially those involving genetic perturbations or chemical treatments. This concern is particularly relevant to conclusions drawn about DEG analysis because the selection of intersample normalization strategies and statistical testing methods is influenced by the expected variance and specific characteristics of the samples under comparison.

- Figure 2a shows that a concerning number of participating laboratories generated low-quality data. In general research situations, low-quality data are rarer than this, thanks to outsourcing robotics-enabled sequencing centers or professional sequencing cores, and the service accompanies routine quality checking. However, the set of low-quality data persists throughout the analysis and appears in multiple analyses that could be more realistic with pre-filtering low-quality data. Particularly for Quartet, where the biological differences in the data are inherently minimal, data quality becomes even more critical. A more constructive approach would be to demonstrate the variability of library quality and related factors first, then proceed to the rest of the analyses with low-quality data removed.

- Regarding the presentation of analyses and results, while the authors' rich display of relatively unprocessed forms of evaluation results, many figures do not convey the point related to the context referred to in the main text. Instead, they look more like figures directly out of the quality check report. I think the publication should become more accessible to readers if, at least, the main figures clearly show the points of the context.

To further enhance the clarity and robustness of the study, the following minor points would need to be addressed:

- Line 127: Please clarify whether the technical replicates conducted by independent laboratories were performed in parallel or serially. This distinction is important because serial replication can primarily reflect between-lab variance, whereas parallel replication encloses both between-lab variances and batch effects together.

- Line 147: Read counts serve as a more direct measure than total base output in RNA-seq analysis and thus offer a more explicit representation of the data.

- Line 161: This explanation must be clarified, possibly stemming from misinterpreting the Illumina FAQ. The FAQ refers to mismatches resulting from random hexamer priming rather than decreases in quality scores. Such mismatches are misincorporated bases in the DNA templates, whereas the quality score is a prediction made by the basecaller, subject to refinement only through post-analysis quality score calibration. Since this drop in quality scores also occurs in DNA sequencing, reverse transcription theories cannot adequately explain it. The actual cause is likely insufficient signal correction for factors such as phasing and crosstalk during signal intensity quantification and basecalling.

- Line 163: Several RNA-seq library preparation kits, including the Illumina TruSeq kit, sequence the reverse strand first. Therefore, the analysis of quality scores should be separately conducted for kits that sequence in forward-reverse and reverse-forward orders to account for this variation accurately.

- Line 172: The analysis should consider the potential strong influence of sequencing depth on duplication rates. Furthermore, since duplicated reads from noncoding RNAs introduce difficulties in accurate detection, a more precise duplication rate estimation could be achieved by restricting the analysis to reads mapped to coding genes.

- Line 203: While the text identifies contamination during library preparation as a source of error, it should also acknowledge potential contamination within the sequencer. This includes contamination of barcodes between physically adjacent reads on the flow cell and carry-over contamination from previous samples through the sequencer's fluidic channels.

- Line 218: Please consider the significant impact of read depth on SNR values. Incorporating read depth into the analysis is crucial for accurately assessing SNR values.
- Line 299: It should be clarified why the thresholds and cutoffs are different. Please specify whether these differences arise from decisions made by individual labs, variations in the handling of each library during analysis, or are inherent to the unique characteristics of each library, such as read depth.
- Line 307: To strengthen the conclusions' validity, verifying whether analogous outcomes are observed when employing a distinct analysis pipeline is desirable. This cross-checking would ensure that the findings are independent of the specificities of the initially used pipeline.
- Line 383–397: This discussion implies a high SNR as an absolute superiority in performance. However, when comparing methodologies like poly(A) selection and rRNA depletion, the direct comparison of SNR might not be equitable due to inherent differences, such as the inclusion of ncRNAs or the effect of poly(A) tail length on RNA quantification. A more nuanced examination of these factors would be preferred.
- Line 405–443: The comparisons among different alignment and quantification software are as much about the chosen algorithmic options as the software choices. The text should detail the rationale behind the selection of specific options for each software to tailor the benchmarking to the study's objectives, ensuring readers understand the impact of these choices.
- Line 422: In junction discovery or isoform detection, read length is as significant as sequencing depth. This needs to be discussed.
- Line 482–501: The section implies that more detected DEGs indicate superior performance. However, considering the biological context of the Quartet dataset, the identification of over 5000 DEGs might be questionable. It's crucial to assess the biological relevance of these DEGs, considering factors like effect size and independent validations, to ensure the DEG analysis accurately reflects the biological phenomena under investigation.

Title "Toward Best Practices in Identifying Subtle Differential Expression with RNA-seq: A Real-World Multi-Center Benchmarking Study Using the Quartet and MAQC Reference Materials"

Manuscript ID: NCOMMS-23-61052-T

Point-by-Point Response to Reviewers

Reviewer #1 (Remarks to the Author):

This study makes a significant contribution by identifying and recommending best practices for RNA-seq data analysis, particularly for detecting subtle differences in differentially expressed genes (DEGs) that have crucial clinical implications. Utilizing the established MAQC datasets along with a unique dataset from Quartet for benchmarking, the research aims to pinpoint factors contributing to inter-laboratory variation. The authors have meticulously designed their evaluation strategies, assessing data quality, accuracy, and reproducibility of gene expression analyses, including differential gene expression. They have compared more than 40 experimental processes and 140 differential analysis pipelines against multiple 'ground truths'. The evaluation is thorough, offering various considerations for effective RNA-seq data analysis and proposing optimal bioinformatics pipelines. The findings are expected to be highly beneficial to the scientific community.

Reply: We are very grateful to the reviewer for recognizing our work's "significant contribution." We fully agree with the reviewer that our findings are indeed expected to be "highly beneficial" to the scientific community.

Considering the worth of this investigation, there are a number of points that I believe will benefit from further clarification. I've outlined below several specific questions regarding the experimental details and evaluation methods (line number corresponds to the merged pdf file):

Reply: We greatly appreciate your constructive comments and suggestions for improving our manuscript.

1. L145: "Basic quality control for raw reads and read alignment" Is there any criterion for filtering low quality samples? such as those with low base quality, low mapping rate, high duplication rates or high contamination. Did you perform any sample level quality control? such as clustering and sample identity check (e.g., SNP based).

Given that multiple QC have been done, it would be great to have a table to summarize each sample with PASS or Fail for each QC.

Reply: Thank you for your constructive questions and suggestions.

1)“Basic quality control for raw reads and read alignment” Is there any criterion for filtering low quality samples? such as those with low base quality, low mapping rate, high duplication rates or high contamination.

Following your suggestion, we have added criteria for filtering low-quality data using basic quality control parameters. Using pre-alignment (such as Q30, the number of total sequencing reads, and duplication rate), post-alignment (such as mapping rate, 5'-3' bias, and percentage of reads in intergenic regions), and sample-level (such as SNR and reads mapped to ERCC genes) quality metrics, RNA-seq data from 16 laboratories were flagged as low quality and excluded from subsequent analysis. Accordingly, we have added the descriptions for filtering low-quality data in the Results and the Materials and Methods sections as follows:

“Using the above multiple metrics, including pre-alignment, post-alignment, and sample-level quality metrics, RNA-seq data from 26 laboratories passed all criteria and were used for subsequent analysis, whereas the other data from remaining 16 laboratories were flagged as low quality and excluded from subsequent analysis to mitigate the impacts of poorer library or sequencing quality, as well as sample cross-contamination (**Supplementary Table 3**) (**Fig. 3**).” (Results, lines 361–366)

“Filtering of low-quality data

To exclusively focus on the impact of methodological differences in experimental processes on RNA-seq performance and to mitigate the effects of low-quality library or sequencing and sample cross-contamination due to experimental execution, we employed pre-alignment, post-alignment, and sample-level quality metrics listed in **Supplementary Table 3**, based on FASTQ, BAM, and expression profiles, respectively, to filter out low-quality RNA-seq data from subsequent analysis. The main pre-alignment quality criteria included: (i) Q30 > 85%, (ii) the number of paired reads > 20 Mb, and (iii) the duplication rate < 30%. The post-alignment quality criteria included: (i) the total mapping rate > 90%, (ii) 5'-3' bias ranging from 0.8 to 1.2, and (iii) the proportion of reads mapped to the intergenic region < 10%. The sample-level quality criteria included: (i) the SNR value based on the MAQC samples > 20, (ii) the SNR

value based on the Quartet samples > 12 , (iii) the difference between SNR17 and SNR18 < 6 , (iv) the proportion of reads mapped to ERCC sequences $< 0.005\%$, and (v) passing the SNP-based sample-identity check.” (Materials and Methods, line 1334–1348)

2) Did you perform any sample level quality control? such as clustering and sample identity check (e.g., SNP based).

Yes, we did perform sample-level quality control, including PCA-based SNR analysis reflecting inter-sample group and inter-replicate clustering in terms of gene expression profiles, and cross-contamination assessments based on the number of reads mapped to the ERCC genes. According to your suggestion, we have added sample-identity check (SNP based) results using NGSCheckMate (v.1.0.1). All laboratories exhibited higher Pearson correlations in variant allele frequencies of SNP between any two out of three technical replicates, compared to between-sample group comparisons. These results indicated the absence of sample swaps or mislabeling. We added detailed descriptions in the Results section and added **Supplementary Figure 26**, as follows:

“Additionally, we performed the sample-level quality control to identify any problematic samples.” (Results, lines 335–336)

“Second, based on the fact that the single nucleotide polymorphisms (SNPs) among technical replicates are theoretically identical, we calculated the pair-wise correlation measures on the variant allele frequencies (VAF) of the SNPs for sample-identity checks. All laboratories exhibited higher Pearson correlations between any two technical replicates than those between-sample group comparisons, indicating no sample swaps or mislabeling (**Supplementary Fig. 26**). Despite this, the correlation coefficients between replicates of the MAQC samples were relatively low, which were similar to values of 0.53 and 0.39 observed between replicates of MAQC A and B, respectively, in the RNA-seq data from the MAQC-III study. This may be attributed to the increased genome complexity of MAQC samples A and B, which were prepared by mixing total RNA samples from ten different cancer cell lines and from the brains of 23 donors, respectively¹².” (Results, lines 339–350)

Fig. 3 Quality flags of RNA-seq data.

Multiple quality metrics were used, including the number of sequencing reads, base quality (Q30), the percentage of reads mapped to the human genome (Mapping rate), gene body bias (5'-3' bias), the percentage of mapped reads that were located in the intergenic region (Intergenic region), SNR based on the MAQC and Quartet samples, the difference between the SNR calculated from any 17 out of 18 samples and the SNR calculated from all 18 samples (SNR17-SNR18), the percentage of reads mapped to ERCC sequences in samples without ERCC mixtures (Cross contamination), SNP-based sample-identity check, and final quality flag.

2. L177: The result shows that some laboratories/samples got very high duplication rates. Did you check the source of it, e.g., by looking at the distribution of duplicated reads of those samples? Did you filter those samples as outliers?

Reply: Thank you for your questions.

1) L177: The result shows that some laboratories/samples got very high duplication rates. Did you check the source of it, e.g., by looking at the distribution of duplicated reads of those samples?

The duplication rates of reads exceeded 30% in seven laboratories. According to your suggestion, we attempted to examine the distribution of duplicated reads to reveal their potential source. Using the dupRadar (v.3.18) package, we found a distinct distribution of duplicated reads across expression levels among these seven laboratories, exhibiting high duplication rates at both high and low expression levels. In contrast, in other RNA-seq data, the duplication rates at low expression levels were relatively low, while the duplication rates at high expression levels remained high. The duplicated reads at high

expression levels are believed to be derived from natural duplicates from highly expressed genes, whereas the abnormally high duplicated reads at low expression levels are attributed to PCR amplification bias (PMID: 27769170, PMID: 27156886).

Additionally, for three of these seven laboratories, the duplication rates were dispersed across all samples, with high standard deviations of $49.31\% \pm 10.3\%$, $38.05\% \pm 9.32\%$, and $34.02\% \pm 6.69\%$ across samples. Therefore, the duplication rates of individual samples in these laboratories are very high. We found a strong negative correlation between duplication rates and the library concentrations in these three laboratories, with Spearman correlation coefficients of -0.986, -0.886, and -0.889. This indicates that the low library concentration resulted in abnormally high duplication rates within the laboratory. Library concentration represents library complexity, and its negative relationship with duplication rates has been reported in previous studies (PMID: 37062860).

To clarify these, we have added necessary descriptions in the Results sections and **Supplementary Figure 20** in the Supplementary Materials as follows:

“These data were characterized by high duplication rates at both low and high expression levels (**Supplementary Fig. 20a**). Excessively duplicated reads at low expression levels were recognized to originate from PCR amplification bias^{31,32}. Within three of these seven laboratories, we also observed large dispersion in duplication rates for all samples, which exhibited a strong negative correlation with library concentration with Spearman correlation coefficients ranging from -0.886 to -0.986, highlighting the influence of the lack of library complexity (**Supplementary Fig. 20b**)³².” (Results, lines 309–316)

“The distribution of duplicated reads across expression levels were examined using dupRadar (v.3.18)³¹.” (Materials and Methods, lines 1463–1464)

Supplementary Figure 20. Distribution of abnormal duplication rates across laboratories and samples. (a) The dupRadar (v.3.18) tool was used to examine the sources of duplicated reads⁶. RNA-seq data from seven laboratories with high duplication rates (>30%) were included, and data from lab28 with a low duplication rate were utilized for comparison. The figure depicts the distribution of read duplication rates across different expression levels for the Quartet M8 sample. The red dashed line represents the cutoff of one read per base pair for high expression levels, while the green dashed line represents the cutoff of 0.5 FPKM for low expression levels. (b) The correlation between duplication rates and library concentrations in the three laboratories shows high standard deviations in duplication rates for all samples.

2) Did you filter those samples as outliers?

We flagged RNA-seq data from 7 laboratories exhibiting a high duplication rate of more than 30% as low quality, and filtered all samples from these laboratories out of subsequent analysis. We did not filter out a specific outlying sample from a given

laboratory. We also displayed these data in **Figure 3**, as shown above in our response to your comment 1(3).

3. L196: In Supplementary Figure 12, a subset of Quartet samples subjected to poly(A) selection exhibited a comparably low percentage of exonic fragments. Have you investigated potential reasons for this trend within this subset of samples?

Reply: Thank you for carefully pointing this out.

For the RNA-seq data from lab06, eight samples exhibited an average percentage of reads mapped to the exonic region as high as 85.71%, and the remaining 16 samples showed a comparatively lower average mapping rate of 47.47%, which was consistent with the characteristics of data obtained using the Poly(A) selection and the rRNA depletion methods, respectively. We reconducted a survey questionnaire on the library preparation and sequencing processes for lab06, which confirmed that the libraries of these two subgroups of samples were indeed prepared using the Poly(A) selection and the rRNA depletion method and sequenced in two different runs, respectively. Therefore, considering other quality control metrics, we flagged the RNA-seq data from lab06 as low-quality data and excluded them from subsequent analysis. We have updated **Supplementary Figures 12 and 13** (now labeled as **Supplementary Figures 23 and 24**) accordingly and clarified the different mRNA enrichment methods used by lab06 for samples in figure legend as follows:

Supplementary Figure 23. Percentage of mapped reads in exonic, intronic, and intergenic regions. For both the Quartet and MAQC samples, the Poly(A) selection protocol is consistently associated with a higher percentage of reads in the exonic regions. For lab06, the libraries for the MAQC samples and two Quartet M8 replicates (R1 and R2) were prepared using the Poly(A) selection method, while the remaining Quartet samples (one M8 replicate and F7, D5, and D6), as well as the mixed samples T1 and T2, were all prepared using the rRNA depletion method. The two subgroups of samples were sequenced in two different runs on the same sequencing instrument. * indicates that the Quartet samples was processed using two different mRNA enrichment methods in lab06.

Supplementary Figure 24. Percentage of exonic reads. There are significant higher exonic reads in the MAQC samples than the Quartet samples for both the Poly(A) selection method and the rRNA depletion method. This may be attributed to more highly expressed genes in the MAQC samples. *** indicates a p -value < 0.001 . The significance was tested using unpaired t-test.

4. L203: Have the RNA samples or libraries identified with potential contamination been excluded from subsequent analysis? Additionally, could you provide information on the number of samples that failed quality control and how many were used in the downstream analysis?

Reply: Thank you very much for your question and suggestion.

1) Have the RNA samples or libraries identified with potential contamination been excluded from subsequent analysis?

Yes, potential cross-contamination was identified in RNA samples or libraries from four laboratories, and RNA-seq data for all samples in these laboratories were excluded from subsequent analysis, as displayed in **Figure 3** (as shown above in our response to your comment 1(3)). These four laboratories exhibited a proportion of reads mapping to ERCC genes exceeding 0.005% in at least one sample without ERCC mixtures, as described in lines 350–360 of the revised manuscript.

“Finally, based on percentage of reads mapped to ERCC reference sequences allowed for the identification of problematic samples or libraries. In four samples (MAQC A, B, and Quartet F7, D5) without ERCC spike-ins, we observed reads counts ranging from 1 to 213,467 mapped to ERCC genes across 38 laboratories (**Supplementary Fig. 27**). Four laboratories showed more than 0.005% of reads mapped to ERCC genes in a specific sample. Particularly, lab10 exhibited an exceptionally high fraction of ERCC reads in the two replicates of MAQC sample A, accounting for 0.8% and 0.06% of the exonic reads. This indicates potential cross-contamination, possibly due to sample, library, or barcode contamination during library preparation³⁴, or misallocation of barcodes and carry-over contamination from previous samples during the sequencing process³⁵⁻³⁷.” (Results, lines 350–360)

2) Additionally, could you provide information on the number of samples that failed quality control and how many were used in the downstream analysis?

Using all QC metrics, RNA-seq data from 16 laboratories failed quality control, and the remaining data from 26 laboratories were used in downstream analysis. We have revised **Supplementary Table 3** to show the number of laboratories passing or failing each quality metric. We added a description of filtering low-quality data in the Results section, as shown above in our response to your comment 1(1).

5. L227: How many samples were classified as poor quality based on signal-to-noise ratio (SNR)? Are these samples the outliers depicted in Figure 2a?

Reply: Thank you for your questions.

Figure 2a depicts the overall quality of expression data from all laboratories, which is obtained from distinct analysis pipelines of each laboratory. Expression data from 19 laboratories were classified as poor quality based on SNR. Among them, 17 laboratories had SNR values based on the Quartet and mixed samples less than 12. Additionally, nine laboratories exhibited differences between SNR17 and SNR18 exceeding six, as depicted by outliers in Figure 2a, involving nine samples. We have modified the text to clarify this as follows:

“A total of 17 laboratories had SNR values based on the Quartet and mixed samples less than 12, which were considered as low quality.” (Results, lines 179–181)

“We observed that the SNR17 values increased by six decibels compared to the corresponding SNR18 values after excluding nine samples in nine laboratories, indicating that these nine samples were low-quality outliers.” (Results, lines 184–186)

We also examined SNR for all laboratories after applying the uniformed data analysis pipeline (Ensembl-STAR-StringTie) to eliminate the effects of different analysis workflows used by laboratories. RNA-seq data from 14 laboratories were classified as poor quality based on SNR. Among them, ten laboratories had SNR values of less than 12 based on the Quartet and mixed samples, while nine laboratories exhibited differences between SNR17 and SNR18 exceeding 6. We have added descriptions in the Results section and **Supplementary Figure 25** as follows:

“First, we examined SNR after applying the fixed data analysis pipeline to eliminate the effects of different bioinformatics workflows used by the laboratories. Data from 14 laboratories were classified as low quality (**Supplementary Figure 25**).” (Results, lines 336–339)

Supplementary Figure 25. SNR after applying the fixed data analysis pipeline. The expression data of 42 laboratories were obtained using the Ensembl-STAR-StringTie pipeline to mitigate the effects of different bioinformatics workflows used by the laboratories. Laboratories are ordered by SNR values. Dots represent SNR values based on any 17 of the 18 samples (12 Quartet and 6 mixed samples) in each laboratory. A dot in dark red represents the SNR17 outlier that increases over six decibels compared to its standard SNR (18-sample SNR) after excluding one low-quality sample in a laboratory, while a dot in cyan represents the SNR17 value that decreases or increases less than six decibels compared to its standard SNR. The red dashed line represents the cutoff of SNR at 12.

6. L370: “Best practices for experimental designs”, you have shown (from supplementary figure 29, 30 and Figure 5) that not a specific factor was identified that can specifically contribute to inter-laboratory variations. To better echo the section title, I think you should make it clearer in the main context to readers about your conclusion.

Reply: Thank you for your valuable suggestion.

In this section, our primary conclusion is that the quality of experimental execution significantly influences the performance of RNA-seq. Additionally, experimental factors such as mRNA enrichment methods, strandedness, and the number of reads mapped to exonic regions are associated with data quality and the quantification of gene expression. Therefore, comprehensive quality control and the selection of experimental protocols should be considered when designing an RNA-seq experiment. To make the conclusion clearer, we revised the main text in this section and **Supplementary Figure 29** (now labeled as **Supplementary Figure 35**) as follows:

“The low quality of experimental execution significantly influences the RNA-seq performance. RNA-seq data from 16 laboratories, failing multiple quality metrics, were considered low experimental quality in library preparation or sequencing processes (**Fig. 3**). Based on four types of ground truth, these laboratories exhibited lower Pearson correlation coefficients or higher RMSE for absolute and relative gene expression measurements, as well as lower MCC for DEG identification, compared to the other 26 laboratories (**Supplementary Fig. 35**). Noticeably, these instances of low quality were unrelated to the choice of experimental methods, as they were distributed across various experimental workflows. Therefore, our results highlighted the importance of multidimensional quality control in experimental design.” (Results, lines 441–450)

“Different experimental protocols tended to influence RNA-seq performance, which should be considered during experimental design. Based on multiple types of ground truths, we assessed the influence of experimental factors using RNA-seq data from 26 laboratories in four aspects: data quality, absolute and relative gene expression, and DEG identification (**Fig. 6**). (i) For expression data quality, the Poly(A) selection method exhibited higher SNR values than the rRNA depletion method. Considering the differences in gene types captured by the two methods, we further examined SNR values for different gene types and observed that the Poly(A) selection method primarily exhibited higher SNR values for protein-coding genes. Conversely, for other gene types, particularly snRNA, the rRNA depletion method demonstrated significantly higher SNR, indicating a more accurate capture of biological differences in these RNAs (**Supplementary Fig. 36**). (ii) For absolute expression levels, the influences of mRNA enrichment method and strandedness were observed, with the rRNA depletion method and stranded-specific libraries exhibiting higher correlation coefficients with the reference datasets. (iii) For relative gene expression and DEG levels, a higher number of reads mapped to the exonic regions showed a strong or moderate correlation with improved accuracy, likely due to more reliable detection of lowly expressed genes³⁸. To validate their impacts, we down-sampled RNA-seq data to different depths and drew similar conclusions. Specifically, lower read depths resulted in decreased accuracy of relative gene expression and DEG identification, while exerting a relatively minor overall impact on SNR and absolute expression levels. However, when sequencing depth decreased to extremely low levels, such as 10 Mb, SNR and accuracy of absolute expression detection also significantly declined

(Supplementary Fig. 37). Furthermore, the MAQC TaqMan datasets revealed significant differences in accuracy associated with read lengths. To further explore the impact of read lengths, we trimmed sequencing reads to different lengths ranging from 25 bp to 150 bp across four laboratories, revealing that longer sequencing reads tended to exhibit higher accuracy, especially in relative gene expression measurements, and either similar or higher accuracy in DEG identification (Supplementary Fig. 38).” (Results, lines 451–479)

Supplementary Figure 35. Low accuracy for low-quality RNA-seq data. Based on four types of ground truth, RNA-seq data from 16 laboratories flagged as low quality exhibited low accuracy for absolute, relative gene expression, and DEG measurements, compared to the remaining RNA-seq data from 26 laboratories.

7. L390: Sequencing was performed with only 100 and 150 base pairs of read length. The data does not conclusively demonstrate that 100bp constitutes best practice. Notably, 50 bp read lengths have been utilized in many projects, including TCGA, indicating the potential value of investigating shorter reads. Additionally, RNA fragment length represents a critical design factor worth examining in conjunction with read length to comprehensively assess performance.

Reply: Thank you for your valuable comments and suggestions.

1) L390: Sequencing was performed with only 100 and 150 base pairs of read length. The data does not conclusively demonstrate that 100 bp constitutes best practice. Notably, 50 bp read lengths have been utilized in many projects, including TCGA, indicating the potential value of investigating shorter reads.

We observed a statistically significant higher accuracy for RNA-seq data with read length of 100 bp compared to 150 bp solely based on the MAQC TaqMan datasets, while no such conclusion was made in the assessment based on other types of ground truth. As you mentioned, read length of 50 bp is also widely used by other projects, and has been demonstrated to exhibit comparable performance in differential expression analysis to that of 100 bp (PMID: 26100517). We agree with you that our observation was not conclusive and does not encompass widely used read lengths to provide best practices. Thus, we have selected RNA-seq data with read length of 150 bp from four laboratories representing different sequencing depth levels, and shortened reads into 100 bp, 75 bp, 50 bp, and 25 bp using Cutadapt (v.4.8). Then, we used STAR for alignment, StringTie for gene quantification, and edgeR for DEG identification. Based on the four types of ground truth, we observed that longer read lengths resulted in more accurate detection of relative gene expression, while the accuracy of absolute gene expression was similar across different read lengths. For DEG identification, the TaqMan datasets for Quartet or MAQC samples revealed similar MCC for different read lengths, whereas the Quartet reference datasets indicated higher MCC values for longer read lengths.

We have added descriptions about the impact of read length in the Results section as follows:

“Furthermore, the MAQC TaqMan datasets revealed significant differences in accuracy associated with read lengths. To further explore the impact of read length, we trimmed sequencing reads to different lengths across four laboratories, revealing that longer sequencing reads tended to exhibit higher accuracy, especially in relative gene expression measurements, and either similar or higher accuracy in DEG identification (Supplementary Fig. 38).” (Results, lines 474–479)

“Cutadapt (v.4.8)⁹⁴ was used to shorten read lengths into 100 bp, 75 bp, 50 bp, and 25 bp.” (Materials and Methods, lines 1461–1462)

Supplementary Figure 38. The impact of read length on the RNA-seq performance. RNA-seq data with read length of 150 bp from four laboratories were shortened into 100 bp, 75 bp, 50 bp, and 25 bp using Cutadapt (v.4.8)⁹. Based on four types of ground truth, the accuracy of absolute, relative gene expression, and identification of differentially expressed genes (DEGs) for different read lengths were examined.

2) Additionally, RNA fragment length represents a critical design factor worth examining in conjunction with read length to comprehensively assess performance.

The RNA fragment length encompasses the lengths of the two adapters, and the insert size (including the two paired reads and their inner distance). Generally, the insert size is considered a critical quality control metric, with longer insert sizes considered to help the identification and quantification of isoforms. To investigate the impact of insert size on gene quantification or DEG identification, we considered insert size as an experimental factor and analyzed its correlation with accuracy metrics. We observed a low Spearman correlation coefficient ranging from -0.51 to 0.365 between insert size and accuracy metrics. However, it should be noted that our analysis was conducted in the presence of other influencing factors, necessitating focused experimental designs to validate the effect of insert size more comprehensively. To clarify this, we have modified **Figure 6** to display the correlation analysis results and added a corresponding description in the Results section as follows:

“In addition, for insert size, a factor considered to influence gene or isoform identification and quantification, our results demonstrated a poor correlation with gene quantification accuracy under the involvement of other factors²². The impact of these factors needs further validation during method development through focused experimental designs controlling for other influencing factors.” (Results, lines 483–488)

Fig. 6 The influence of experimental factors based on different performance metrics

Performance metrics included SNR for data quality, Pearson correlation coefficient for accuracy of absolute and relative expression, RMSE for recovery of mixing ratios, and MCC for the accuracy of DEG identification. Exonic reads indicate the number of mapped reads located in the exonic regions. Exonic reads were considered as an experimental factor as they could be determined by the total sequencing depth and the mRNA enrichment method chosen. The impact of exonic reads on data quality and accuracy is evaluated using Spearman correlation analysis. Significance testing was conducted based on normal distribution assumptions using one-way analysis of variance (ANOVA) and paired t-tests; or, in cases where normal distribution was not observed, independent samples were subjected to Kruskal-Wallis test and Mann-Whitney U test. ** indicates a p-value < 0.05. ns, not significant; SNR, Signal-to-Noise

Ratio; RMSE, Root Mean Square Error; MCC, Matthews Correlation Coefficient; TN, True Negative; TP, True Positive; FN, False Negative; FP, False Positive.

8. L391: In Figure 5, how is exonic coverage defined? It sounds like a quality control (QC) outcome rather than an aspect of the experimental design.

Reply: Thank you for pointing out this.

1) In Figure 5, how is exonic coverage defined?

Exonic coverage in Figure 5 indicates the number of mapped reads located in exonic regions. To convey this clearly, we have revised "exon coverage" to "exonic reads" and added an explanation in the legend of **Figure 5** (now labeled as **Figure 6**).

“Exonic reads indicate the number of mapped reads located in the exonic regions.”
(Figure 6, lines 886–887)

2) It sounds like a quality control (QC) outcome rather than an aspect of the experimental design.

As you mentioned, the number of reads in the exonic regions is often used as a quality control metric to ensure that there are sufficient reads for reliable gene or transcript quantification. The number of exonic reads can also be considered as an experimental design factor, because the expected number of exonic reads can be determined by setting the experimental protocols when designing an RNA-seq experiment. For example, the number of exonic reads correlates with total sequencing depth, which could be determined by specifying RNA input, library input, and the number of PCR cycles. Additionally, the choice of the mRNA enrichment methods also determines the percentage of reads mapped to exonic regions. Therefore, we included exonic reads as an experimental factor to investigate its correlation with other measures of data quality and the accuracy of gene quantification and DEG identification. We have added a description of the links between exonic reads and experimental design factors in the legend of **Figure 5** (now labeled as **Figure 6**) as follows:

“Exonic reads were considered as an experimental factor as they could be determined by the total sequencing depth and the mRNA enrichment method chosen.” (Figure 6, lines 887–889)

9. L930: Given that ground truth expression values and DEGs are central to the paper's evaluations, it is essential to include a dedicated section in the methods part to describe in detail how the ground truths were generated.

Reply: Thank you for your suggestion.

We have moved the section "The ground truth inherent in RNA reference samples" from the Supplementary Materials to the Materials and Methods section, as follows:

“The ground truth inherent in RNA reference samples

The sample design introduced four types of ground truth, comprising three reference datasets: the Quartet reference datasets and the TaqMan datasets for the Quartet and MAQC samples, and two built-in truths: ERCC spike-in ratios and the mixing ratios in T1 and T2 samples.

(i) Quartet reference datasets

The Quartet reference datasets provided relative gene expression and DEG results. They were obtained from the Quartet data portal (<https://chinese-quartet.org/#/dashboard>)⁶¹, consisting of 31,155 results regarding gene relative expression and DEGs in comparisons of M8/D6, F7/D6, and D5/D6²³. After intersecting with the gene list of Ensembl release-109 annotation, the remaining 30,976 results (including 5,036 DEGs) comprised 76.8% of protein-coding genes, 13.7% of long non-coding RNAs, 1.1% of small non-coding RNAs, 6.4% of pseudogenes, and 1.9% of immunoglobulin/T-cell receptor gene segments.

(ii) TaqMan datasets for the Quartet samples

The TaqMan datasets for the Quartet samples included absolute, relative gene expression and DEG results (**Supplementary Table 7**). RT-qPCR assays were conducted for 90 genes selected from the Quartet reference datasets. Normalized gene expression was quantified using the Delta Ct method, with the *CIORF43* gene serving as the endogenous control⁶². Fold changes were calculated using the Delta Delta Ct method for the three Quartet sample comparisons (M8/D6, F7/D6, and D5/D6). A gene is classified as a DEG when the student's t-test *p-value* < 0.05 and fold change ≥ 2 or \leq

0.5¹⁵. The TaqMan datasets and the Quartet reference datasets have a high consistency in terms of fold change (Pearson correlation = 0.93), and DEG identification (88%, 162/184) (**Supplementary Fig. 58**).

(iii) TaqMan datasets for the MAQC samples

The TaqMan datasets for the MAQC samples also included absolute, relative gene expression and DEG results. Normalized expression values for 1044 genes were obtained from the Gene Expression Omnibus database (accession number GSE5350), and 830 genes remained after removing genes with undetectable CT values (CT>35 or CT=0). Fold changes between the MAQC samples A and B were calculated as the ratios between the mean expression values of the four technical replicates. DEGs were identified using the same method for the Quartet TaqMan datasets.

(iv) The built-in truth of ERCC spike-in ratios

ERCC spike-in ratios were used to assess both absolute and relative gene expression accuracy. ERCC RNA controls comprised 92 genes in Mix 1 and Mix 2 with known wide-ranging concentrations, which were used for assessing absolute expression levels. ERCC genes could be further divided into four subgroups labeled as A, B, C, and D with respective ratios of 4:1, 1:2, 2:3, and 1:1 between Mix 1 and Mix 2 for each group, which facilitated the assessment of relative expression levels.

(v) The built-in truth of mixing ratios in samples T1 and T2

This built-in truth of mixing ratios represents expected ratios in relative expression between T1/D6 or T2/D6 and M8/D6, which was used to assess the accuracy and reproducibility of relative gene expression. Translating the known mixing ratios of 3:1 and 1:3 in RNA samples into gene expression ratios between samples followed methods described in previous studies¹⁴, involving two steps.

First, the expected absolute gene expression of genes in T1 and T2 should be calculable from the absolute expression in M8 and D6 using Equation (1):

$$\log_2 E_{T1/T2} = \log_2 (k1 \times E_{M8} + k2 \times E_{D6}) \quad (1)$$

where $E_{T1/T2}$ represents the expected absolute gene expression in T1 or T2, and E_{M8} and E_{D6} respectively represent the observed absolute gene expression in M8 and D6. $k1$ and $k2$ represent mixing coefficients for calculating gene expression in T1 and T2, respectively.

Second, using gene expression in D6 as the common denominator, the expected relative gene expression in comparisons of T1/D6 or T2/D6 could be calculated from the relative expression observed in the comparison of M8/D6, utilizing Equation (2):

$$\begin{aligned}\log_2 E_{\frac{T1}{D6}} &= \log_2 \left(k1 + k2 \times 2^{\log_2 E_{\frac{M8}{D6}}} \right) \\ \log_2 E_{\frac{T2}{D6}} &= \log_2 \left(k2 + k1 \times 2^{\log_2 E_{\frac{M8}{D6}}} \right)\end{aligned}\quad (2)$$

where $E_{\frac{T1}{D6}/\frac{T2}{D6}}$ represents the expected relative gene expression in comparisons of T1/D6 or T2/D6, and $E_{\frac{M8}{D6}}$ represents the observed relative gene expression in the comparison of M8/D6. Due to potentially different mRNA proportions in different samples, as stated by Shippy R et al.⁶³, the mixing coefficients for M8 and D6 may not adhere strictly to the theoretical values of 1/4 and 3/4. A correction value, z , was introduced to correct the mixing coefficients: $k1 = z/(z+3)$ and $k2 = 3z/(3z+1)$, which was determined using RT-qPCR assays. In brief, ten genes with a broad range of expression levels were tested using RT-qPCR, and the corresponding z -values for each gene were calculated using Equation (1) based on gene expression results from different samples. The obtained average z values were 0.974 ± 0.06 for T1 and 0.949 ± 0.09 for T2. To validate the z -values, RNA-seq data from the top ten laboratories with higher accuracy for recovering ERCC spike-in ratios were used, resulting in z -values of 0.965 ± 0.024 and 0.941 ± 0.026 for T1 and T2, respectively. Finally, the z values from RT-qPCR assays were used.” (Materials and Methods, lines 1197–1265)

10. L1047: Bioinformatics pipelines should be both reproducible and publicly available. Workflow languages, such as CWL, WDL, Nextflow, or Snakemake, are recommended to construct the pipelines, as they can help simplify and standardize workflows, thus facilitating easier sharing and reproduction. Furthermore, in addition to the focus on data availability (L1121), I would like to suggest that the authors make the reproducible code/workflows publicly accessible alongside the manuscript.

Reply: Thank you for your suggestion.

We have integrated the two types of gene annotation, three alignment tools, and eight quantification tools into a single workflow using Snakemake. Users can input sequencing data and specify the desired analysis pipeline by selecting parameters, thereby obtaining gene expression data. Since conducting differential expression

analysis involves some customized analysis steps, such as selecting a series of thresholds for filtering low-expression genes for optimal performance and comparing them with the reference datasets, we separately provided custom scripts written in R for these purposes. Additionally, we have also uploaded scripts for the assessment of gene expression accuracy based on the ground truth. We have added descriptions in the Data availability section as follows:

“The 28 gene quantification pipelines were integrated using Snakemake and are available on GitHub at https://github.com/lyaqing/snakemake_rnaseq.git. Custom scripts for the assessment of data quality and accuracy of gene expression and DEGs are available on GitHub at <https://github.com/wangduo-ux/Assessment-of-RNA-seq-performance.git>.” (Materials and Methods, line 1472–1476)

Reviewer #2 (Remarks to the Author):

Toward Best Practice in Identifying Subtle Differential Expression with RNA-seq: A Real-World Multi-Center Benchmarking Study Using Quartet and MAQC Reference Materials

This study thoroughly evaluated real-world RNA-seq by analysing Quartet and MAQC samples, particularly when detecting subtle differential expression levels. They also investigated sources of variation in gene expression data at experimental and bioinformatics aspects. In several studies, it has been already discovered that different sample processing protocols, platforms, and analysis pipelines used across laboratories compromise the accuracy and reproducibility of RNA-seq. In this study, they are also trying to find the sources of inter-laboratory variations. They generated RNA-seq data across 45 independent laboratories using Quartet RNA samples with spike-ins of ERCC controls, and MAQC RNA samples, where each laboratory used their in-house experimental protocol and analysis pipeline. The insights they provide in this paper are interesting and informative however certain aspects mentioned are vague and require further elaboration. Specifically, the method section lacks explanations and details about the evaluation of variations and reproducibility. For example, in evaluating the variation introduced by each bioinformatics tool, the methodology to mask the effect of other steps of analysis and their contribution in causing variations needs to be explained. Please address the following comments:

Reply: Thank you very much for considering the insights provided in our paper "interesting and informative". Your constructive comments and suggestions proved to be very helpful to improve our manuscript. In particular, we have provided more detailed information in the Materials and Methods section.

For example, in response to your example question, we have explained the principles of quantifying variations in gene expression differences attributed to each factor using principal variance component analysis (PVCA) in reply to your sixth comment, and have added the details in the Materials and Methods section accordingly.

1- How do you define subtle differential expressions?

Reply: Thank you for your question.

We use the term "subtle differential expression" to refer to overall minor differences in gene or transcript expression between sample groups, manifesting as a small number of differentially expressed genes (DEGs) or transcripts (DETs). This is similar to the expression differences observed between clinically relevant sample groups, such as between different disease subtypes or stages. For the two types of RNA reference materials used in this study, the Quartet samples could be considered to represent subtle differential expression between sample groups (D5, D6, M7, and M8), which demonstrate an average of around 2,000 DEGs, compared to more than 15,000 DEGs observed between the two MAQC samples (A and B).

We have added an explanation of subtle differential expression in the Introduction section, and clarified that our study refers subtle differential expression as smaller gene expression differences between the Quartet samples in comparison to those between the MAQC samples. The modified text is as follows:

“Noticeably, clinically relevant biological differences among study groups are often small, especially between certain diseases and normal tissues^{5, 6}, or between different disease subtypes or stages⁷⁻¹¹, implying minor changes in gene expression between sample types. We refer to such minor expression differences between sample groups with similar transcriptome profiles as subtle differential expression, which typically manifests in the detection of fewer differentially expressed genes (DEGs).” (Introduction, lines 58–64)

“Thus, the Quartet samples could reflect subtle differential expression, providing a unique opportunity for the assessment and benchmarking of transcriptome profiling at subtle differential expression levels in a reference-based manner.” (Introduction, lines 103–106)

2- How built-in truth is known based on mixed-ratios of T1 and T2 (lines 231 to 233)? I am not convinced how fold change can be used for T1 and T2 in measuring the accuracy and reproducibility (Supplementary notes section 2.1).

Reply: Thank you for pointing out this.

1) How built-in truth is known based on mixed-ratios of T1 and T2 (lines 231 to 233)?

The built-in truth based on mixing ratios of T1 and T2 refers to the expected ratios of relative gene expression between samples T1 or T2, and M8, with the D6 serving as a denominator for relative expression calculation. The expected ratios in gene expression are derived from the mixing ratios of RNA samples (T1 and T2). Similar to the previous MAQC-III (SEQC) study (PMID: 25150838), the translation of mixing ratios of RNA samples into gene expression ratios followed two steps: (i) calculating expected absolute gene expression in T1 and T2 from that in M8 and D6 using mixing coefficients, (ii) calculating expected relative gene expression in T1 or T2 from those in M8. To further clarify the details of generating built-in truth based on mixed ratios of T1 and T2, we have added the description in the Materials and Methods section as follows:

“(v) The built-in truth of mixing ratios in samples T1 and T2

This built-in truth of mixing ratios represents expected ratios in relative expression between T1/D6 or T2/D6 and M8/D6, which was used to assess the accuracy and reproducibility of relative gene expression. Translating the known mixing ratios of 3:1 and 1:3 in RNA samples into gene expression ratios between samples followed methods described in previous studies¹⁴, involving two steps.

First, the expected absolute gene expression of genes in T1 and T2 should be calculable from the absolute expression in M8 and D6 using Equation (1):

$$\log_2 E_{T1/T2} = \log_2 (k1 \times E_{M8} + k2 \times E_{D6}) \quad (1)$$

where $E_{T1/T2}$ represents the expected absolute gene expression in T1 or T2, and E_{M8} and E_{D6} respectively represent the observed absolute gene expression in M8 and D6. $k1$ and $k2$ represent mixing coefficients for calculating gene expression in T1 and T2, respectively.

Second, using gene expression in D6 as the common denominator, the expected relative gene expression in comparisons of T1/D6 or T2/D6 could be calculated from the relative expression observed in the comparison of M8/D6, utilizing Equation (2):

$$\begin{aligned}\log_2 E_{\frac{T1}{D6}} &= \log_2 \left(k1 + k2 \times 2^{\frac{\log_2 E_{M8}}{D6}} \right) \\ \log_2 E_{\frac{T2}{D6}} &= \log_2 \left(k2 + k1 \times 2^{\frac{\log_2 E_{M8}}{D6}} \right)\end{aligned}\quad (2)$$

where $E_{\frac{T1}{D6}/\frac{T2}{D6}}$ represents the expected relative gene expression in comparisons of T1/D6 or T2/D6, and $E_{\frac{M8}{D6}}$ represents the observed relative gene expression in the comparison of M8/D6. Due to potentially different mRNA proportions in different samples, as stated by Shippy R et al.⁶³, the mixing coefficients for M8 and D6 may not adhere strictly to the theoretical values of 1/4 and 3/4. A correction value, z , was introduced to correct the mixing coefficients: $k1 = z/(z+3)$ and $k2 = 3z/(3z+1)$, which was determined using RT-qPCR assays. In brief, ten genes with a broad range of expression levels were tested using RT-qPCR, and the corresponding z -values for each gene were calculated using Equation (1) based on gene expression results from different samples. The obtained average z values were 0.974 ± 0.06 for T1 and 0.949 ± 0.09 for T2. To validate the z -values, RNA-seq data from the top ten laboratories with higher accuracy for recovering ERCC spike-in ratios were used, resulting in z -values of 0.965 ± 0.024 and 0.941 ± 0.026 for T1 and T2, respectively. Finally, the z values from RT-qPCR assays were used.” (Materials and Methods, lines 1237–1265)

2) I am not convinced how fold change can be used for T1 and T2 in measuring the accuracy and reproducibility (Supplementary notes section 2.1).

We performed nonlinear robust fit for relative gene expression in comparisons of T1/D6 or T2/D6, and those in the comparison of M8/D6 for each laboratory, and compared the fitting curves to the expected curve defined by Equation (2) above. As shown in Figure 2h, a high degree of clustering among the points representing genes suggests high reproducibility within a laboratory, whereas widely scattered points indicate low reproducibility. Furthermore, if the points closely adhere to the expected curve, it indicates high accuracy, while deviation from the curve suggests low accuracy. Additionally, we used RSME values to quantify the deviations between the expected and observed relative gene expression in comparisons of T1/D6 or T2/D6, which reflect the accuracy. Therefore, the mixing ratios of T1 and T2 provided a reasonable assessment of the accuracy of RNA-seq in capturing known gene expression ratios and the reproducibility of gene quantification within the laboratory. The approach of using

predefined mixing ratios as "ground truth" to assess RNA-seq or microarray performance has been employed in previous studies, and is considered an effective method in the absence of a gold standard method (PMID: 27899618, PMID: 25150838, and PMID: 16964226). We have added an explanation of applying mixing ratios to assess performance in the Materials and Methods section as follows:

“The built-in truth of mixing ratios in T1 and T2 was employed to assess the accuracy of RNA-seq in capturing known gene expression ratios between samples and the reproducibility of gene quantification within the laboratory^{14, 63, 64}. A nonlinear robust fit (nlrob) was performed for relative gene expression in comparisons of T1/D6 or T2/D6 and M8/D6 for each laboratory, and the fitting curves were compared to the expected curves defined by Equation (2) (**Fig. 1b**). The closeness between the laboratory data and the expected curve indicates accuracy, while the clustering of laboratory data represents reproducibility. Additionally, the RMSE between the observed and the expected relative expression from Equation (2) was calculated for comparisons of T1/D6 or T2/D6.” (Materials and Methods, lines 1289–1298)

3- What does SNR17 and SNR18 mean and how the difference between SNR17 and SNR18 are related to the identification of random failures in technical replicates? What do you mean by random failures (lines 227 to 230)?

Reply: Thank you for your questions.

1) What does SNR17 and SNR18 mean and how the difference between SNR17 and SNR18 are related to the identification of random failures in technical replicates?

SNR18 indicates the SNR values calculated using the gene expression profiles of all the Quartet and mixing samples, including M8, F7, D5, D6, T1, and T2 (a total of 18 RNA-seq libraries, with three technical replicates for each of the six sample groups). SNR17 indicates the SNR values calculated using the gene expression profiles of any 17 out of the 18 samples. When the removal of any technical replicate leads to a significant increase in the SNR17 value compared to the SNR18, it implies that the removed sample is an outlier with low quality resulting from random library or

sequencing failures. We have further explained SNR18, SNR17, and the difference between them in the Materials and Methods section as follows:

“A gene was included for PCA analysis and subsequent SNR calculation if more than one read was mapped to it in any one of the samples included. SNR calculated using the absolute gene expression profiles from 24 samples including the MAQC samples (SNR24) and 18 samples excluding the MAQC samples (SNR18), represented the ability to distinguish large and subtle biological differences from technical noises among technical replicates, respectively. Additionally, SNR17 was calculated from any 17 out of the 18 samples (12 Quartet and 6 mixed samples).” (Materials and Methods, lines 1268–1275)

2) What do you mean by random failures (lines 227 to 230)?

Random failure refers to occasional quality outliers of library construction or sequencing for individual samples within a test batch due to random factors such as pipetting errors. It is in contrast to systematic failure, which indicates low quality that occurs in a deterministic way to a certain cause across all samples in a batch. To convey this more clearly, we have added the explanation for random failure in the Results and the Materials and Methods sections as follows:

“Moreover, SNR examinations allowed for identifying random library or sequencing failures in the individual replicates by calculating SNR17 from any 17 out of the 18 samples (12 Quartet and 6 mixed samples). We observed that the SNR17 values increased by six decibels compared to the corresponding SNR18 values after excluding nine samples in nine laboratories, indicating that these nine samples are low-quality outliers (**Fig. 2a**).” (Results, lines 181–186)

“If the removal of a particular sample results in a significant increase in SNR17 compared to SNR18 (with a difference greater than 6), the quality of the RNA-seq data of the excluded sample is considered low and the sample is treated as an outlier. Such random occurrences of problematic library preparation or sequencing in individual samples are referred to as random failures.” (Materials and Methods, lines 1275–1280)

4- What is DEG classification here, does it mean the same genes are reported as DEGs? The number of DEGs says nothing about the consistency based on my understanding (lines 280 to 282).

Reply: Thank you for your comment and question.

We use “DEG classification” to represent genes detected with or without differences in expression levels between two sample groups, reported as differentially expressed genes (DEGs) or non-DEGs, respectively. To convey this more clearly, we have revised the term “DEG identification” throughout the manuscript, following a previous study (PMID: 37507115).

As you mentioned, the number of DEGs cannot serve as a metric representing consistency. We initially intended to display differences in the number of DEGs between laboratories, without conveying consistency. Here, we primarily relied on the penalized MCC metric to assess the accuracy and consistency of DEGs across laboratories. To avoid confusion, we modified the text (lines 280 to 282 in the original manuscript) as follows:

“The accuracy of DEG identification, as assessed based on the Quartet reference datasets and the TaqMan datasets for Quartet and MAQC samples, also exhibited variations across laboratories. Due to the varied number of genes inputted for differential expression analysis, true positives ranging from 0.03% to 78.6%, from 1.2% to 82.0%, and from 0.2% to 52.9% of the three reference datasets, respectively, were not reported across laboratories. Consequently, we categorized these instances as false negatives, and employed a penalized Matthews Correlation Coefficient (MCC) to assess the accuracy of DEG identification (**Fig. 1b and Supplementary Notes, Section 2.1**).” (Results, lines 238–246)

5- It would make reading the paper easier if you just mentioned the 17 factors, rather than referring to the supplementary table (lines 319 and 320).

Reply: Thank you for the great suggestion.

We have added the specific factors in experimental and bioinformatics processes into the Materials and Methods section as follows:

“We examined a total of 15 experimental factors, including RNA input, mRNA enrichment method, strandedness, library kit, library concentration, read length, sequencing platform, flowcell, lane, base quality, GC content, duplication rate, the number of reads in exonic region, insert size, and 5' to 3' bias, and four bioinformatics factors, including annotation type, alignment tool, quantification tool, and normalization method. Sample grouping also served as a factor in PVCA, allowing for a comparison between biological differences among samples and variations introduced by various factors.” (Materials and Methods, lines 1362–1369)

6- How the percentage of variation is calculated in line 324 and how do you know they were specifically coming from these factors? I did not find any explanation about this part in the method section.

Also what are the variations based on, is it DEG or absolute differences? The same applies to the percentages of variations calculated for bioinformatics tools/steps. How do you know they are exactly coming from a specific tool? How do you exclude the influence of other factors?

Reply: Thank you for your questions.

1) How the percentage of variation is calculated in line 324 and how do you know they were specifically coming from these factors? I did not find any explanation about this part in the method section. The same applies to the percentages of variations calculated for bioinformatics tools/steps. How do you know they are exactly coming from a specific tool? How do you exclude the influence of other factors?

We performed principal variance component analysis (PVCA) to evaluate the sources of variation from experimental and bioinformatics processes using the pvca package (v.1.40.0), which was initially used for assessing batch effects in microarray experiments (PMID: 18549499), and later applied to analyze factors contributing to variations in RNA-seq expression data as well (PMID: 23393029). PVCA leverages the strengths of two popular data analysis methods, including principal component analysis (PCA) for efficiently reducing data dimensionality, initially proposed by Karl Pearson

in 1901 (DOI: 10.1080/14786440109462720), and variance components analysis (VCA) fitting a mixed linear model. The first few principal components are selected to retain the majority of the variations in gene expression data. Then, the particular principal components were considered as a univariate response to fit a linear mixed model. All experimental or bioinformatics factors as well as the interactions of these factors are considered as random effects in the mixed model. Finally, the variance component for each factor is estimated through restricted maximum likelihood (REML) and averaged with their corresponding eigenvalues as weights.

Therefore, here the percentage of variation, such as 17.9% described in line 324 of the original manuscript, is calculated as the proportion of average variance components for each experimental or bioinformatics factor in the total variance. For experimental or bioinformatics factors, the variance components specific to each factor are obtained through REML, wherein the likelihood function between the random effects and the observed data, such as gene expression data, is constructed, and the variance parameters of all random effects in the mixed linear model are iteratively adjusted to maximize the likelihood function. We have added an explanation of the PVCA analysis in the Materials and Methods section as follows:

“Principal variance component analysis

Principal variance component analysis (PVCA) was performed to estimate the sources of variation in absolute or relative expression profiles from the experimental and bioinformatics processes using the pvca package (v.1.40.0)⁷². In brief, after filtering out low-quality data, the expression data for all samples from the remaining RNA-seq data were used as the input for PVCA. PVCA involved four basic steps: (i) perform PCA and select the first few principal components to retain the majority of the variations in the gene expression data, (ii) fit a mixed linear model separately to each principal component with all factors of interest as random effects, (iii) estimate the variance components of each factor through restricted maximum likelihood and average the estimated variance components with their corresponding eigenvalues as weights for each factor, and (iv) standardize the weighted average variance components by dividing by their sum, so that the proportion of variance components reflects the magnitude of sources of variation⁷³.” (Materials and Methods, lines 1349–1362)

2) Also what are the variations based on, is it DEG or absolute differences?

The variations we mentioned refer to the differences in absolute or relative gene expression levels caused by different laboratories, experimental processes, and analysis pipelines. To clarify this, we further specified the gene expression data utilized in the "Sources of variation from experimental/bioinformatics processes" section as follows:

“The significant inter-laboratory variations necessitated the investigation of the sources. Here, we focused on the magnitude of variations across 26 laboratories in absolute or relative gene expression introduced by each RNA-seq process.” (Results, lines 369–371)

“The inter-laboratory variations in absolute gene expression levels introduced by experimental processes were significant in both Quartet (**Fig. 4a**) and MAQC samples (**Fig. 4b**), especially impacting subtle differential expression measurements.” (Results, lines 384–386)

“In relative gene expression levels, the proportion of variation attributed to experimental factors decreased to below 20% for both the Quartet and MAQC samples, which was observed when employing two different fixed analysis pipelines (**Supplementary Fig. 32**)” (Results, lines 403–406)

“In absolute gene expression levels, each bioinformatics step introduced variations, with a greater impact on the detection of subtle differential expression (**Figs. 5a–b**).” (Results, lines 422–423)

“The relative gene expression also helped reduce variations from bioinformatics processes, as indicated by the increased consistency of relative gene expression across different analysis pipelines compared to absolute expression (**Figs. 5c–d**).” (Results, lines 430–432)

7- in line 445 it is stated that “the performance ranking of all quantification pipelines supported the superior performance of opting for Ensembl gene annotation, any alignment tool, and either featureCounts or HTSeq for quantification”. Was RMSE the

lowest over all reference datasets or did you take the average over all three to make such a conclusion? Is RMSE calculated based on absolute expression levels? What are the colors in Figure 38 in supplementary material? Some quantification tools return TPM and some return gene counts, did you convert them all into one gene count before comparing?

Reply: Thank you for your questions and call for clarification.

1) in line 445 it is stated that “the performance ranking of all quantification pipelines supported the superior performance of opting for Ensembl gene annotation, any alignment tool, and either featureCounts or HTSeq for quantification”. Was RMSE the lowest over all reference datasets or did you take the average over all three to make such a conclusion? What are the colors in Figure 38 in supplementary material?

As you mentioned, the quantification pipelines with the lowest RMSE were considered high accuracy. We ranked all pipelines using RMSE based on each ground truth, and then averaged the rankings based on the four types of ground truth to obtain the final ranking. To clearly show the performance ranking of 28 quantification pipelines based on four types of ground truth, we have revised **Supplementary Figure 38** (now labeled as **Supplementary Figure 48**) and added the explanations about the performance ranking in the figure legend. The colors in Supplementary Figure 38 in the original manuscript indicate high-quality benchmark datasets from 13 laboratories, and there are no different colors in the revised figure.

	Overall Rank	Rank	mean ± sd	Rank	mean ± sd	Rank	mean ± sd	Rank	mean ± sd	Rank	mean ± sd	RMSE
RefSeq_HISAT2_StringTie	1	9	0.314 ± 0.041	10	0.883 ± 0.106	2	1.137 ± 0.066	2	0.421 ± 0.084	2	0.825 ± 0.184	
RefSeq_STAR_StringTie	2	10	0.315 ± 0.044	8	0.881 ± 0.113	3	1.139 ± 0.065	1	0.42 ± 0.085	4	0.84 ± 0.193	
Ensembl_Subread_featureCounts	3	1	0.289 ± 0.032	1	0.834 ± 0.087	8	1.159 ± 0.067	10	0.488 ± 0.097	9	0.901 ± 0.201	
Ensembl_STAR_featureCounts	4	2	0.294 ± 0.032	5	0.873 ± 0.101	9	1.163 ± 0.066	6	0.484 ± 0.099	8	0.898 ± 0.203	
Ensembl_HISAT2_StringTie	5	8	0.308 ± 0.041	14	0.905 ± 0.111	5	1.157 ± 0.066	3	0.428 ± 0.087	1	0.823 ± 0.183	
Ensembl_STAR_StringTie	6	11	0.316 ± 0.042	13	0.905 ± 0.109	6	1.159 ± 0.066	4	0.43 ± 0.085	3	0.839 ± 0.192	
Ensembl_STAR_HTSSeq	7	5	0.302 ± 0.033	3	0.873 ± 0.1	13	1.169 ± 0.066	7	0.485 ± 0.098	11	0.902 ± 0.204	
Ensembl_HISAT2_featureCounts	8	3	0.296 ± 0.033	7	0.877 ± 0.1	12	1.167 ± 0.068	12	0.49 ± 0.101	10	0.902 ± 0.2	
Ensembl_STAR_STAR	9	6	0.302 ± 0.033	4	0.873 ± 0.1	14	1.169 ± 0.066	8	0.485 ± 0.098	12	0.902 ± 0.204	
Ensembl_HISAT2_HTSSeq	10	7	0.304 ± 0.034	6	0.877 ± 0.099	15	1.17 ± 0.067	13	0.491 ± 0.1	7	0.897 ± 0.202	
Ensembl_Subread_HTSSeq	11	4	0.3 ± 0.033	2	0.849 ± 0.094	11	1.167 ± 0.066	14	0.494 ± 0.096	18	0.912 ± 0.205	
RefSeq_Subread_StringTie	12	20	0.38 ± 0.048	9	0.882 ± 0.112	19	1.19 ± 0.072	5	0.463 ± 0.081	6	0.849 ± 0.192	
Ensembl_Subread_StringTie	13	23	0.393 ± 0.039	12	0.892 ± 0.099	21	1.207 ± 0.068	9	0.488 ± 0.081	5	0.844 ± 0.191	
RefSeq_Salmon	14	22	0.388 ± 0.058	17	0.932 ± 0.103	1	1.123 ± 0.048	15	0.495 ± 0.09	22	0.976 ± 0.219	
RefSeq_HISAT2_featureCounts	15	19	0.377 ± 0.044	19	0.935 ± 0.124	4	1.151 ± 0.083	22	0.534 ± 0.107	16	0.91 ± 0.204	
RefSeq_HISAT2_featureCounts	16	18	0.373 ± 0.044	21	0.936 ± 0.122	7	1.159 ± 0.079	21	0.531 ± 0.106	17	0.911 ± 0.202	
RefSeq_Subread_featureCounts	17	12	0.333 ± 0.041	18	0.933 ± 0.097	16	1.173 ± 0.08	19	0.526 ± 0.103	19	0.914 ± 0.204	
RefSeq_Subread_HTSSeq	18	14	0.343 ± 0.041	20	0.936 ± 0.092	10	1.166 ± 0.079	20	0.529 ± 0.104	20	0.926 ± 0.207	
RefSeq_STAR_featureCounts	19	16	0.365 ± 0.036	24	0.947 ± 0.107	23	1.214 ± 0.067	16	0.514 ± 0.104	13	0.907 ± 0.205	
RefSeq_STAR_HTSSeq	20	17	0.369 ± 0.035	22	0.946 ± 0.109	22	1.207 ± 0.069	18	0.516 ± 0.105	14	0.91 ± 0.206	
RefSeq_kallisto	21	15	0.359 ± 0.048	11	0.892 ± 0.099	17	1.175 ± 0.094	25	0.657 ± 0.106	26	1.008 ± 0.217	
RefSeq_STAR_STAR	22	21	0.385 ± 0.033	23	0.946 ± 0.109	26	1.255 ± 0.064	17	0.514 ± 0.105	15	0.91 ± 0.206	
RefSeq_Sailfish	23	13	0.341 ± 0.043	15	0.916 ± 0.102	24	1.223 ± 0.1	27	0.708 ± 0.104	24	0.983 ± 0.225	
RefSeq_Bowtie_RSEM	24	27	0.494 ± 0.064	16	0.92 ± 0.128	25	1.223 ± 0.068	11	0.489 ± 0.114	27	1.09 ± 0.276	
Ensembl_Salmon	25	28	0.524 ± 0.073	26	1.025 ± 0.122	18	1.185 ± 0.071	24	0.575 ± 0.102	23	0.979 ± 0.22	
Ensembl_Bowtie_RSEM	26	25	0.447 ± 0.078	25	1.003 ± 0.137	20	1.192 ± 0.079	23	0.55 ± 0.119	28	1.09 ± 0.275	
Ensembl_Sailfish	27	24	0.445 ± 0.046	28	1.046 ± 0.125	28	1.342 ± 0.118	28	0.736 ± 0.103	21	0.976 ± 0.221	
Ensembl_kallisto	28	26	0.459 ± 0.046	27	1.046 ± 0.112	27	1.269 ± 0.109	26	0.694 ± 0.106	25	0.989 ± 0.216	
Ground truth												
			Quartet Reference datasets		Quartet TaqMan datasets		MAQC TaqMan datasets		Mixing ratios in T1 and T2		ERCC spike-in Ratios	

Supplementary Figure 48. The performance ranking of 28 quantification pipelines at relative expression levels. The four types of ground truths, arranged from left to right, include the Quartet reference datasets, the TaqMan datasets for the Quartet and MAQC samples, the built-in truth regarding the recovery of mixed ratios in T1 and T2, and the built-in truth of ERCC spike-in ratios. The average Root Mean Square Error (RMSE) between relative gene expression and the ground truth was assessed across 13 benchmark datasets. Rankings were determined in ascending order based on RMSE, and the final ranking was calculated as the average of rankings derived from the five truths.

2) Is RMSE calculated based on absolute expression levels?

We used the Quartet reference datasets, the Quartet and MAQC TaqMan datasets, and the built-in truth of the mixing ratios in T1 and T2 as “ground truths”, and RMSE was calculated based on relative expression levels. We have clarified this in the Results section as follows:

“We further examined the accuracy of these pipelines for relative expression measurements by comparing them to three reference datasets and the two types of built-

in truths, revealing similar clustering patterns based on the accuracy metric (RMSE) (Fig. 7a)." (Results, lines 528–531)

"Overall, our performance ranking of all quantification pipelines for relative gene expression measurements based on the four types of ground truth supported the selection of Ensembl gene annotation (or RefSeq annotation when using StringTie) and genome-alignment quantification strategy for gene quantification purposes (Supplementary Fig. 48)." (Results, lines 546–550)

3) Some quantification tools return TPM and some return gene counts, did you convert them all into one gene count before comparing?

As you pointed out, some quantification tools, such as Salmon, kallisto, and Sailfish, return TPM, while featureCounts and HTSeq return gene counts. We converted them into FPKM for comparison between tools and for comparison with the reference datasets. We have clarified this in the Materials and Methods section as follows:

"The outputs of all quantification tools were converted to FPKM for comparison with the ground truth or between tools." (Materials and Methods, lines 1418–1419)

8- RSEM uses either Bowtie, Bowtie2, or Star for alignment. Which one did you use? Please clarify that in your figures and methods section.

Reply: Thank you for carefully pointing this out.

RSEM used Bowtie for alignment. We have modified "Ensembl_RSME" to "Ensembl_Bowtie_RSME" and "RefSeq_RSEM" to "RefSeq_Bowtie_RSEM" in **Figures 4 and 7** (now labeled as **Figures 5 and 8**) and **Supplementary Figures 34-35 and 37-38** (now labeled as **Supplementary Figures 45, 46, and 48**), and clarified this in the Materials and Methods section as follows:

"Eight gene or transcript quantification tools were included, consisting of genome-alignment quantification tools like featureCounts (v.2.0.3)⁷⁸, HTSeq (v.2.0.2)⁷⁹, StringTie (v.2.2.1)⁶⁷, and STAR, one transcriptome-alignment quantification tool, RSEM (v.1.3.1)⁸⁰ with Bowtie (v.1.3.1) for alignment⁸¹, as well as three alignment-free

quantification tools, including kallisto (v.0.48.0)⁷⁵, Salmon (v.1.10.1)⁶⁸, and Sailfish (v.0.9.0)⁷⁶. According to the different quantification principles, these quantification tools can be divided into exon-level tools (featureCounts, HTSeq, and STAR) and transcript-level tools (Kallisto, Salmon, Sailfish, RSEM, and StringTie), which quantify gene expression by counting or estimating reads mapped to exons or transcripts, respectively.” (Materials and Methods, lines 1387–1396)

9- It was concluded that “the impact of different alignment tools on differential expression measurement is minimal”. Please add more details on this. Alignment is one of the important steps in the quantification of RNA-seq and most of the quantification tools rely on the input alignment for the quantification. Different alignment tools handle ambiguity in different ways and therefore that will impact the results of the quantification tools. I am not convinced that alignment tools can be picked arbitrarily in quantification pipelines.

Reply: Thank you for your insightful comments.

As you mentioned, different alignment tools handle ambiguous sequencing reads in different ways, which may potentially impact the quantification of certain genes. Previous studies have shown that different quantification tools exhibit significant variations in alignment efficiency and accuracy, particularly for samples with high genomic complexity (PMID: 27941783). Moreover, some alignment tools tend to misalign sequencing reads, especially for ambiguous genes such as pseudogenes (PMID: 38066001). These lead to differences in the accuracy of gene quantification and DEGs as observed by Li et al. and Tong et al. (PMID: 28680106, PMID: 33087762). We also observed the influence of three different alignment tools on the accuracy of gene quantification and DEG identification; for example, Subread showed low accuracy when combined with StringTie.

In the original manuscript, we described that the impact of different alignment tools on accuracy was relatively smaller compared to the gene annotation or gene quantification tools, based on the smaller RMSE or closer correlation coefficients between alignment tools, especially in relative gene expression levels. According to your suggestion, to avoid any potential confusion, we have revised the corresponding content in the Table 1, Results, and Discussion sections as follows:

“The three different alignment tools demonstrated similar accuracy, especially when using Ensembl annotation and quantification tools such as featureCounts and HTSeq. When using StringTie for quantification, STAR and HISAT2 outperformed Subread (Supplementary Fig. 47).” (Results, lines 537–540)

“Second, for alignment tools, STAR with Ensembl annotation resulted in the highest mapping rate. Similar to previous studies, the three alignment tools generally exhibited similar accuracy in gene quantification or DEG identification^{20, 50}. Specific cases have also been observed, such as lower accuracy with Subread when using the StringTie for gene quantification. The performance of alignment tools also varied depending on factors including the genomic complexity of different sample types and sequencing data characteristics, which should be considered when selecting alignment tools^{20, 51}. Specifically, differences in efficiency in aligning reads to ambiguous genes, such as pseudogenes, significantly impact the accuracy of DEGs^{52, 53}.” (Discussion, lines 705–714)

“The selection of alignment tools should consider factors such as the genome complexity of samples and characteristics of sequencing data⁵⁰⁻⁵²” (Table 1)

10- How table 1 is related to this statement “The experimental design is generally considered to be centered around addressing the biological questions of interest (Table 1) lines 555 and 556?”

Reply: Thank you for carefully point this out.

In this paragraph, we primarily aim to discuss the considerations when designing RNA experiments, which include the development of quality control to identify low-quality experimental execution and the selection of suitable experimental protocols. Low-quality of experimental execution could result in lower accuracy of gene quantification and DEG identification, highlighting the importance of comprehensive quality control in experimental design to identify them. The different experimental protocols should be chosen based on an understanding of the specific influences of each experimental method on RNA-seq performance, and should consider the research objectives (refers

to “the biological questions of interest” in the original manuscript). To convey this clearer and echo the content of **Table 1**, we have modified the text as follows:

“The experimental design centered on establishing strict quality control to avoid the influence of low-quality experimental execution, as well as choosing experimental protocols for specific research objectives, considering their impact on RNA-seq performance (**Table 1**).” (Discussion, lines 657–660)

11- In line 591 it is mentioned that a threshold chosen based on the maximum number of DEGs is practical. How this threshold is chosen?

Reply: Thank you for your question.

As mentioned in the Materials and Methods section, low-expression genes were filtered using a series of thresholds ranging from low to high, followed by differential expression analysis. If benchmark datasets are available, it is possible to balance sensitivity and specificity by selecting the optimal threshold. Otherwise, opting for the threshold corresponding to the maximum number of DEGs can be practical (as shown in Supplementary Figure 51 in the revised manuscript), as it results in the highest sensitivity and acceptable specificity in DEG detection. We have modified the text to improve clarity as follows:

“If benchmark datasets are available, the optimal thresholds could be chosen by balancing sensitivity and specificity after applying a series of thresholds ranging from low to high to filter low-expression genes. Otherwise, it is practical to achieve optimal sensitivity and acceptable specificity of DEG identification by choosing the threshold corresponding to the highest number of DEGs.” (Results, lines 716–721)

12- How correlation coefficients for assessing gene expression accuracy is calculated in line 607?

Reply: Thank you for your question.

The Pearson correlation coefficient between RNA-seq and ground truth was used for assessing the accuracy of gene expression measurements. We have modified “correlation coefficient” to “Pearson correlation coefficient” in the original line 607. To

clarify the calculation of correlation coefficients, we added the following content to the Materials and Methods section:

“The Pearson correlation coefficient was used to evaluate the accuracy of gene expression. This involved examining the correlation between \log_2 FPKM and ERCC concentrations as well as TaqMan gene quantification results, for assessing absolute gene expression, and examining the correlation between sample-pair relative expression and gene fold change in the TaqMan datasets and the Quartet reference datasets, for assessing relative gene expression.” (Materials and Methods, lines 1281–1286)

13- in the supplementary material and notes section 2.4 it was mentioned that only the genes that have at least one read in all replicates are considered. What alignment tool did you use for this part? what is the logic/ justification behind ignoring the genes that don't have a read mapped to that gene in all replicates?

Reply: Thank you for your valuable comments and questions.

1) What alignment tool did you use for this part?

In Supplementary Notes Section 2.4, we calculated the number of expressed genes using count data submitted by the 45 laboratories. Thus, alignment tools included STAR and HISAT2 among laboratories, as depicted in Supplementary Table 1. We clarified this point in the Supplementary Notes Section 2.4 (now labeled as Section 2.3), and added annotations for alignment tools for each laboratory in **Supplementary Figure 62** as follows:

“Based on gene count data from 45 laboratories, each employing their own analysis pipeline, we compared the number of detected or expressed genes across all laboratories after applying different thresholds.” (Supplementary Materials, lines 605–607)

2) what is the logic/justification behind ignoring the genes that don't have a read mapped to that gene in all replicates?

We note that the setting of thresholds for the identification of expressed genes is typically empirical. As you correctly mentioned, using a threshold requiring at least one read in each of the three technical replicates indeed excludes some expressed genes that may have reads present in only one or two out of the three technical replicates. Therefore, we have applied three types of thresholds previously utilized by Yu et al. (PMID: 37679545) and the MAQC-III Consortium (PMID: 25150838), to determine the number of detected genes across all laboratories. These thresholds include at least one read in any one replicate to identify expressed genes, at least 10 reads in any one technical replicate, and at least 3 reads in at least two out of three replicates. When employing these different thresholds, the number of detected genes demonstrates similar trends across laboratories. Thus, we have added a description to clarify this in the Supplementary Notes Section 2.4 (now labeled as Section 2.3), and modified **Supplementary Figure 62** as follows:

“Three types of thresholds previously used by Yu et al.¹⁹ and the MAQC-III Consortium²⁰ were incorporated, including with at least one read in any one technical replicate, with at least 10 reads in any one technical replicate, and with at least 3 reads in at least two out of three replicates” (Supplementary Materials, line 607–610)

Supplementary Figure 56 now is labeled as Supplementary Figure 62.

Supplementary Figure 62. The number of genes detected by all laboratories. Genes are considered detectable (expressed) in a sample group if there is (a) at least one read in any one technical replicate, (b) at least 10 reads in any one technical replicate, and (c) more than three reads in at least two out of the three replicates. Three laboratories exclusively focused on protein-coding genes for downstream analysis, five laboratories excluded pseudogenes from their analysis, while the remaining 37 laboratories analyzed genes of all biotypes. (d) The number of sequencing reads across laboratories. (e) The number of sequencing reads mapped to the exonic regions across laboratories. Higher sequencing depth generally leads to the detection of more genes, although exceptions exist, such as when alignment rates are low or duplication rates are high.

14- Figure 2a- labs are not labeled.

Reply: Thank you for carefully pointing out this.

We have labeled labs in Figure 2a as follows:

Fig. 2 RNA-seq performance metrics for real-world laboratories

(a) SNR values to measure the quality of expression data submitted from 45 laboratories. Laboratories were ordered by SNR values. Dots represented SNR values based on any 17 of the 18 samples (12 Quartet and 6 mixed samples). A dot in dark red represented SNR17 outlier that increased over six decibels compared to its standard SNR (18-sample SNR) after excluding one low-quality library in a laboratory, while a dot in orange represented an acceptable SNR17 value that decreased or increased less six decibels compared to its standard SNR. The red dashed line represents the cutoff of SNR at 12. (b) The gene types of interest for all laboratories and the corresponding

number of genes supported by at least one read in any one technical replicate (**Supplementary Notes, Section 2.3**). Three laboratories analyzed only protein-coding genes, five laboratories excluded pseudogenes from their analysis, and the remaining 37 laboratories analyzed all gene types. **(c)** Comparison of absolute expression levels to the TaqMan datasets and ERCC concentrations on the \log_2 scale. **(d)** Scatterplots of PCA on RNA-seq data of all laboratories in absolute expression levels, **(e)** and relative expression levels. The circles of the same color represent all replicates across all laboratories for each sample. **(f)** Assessment of relative expression levels using Pearson correlation coefficient and the Root Mean Square Error (RMSE) based on the Quartet reference datasets and the TaqMan datasets on the \log_2 scale. **(g)** ERCC spike-in ratios can be recovered increasingly well at higher expression levels. **(h)** A consistency test for recovering the expected sample mixing ratio in samples T1 and T2. The red and cyan solid line traces the expected curve after mRNA/total-RNA shift correction. The grey dashed lines indicate the fitted curves from data of laboratories. The ERCC genes are shown in black, and the other human genes are shown in grey. **(i)** The ability to recover expected mixing ratios was measured using RMSE between the observed expression profiles and the expected expression profiles. As genes with low fold changes were progressively filtered out, the RMSE across all laboratories decreased, indicating an increase of accuracy. The different colors in the box plots represent varying percentage of filtered genes. **(j)** Comparison of differentially expressed genes to the Quartet reference datasets and the TaqMan datasets using Matthews Correlation Coefficients (MCC).

15- In Supplementary Figure 17 gene types should be labeled

In a few other plots, the labs are not labeled. Please double-check.

Typo: TPE -> TPR

Supplementary Figure 45. Comparison of the optimal threshold values determined by maximum number of DEGs and highest TPR. For all RNA-seq from laboratories, the threshold values corresponding to the highest TPE and the maximum number of DEGs were calculated and compared for Quartet (up) and MAQC (down) samples. TPR, true positive rate.

Typo: accuracy -> accurate

Supplementary Figure 20. Relative expression measurements were more accuracy than absolute expression measurements. Based on TaqMan datasets for both Quartet and

MAQC samples, relative expression consistently exhibited Pearson higher correlation coefficients when compared to absolute expression.

Reply: Thank you very much for carefully pointing out these errors.

1) In Supplementary Figure 17 gene types should be labeled

We have labeled the gene biotypes in **Supplementary Figure 17** (now labeled as **Supplementary Figure 3**) as follows:

Supplementary Figure 3. The gene lengths and expression levels for five gene types.

Five gene types include protein-coding genes, long non-coding RNA (lncRNA), pseudogenes, small non-coding RNA (sncRNA), and others (such as immunoglobulin/T-cell receptor gene segments). (a) Pseudogenes, sncRNA, and immunoglobulin/T-cell receptor gene segments are generally shorter in length, corresponding to lower inter-laboratory reproducibility (Supplementary Figure 2). (b) In both the Quartet and MAQC samples, the average gene expression from all RNA-seq data from 42 laboratories is calculated. The lncRNA, sncRNA, immunoglobulin/T-cell receptor gene segments, and pseudogene exhibit lower expression levels compared to protein-coding genes, which could also explain the lower inter-laboratory reproducibility for these gene types (Supplementary Figure 2).

2) In a few other plots, the labs are not labeled. Please double-check.

We have carefully checked all figures to ensure that the labs are properly labeled. For example, we have labeled **Supplementary Figure 40** (now labeled as **Supplementary Figure 50**) in the original manuscript as follows:

Supplementary Figure 50. Distribution of gene expression using different normalization methods. The gene expression was quantified using the Ensembl-STAR-StringTie pipeline. Each normalization method was compared across 13 benchmark datasets for the Quartet and MAQC samples.

3) Typo: TPE -> TPR

Supplementary Figure 45. Comparison of the optimal threshold values determined by maximum number of DEGs and highest TPR. For all RNA-seq from laboratories, the threshold values corresponding to the highest TPE and the maximum number of DEGs were calculated and compared for Quartet (up) and MAQC (down) samples. TPR, true positive rate.

Typo: accuracy -> accurate

Supplementary Figure 20. Relative expression measurements were more accuracy than absolute expression measurements. Based on TaqMan datasets for both Quartet and MAQC samples, relative expression consistently exhibited Pearson higher correlation coefficients when compared to absolute expression.

We have corrected these typos and thoroughly checked throughout the manuscript.

“Supplementary Figure 55. Comparison of the optimal threshold values determined by the maximum number of DEGs and highest TPR. For all RNA-seq from laboratories, the threshold values corresponding to the highest TPR and the maximum number of DEGs were calculated and compared for Quartet (up) and MAQC (down) samples. TPR, true positive rate.” (Supplementary Materials, lines 523–527)

“Supplementary Figure 6. Relative expression measurements were more accurate than absolute expression measurements. Based on the TaqMan datasets for both Quartet and MAQC samples, relative expression consistently exhibited higher Pearson correlation coefficients when compared to absolute expression.” (Supplementary Materials, lines 148–151)

Reviewer #3 (Remarks to the Author):

Wang et al. undertook a comprehensive analysis of over 1000 RNA-seq datasets from 45 independent laboratories to explore the variability and noise inherent in transcriptome analysis via RNA-seq. Their study aimed to establish best practices for experimental design and analysis pipelines, mainly focusing on identifying differentially expressed genes and quantifying the absolute and relative expression levels of transcripts. Unlike the MAQC dataset, which is characterized by significant gene expression variances among samples, Wang et al. additionally utilized the Quartet dataset, notable for its minimal gene expression differences. This addition allows their methodological evaluations to mirror typical biological research conditions better. By leveraging the Quartet and MAQC datasets, the authors thoroughly evaluated various quantitative indicators across multiple methods and under various conditions. Their work provides essential insights for designing and analyzing future transcriptome studies, reflecting a more realistic and applicable approach to the challenges of RNA-seq analysis.

Despite the significant insights offered by the analyses, the study scheme and the composition of datasets have inherent limitations that necessitate careful interpretation of the findings. These limitations must be clearly communicated to ensure the readers can accurately grasp the insights provided before the study's publication. In particular, the following major points will require careful consideration.

Reply: We are grateful for the reviewer's positive assessment on the value of our work that "provides essential insights for designing and analyzing future transcriptome studies, reflecting a more realistic and applicable approach to the challenges of RNA-seq analysis." We greatly appreciate the reviewer for the insightful and constructive comments and suggestions aimed to help us improve our manuscript. In the following pages, we tried our best to respond to each of the reviewer's comments and suggestions, particularly those on the "inherent limitations that necessitate careful interpretation of the findings." We sincerely hope that the reviewer will find our response thorough and acceptable.

1. The manuscript extensively uses SNR calculations through PCA as a primary quantitative metric for library quality assessment. However, an issue arises, as depicted in Figure 2A, where SNR values are notably higher for low-quality outliers, considered random failures. Although the utilization of PCA-based SNR is discussed in the original paper on the Quartet dataset, I believe that relying on PCA-based SNR as a universal quality metric is still premature without accounting for additional factors that could affect its monotonicity with actual library quality.

Reply: Thank you very much for your valuable comments.

We apologize that we have not clearly stated what Figure 2a means. We did not mean that “SNR values are notably higher **for** low-quality outliers,” as the reviewer appeared to have interpreted. Instead, what we meant to say in Figure 2a is “SNR values are notably higher **after excluding** low-quality outliers”. The notably high SNR values in Figure 2a indicate significantly increased SNR17 values (calculated using gene expression profiles of any 17 out of 18 samples) compared to SNR18 (calculated using gene expression profiles of all 18 samples). In other words, this means significantly increased SNR values after excluding the data from the low-quality sample due to random failures. To clarify the meaning of outliers in Figure 2a, we have added an explanation in the legend of **Figure 2** as follows:

“A dot in dark red represented an SNR17 outlier that increased over six decibels compared to its standard SNR (18-sample SNR) after excluding one low-quality library in a laboratory, while a dot in orange represented an acceptable SNR17 value that decreased or increased less than six decibels compared to its standard SNR.” (Figure 2, lines 823–827)

As you mentioned, the monotonicity between SNR and library quality may be influenced by other factors. Thus, for comprehensive quality control, we also used some basic quality control metrics that could reflect library and sequencing quality. For example, in addition to SNR, we utilized pre-alignment metrics such as base quality, duplication rate, and sequencing depth, and post-alignment metrics including mapping rate, 5'-3' bias, and the reads mapped to intergenic regions. Furthermore, we performed sample-level QC checks, including a single nucleotide polymorphism (SNP) based sample identity check to identify sample swaps or mislabeling, and an examination of

the number of reads mapped to ERCC genes in samples without ERCC mixtures to identify cross-contamination. We have revised **Supplementary Table 3** and added **Figure 3** to display the quality control outcomes for RNA-seq data from all laboratories using these metrics.

Fig. 3 Quality flags of RNA-seq data

Multiple quality metrics were used, including the number of sequencing reads, base quality (Q30), the percentage of reads mapped to the human genome (Mapping rate), gene body bias (5'-3' bias), the percentage of mapped reads that were located in the intergenic region (Intergenic region), SNR based on the MAQC and Quartet samples, the difference between the SNR calculated from any 17 out of 18 samples and the SNR calculated from all 18 samples (SNR17-SNR18), the percentage of reads mapped to ERCC sequences in samples without ERCC mixtures (Cross contamination), SNP-based sample-identity check, and final quality flag.

2. - The study leverages the MAQC and Quartet datasets throughout the manuscript, complicating these findings' generalization. The MAQC dataset demonstrates substantial between-sample variation, whereas the Quartet dataset exhibits minimal variation. Applying the study's conclusions to conventional RNA-seq analyses may be limited, especially those involving genetic perturbations or chemical treatments. This concern is particularly relevant to conclusions drawn about DEG analysis because the selection of intersample normalization strategies and statistical testing methods is influenced by the expected variance and specific characteristics of the samples under comparison.

Reply: Thank you very much for your constructive comments.

We fully agree with you that the reference materials and datasets used in a benchmark study determine the generalizability of research findings. Ideally, the sample characteristics should be considered when drawing conclusions. As reported in previous studies, the interpretation of RNA-seq performance may vary depending on the biological complexity of samples or treatment effect sizes, such as those with genetic perturbations or chemical treatments (PMID: 25150839). Particularly for DEG analysis, where each analysis tool uses a data distribution model, inter-sample normalization strategy, and statistical test, tailored to the expected variances in read counts or expression values, their efficiency could be potentially influenced by specific variances and characteristics of the samples (PMID: 20196867). To clarify the inherent limitations in our conclusions' generalizability associated with samples, we have added text to the Discussion section as follows:

“Limitations should also be noted, considering the influence of the biological complexity of samples on RNA-seq performance. The interpretation of our findings should fully consider the characteristics of the samples. This study utilized two sets of distinctly different reference materials, representing conditions of small (Quartet) and large (MAQC) biological differences, which may not fully reflect more complex biological experiments, such as those involving genetic perturbations or chemical treatments^{57,58}. Previous studies have reported that the RNA-seq performance may vary depending on the biological complexity of samples⁵⁷. This concern is particularly relevant in DEG analysis, where each analysis tool uses a distinct data distribution model, inter-sample normalization strategy, and statistical test, tailored to the expected variances in read counts or expression values. Consequently, the performance of the tools could be potentially influenced by specific variances and characteristics of the samples⁵⁹.” (Discussion, lines 757–769)

As the reviewer correctly pointed out, this study strategically selected the two sets of widely used reference materials, the Quartet and MAQC samples, to represent different experimental conditions with small and large biological differences, respectively. These reference materials could reflect the fundamental analytical performance of RNA-seq. This means that if technical noises are difficult to avoid in the analysis of these reference materials, it becomes significantly more challenging to distinguish true biological differences in samples with high biological complexity, such as those involving genetic

perturbations or chemical treatments. In other words, our study acts as a “gatekeeping test” for delving into deeper biological explorations, emphasizing the necessity of achieving high levels of analytical performance with these reference materials as a foundational benchmark before embarking on more complex biological experiments. Additionally, despite the context-specific nature of the Quartet and MAQC samples, we also revealed some similar conclusions with previous studies using other types of samples, for example, the impact of different differential expression analysis tools (PMID: 24300110). Therefore, our insights obtained from the Quartet and MAQC samples can be used to enhance the quality and interpretation of a variety of RNA-seq experiments. To clarify the significance of providing a foundation for RNA-seq performance using the two types of reference materials, we have added contents to the Discussion section as follows:

“Despite this, our findings also exhibited similarities with previous studies that used different samples, such as the assessment of different DEG analysis tools⁶⁰. Additionally, the two sets of fundamental RNA reference materials allow for the establishment of a robust foundation of RNA-seq performance across laboratories before embarking on more complex biological experiments. This means that if technical noises cannot be distinguished even in analyzing these reference materials, detecting true biological differences in samples with higher biological complexity or small treatment effects will be even more challenging. We underscored the critical role of comprehensive quality control, meticulous experimental design, and the selection of bioinformatics tools, and these insights obtained from the Quartet and MAQC samples are expected to enhance the quality and interpretation of a variety of RNA-seq experiments.” (Discussion, lines 769–780)

3. - Figure 2a shows that a concerning number of participating laboratories generated low-quality data. In general research situations, low-quality data are rarer than this, thanks to outsourcing robotics-enabled sequencing centers or professional sequencing cores, and the service accompanies routine quality checking. However, the set of low-quality data persists throughout the analysis and appears in multiple analyses that could be more realistic with pre-filtering low-quality data. Particularly for Quartet, where the biological differences in the data are inherently minimal, data quality becomes even more critical. A more constructive approach would be to demonstrate the variability of

library quality and related factors first, then proceed to the rest of the analyses with low-quality data removed.

Reply: Thank you very much for your valuable comments and suggestions.

As you mentioned, we observed that among the expression data submitted by the 45 laboratories, 17 had SNR values less than 12. Even after employing a fixed analysis pipeline, the expression data from 10 laboratories still had SNR values below 12. Considering other quality metrics simultaneously, RNA-seq data from 16 laboratories were flagged as low quality, which seems more common than previously reported by the MAQC-III (SEQC) study (PMID: 25150838). This is mainly due to the fact that in our study individual laboratories had the freedom to employ their distinct experiment and data analysis workflows, particularly in manually preparing all libraries, reflecting real-world scenarios, which are different from data generated by outsourcing, robotics-enabled sequencing centers or professional sequencing cores.

In the original manuscript, we indeed included RNA-seq data from all laboratories in certain analyses, such as the investigation of sources of variation in gene expression. We fully agree with your suggestion of pre-filtering low-quality data to ensure more reliable conclusions. According to your suggestion, we have reorganized the order of the Results section and clarified that low-quality data were excluded from subsequent analyses.

First, in the section titled “Significant variations in detecting subtle differential expression”, we presented the results of a comprehensive assessment of RNA-seq performance in terms of data quality, the accuracy of gene expression and DEG identification. These results revealed the significant variations among laboratories in real-world scenarios. We have moved this section to follow the Study Design section in the Results.

Subsequently, in the section titled “Quality control check for filtering of low-quality RNA-seq data”, we performed QC checks at pre-alignment, post-alignment, and sample levels for all RNA-seq data. RNA-seq data from 16 laboratories were flagged as low quality, and excluded from subsequent analysis. The added contents are as follows:

“Using the above multiple metrics, including pre-alignment, post-alignment, and sample-level quality metrics, RNA-seq data from 26 laboratories passed all criteria and

were used for subsequent analysis, whereas the other data from remaining 16 laboratories were flagged as low quality and excluded from subsequent analysis to mitigate the impacts of poorer library or sequencing quality, as well as sample cross-contamination (**Supplementary Table 3**) (**Fig. 3**)” (Results, lines 361–366)

Finally, in the section titled “Sources of variation from the experimental processes”, we included the remaining RNA-seq data from 26 laboratories to investigate sources of variation from experimental processes. Overall, we obtained similar conclusions to those previously observed, indicating that the primary sources of variation in absolute gene expression from experimental processes include mRNA enrichment methods, strandedness, library kits, read length, and RNA input. Variations caused by these factors exceed biological differences among Quartet samples but are smaller than those observed among MAQC samples. In relative expression levels, the inter-laboratory variations of gene expression significantly decreased. We have made the necessary modifications to the content as follows:

“Sources of variation from the experimental processes

The significant inter-laboratory variations necessitated the investigation of the sources. Here, we focused on the magnitude of variations across 26 laboratories in absolute or relative gene expression introduced by each RNA-seq process. To exclusively investigate variation from the experimental process, we employed a uniform data analysis pipeline for all RNA-seq raw data, involving fastp for data pre-processing, Ensembl gene annotation, STAR for read alignment, and StringTie for gene quantification. When compared to the original expression data, the variations in SNR and the accuracy of gene expression measurements decreased across laboratories, with a significant improvement observed in some laboratories (**Supplementary Fig. 28–29**). Similar results were observed when an alternative gene quantification pipeline (for example, RefSeq and Salmon) was used for gene quantification (**Supplementary Fig. 30**). These findings indicated that the fixed pipeline effectively eliminated variations introduced by various bioinformatics processes employed by the laboratories. The variations arising from different RNA processing methods, library preparation protocols, and sequencing platforms among laboratories represent ‘experimental noise’.

The inter-laboratory variations in absolute gene expression levels introduced by experimental processes were significant in both the Quartet (**Fig. 4a**) and MAQC samples (**Fig. 4b**), especially impacting subtle differential expression measurements. We quantified the relative contribution of technical and biological factors to the total variations by principal variance component analysis (PVCA) based on absolute expression data from all laboratories for all samples. A total of 15 factors from the experimental process were considered (**Supplementary Table 4**), which introduced significantly greater variations than biological differences among the Quartet samples (85.1% vs. 5.8%), with mRNA enrichment methods and strandedness as the primary sources (**Fig. 4c**). Additionally, other factors, including library preparation kits, read lengths, the number of exonic reads, RNA inputs, and their interactions also contributed to more than 25% of the variations. These factors also corresponded to clustering patterns of gene expression consistency across laboratories (**Fig. 4f**). In contrast, while the MAQC samples revealed similar sources, variations from experimental factors were lower than biological differences between the MAQC samples (38.2% vs. 61.2%) (**Fig. 4d**). Employing the alternative analysis pipeline (RefSeq and Salmon) also revealed similar sources of variations, with these introduced variations representing approximately 15-fold and 0.4-fold of biological differences in the Quartet and MAQC samples, respectively (**Supplementary Fig. 31**).

In relative gene expression levels, the proportion of variation attributed to experimental factors decreased to below 20% for the Quartet and MAQC samples, respectively, which was observed when employing two different fixed analysis pipelines (**Supplementary Fig. 32**). This indicated that relative expression could effectively correct for the influence of experimental factors. The increased consistency of relative gene expression between any two laboratories compared to absolute expression further confirmed this (**Fig. 4e-f**).” (Results, lines 368–409)

Additionally, in the section titled “Best practices for experimental designs”, we assessed the impact of each experimental factor on RNA-seq performance using remaining RNA-seq data from 26 laboratories. Overall, we underscored the significant influence of the quality of experimental execution. We also elucidated the impact of mRNA enrichment methods and strandedness on the accuracy of absolute expression measurements, as well as the influence of sequencing depth, read length, and other

potential factors on the accuracy of relative expression and DEG identification. We have revised the content in the Results section as follows:

“Best practices for experimental designs

The low quality of experimental execution significantly influences the RNA-seq performance. RNA-seq data from 16 laboratories, failing multiple quality metrics, were considered low experimental quality in library preparation or sequencing processes (**Fig. 3**). Based on four types of ground truth, these laboratories exhibited lower Pearson correlation coefficients or higher RMSE for absolute and relative gene expression measurements, as well as lower MCC for DEG identification, compared to the other 26 laboratories (**Supplementary Fig. 35**). Noticeably, these instances of low quality were unrelated to the choice of experimental methods, as they were distributed across various experimental workflows. Therefore, our results highlighted the importance of multidimensional quality control in experimental design.

Different experimental protocols tended to influence RNA-seq performance, which should be considered during experimental design. Based on multiple types of ground truths, we assessed the influence of experimental factors using RNA-seq data from 26 laboratories in four aspects: data quality, absolute and relative gene expression, and DEG identification (**Fig. 6**). (i) For expression data quality, the Poly(A) selection method exhibited higher SNR values than the rRNA depletion method. Considering the differences in gene types captured by the two methods, we further examined SNR values for different gene types and observed that the Poly(A) selection method primarily exhibited higher SNR values for protein-coding genes. Conversely, for other gene types, particularly snRNA, the rRNA depletion method demonstrated significantly higher SNR, indicating a more accurate capture of biological differences in these RNAs (**Supplementary Fig. 36**). (ii) For absolute expression levels, the influences of the mRNA enrichment method and strandedness were observed, with the rRNA depletion method and stranded-specific libraries exhibiting higher correlation coefficients with the reference datasets. (iii) For relative gene expression and DEG levels, a higher number of reads mapped to exonic regions showed a strong or moderate correlation with improved accuracy, likely due to more reliable detection of lowly expressed genes³⁸. To validate their impacts, we down-sampled RNA-seq data to different depths and drew similar conclusions. Specifically, lower read depths resulted in decreased accuracy of relative gene expression and DEG identification, while

exerting a relatively minor overall impact on SNR and absolute expression levels. However, when sequencing depth decreased to extremely low levels, such as 10 Mb, SNR and accuracy of absolute expression detection also significantly declined (Supplementary Fig. 37). Furthermore, the MAQC TaqMan datasets revealed significant differences in accuracy associated with read lengths. To further explore the impact of read lengths, we trimmed sequencing reads to different lengths ranging from 25 bp to 150 bp across four laboratories, revealing that longer sequencing reads tended to exhibit higher accuracy, especially in relative gene expression measurements, and either similar or higher accuracy in DEG identification (Supplementary Fig. 38).” (Results, lines 440–479)

The added or modified figures are relabeled in the revised manuscript as follows:

Fig. 4 Sources of variation from the experimental processes

Scatterplots of PCA on RNA-seq data from 26 laboratories for the (a) Quartet samples, and (b) MAQC samples after applying the fixed analysis pipeline. The circles of the same color represent all replicates across all laboratories for each sample. Principal variance component analysis quantifies the proportion of variance explained by each experimental factor for the (c) Quartet samples, and (d) MAQC samples. Heatmap and hierarchical clustering of different laboratories based on the RMSE at (e) absolute expression levels, and (f) relative expression levels for the Quartet samples. RMSE, Root Mean Square Error.

Fig. 6 The influence of experimental factors based on different performance metrics

Performance metrics included SNR for data quality, Pearson correlation coefficient for accuracy of absolute and relative expression, RMSE for recovery of mixing ratios, and MCC for the accuracy of DEG identification. Exonic reads indicate the number of mapped reads located in the exonic regions. Exonic reads were considered as an experimental factor as they could be determined by the total sequencing depth and the mRNA enrichment method chosen. The impact of exonic reads on data quality and accuracy is evaluated using Spearman correlation analysis. Significance testing was conducted based on normal distribution assumptions using one-way analysis of

variance (ANOVA) and paired t-tests; or, in cases where normal distribution was not observed, independent samples were subjected to Kruskal-Wallis test and Mann-Whitney U test. ** indicates a p-value < 0.05. ns, not significant; SNR, Signal-to-Noise Ratio; RMSE, Root Mean Square Error; MCC, Matthews Correlation Coefficient; TN, True Negative; TP, True Positive; FN, False Negative; FP, False Positive.

Supplementary Figure 28. Data quality (SNR values) after applying fixed analysis pipelines. The two gene quantification pipelines (Ensembl-STAR-StringTie and RefSeq-Salmon) were applied for RNA-seq data from 26 laboratories. SNR based on Quartet and mixing samples was compared across expression data submitted from laboratories and from fixed quantification pipelines. The variations in SNR values across laboratories significantly decreased after using fixed pipelines, indicating efficient elimination of bioinformatics noises. More than half of the laboratories demonstrated an increased SNR after applying the uniformed analysis pipeline, especially for those with initially lower SNR values, indicating the failure of bioinformatics pipelines in these laboratories. However, some laboratories experienced a decrease in SNR values after applying the uniformed analysis pipeline. This observation may be related to the different number of genes considered, as the uniformed analysis pipeline encompassed all genes with non-zero read counts, whereas real laboratories included a smaller gene set due to their gene type preferences and the selection of gene annotation and analysis tools. SNR, signal-to-noise ratio.

Supplementary Figure 29. The accuracy of absolute and relative expression after applying the fixed analysis pipeline (Ensembl-STAR-StringTie). RNA-seq data from 26 laboratories that passed QC metrics were included. The accuracy between expression data submitted from laboratories and those obtained using Ensembl annotation, STAR for alignment, and StringTie for gene quantification, was compared. **(a)** The accuracy of absolute gene expression was assessed based on the TaqMan datasets for Quartet and MAQC samples, and ERCC concentrations. **(b)** The accuracy of relative gene expression was assessed based on the Quartet reference datasets, the TaqMan datasets for Quartet and MAQC samples, and built-in truth (ERCC spike-in ratios and mixing ratios in T1 and T2). After applying the analysis pipeline, the variations in accuracy metrics decreased, including Pearson correlation coefficients and RMSE. Some laboratories demonstrated improved accuracy when using the fixed pipeline. The red dots represent the metric calculated from the data submitted by laboratories, whereas the cyan dots indicate the metric after applying the fixed analysis pipeline. The circles indicate the Pearson correlation coefficient between laboratories and reference datasets, and the diamonds indicate the Root Mean Square Error (RMSE).

Supplementary Figure 30. The accuracy of absolute and relative expression after applying the fixed analysis pipeline (RefSeq-Salmon). RNA-seq data from 26 laboratories that passed QC metrics were included. The accuracy between expression data submitted from laboratories and those obtained using RefSeq annotation and Salmon for gene quantification, was compared. **(a)** The accuracy of absolute gene expression was assessed based on the TaqMan datasets for Quartet and MAQC samples, and ERCC concentrations. **(b)** The accuracy of relative gene expression was assessed based on the Quartet reference datasets, the TaqMan datasets for Quartet and MAQC samples, and built-in truth (ERCC spike-in ratios and mixing ratios in T1 and T2). Applying RefSeq-Salmon across laboratories also led to decreased variations in accuracy, indicating that the variations introduced from different bioinformatics pipelines were effectively mitigated.

Supplementary Figure 31. PVCA of variations in absolute and relative based on different pipeline (RefSeq-Salmon). Principal variance component analysis quantifies the proportions of variance explained by each experimental factor in the (a) absolute expression levels and (b) relative expression levels for the Quartet samples. The proportions of variance explained by each experimental factor in the (c) absolute expression levels and (d) relative expression levels for the MAQC samples.

Supplementary Figure 32. Calculation of relative expression could eliminate the variations from experimental process. Scatterplots of PCA on RNA-seq data of 26 laboratories that passed QC metrics for **(a)** the Quartet samples and **(b)** the MAQC samples at relative expression levels. Principal variance component analysis quantifies the proportion of variance explained by each experimental factor for **(c)** the Quartet samples and **(d)** the MAQC samples in relative expression levels. The red bars represent biological differences between samples. When calculating relative expression, the relative contribution of each experimental factor to total variations decreased compared to the biological difference across the Quartet (up) or MAQC (down) samples. SNR, signal-to-noise ratio. "Residual" represents unexplained variations.

Supplementary Figure 35. Low accuracy for low-quality RNA-seq data. Based on four types of ground truth, RNA-seq data from 16 laboratories flagged as low quality exhibited low accuracy for absolute, relative gene expression, and DEG measurements, compared to the remaining RNA-seq data from 26 laboratories.

Supplementary Figure 36. Comparisons of SNR between two mRNA enrichment methods for different gene types. Using RNA-seq data from 26 laboratories passing quality metrics, the SNR values were calculated for five gene types, including protein-coding genes, small non-coding RNA (snRNA), large non-coding RNA (snRNA), pseudogenes, and others (Immunoglobulin/T-cell receptor gene segments). The SNR for the two mRNA enrichment methods (Poly(A) selection and rRNA depletion) was compared, and the significance of the difference was tested using unpaired t-test. ** indicates a *p*-value < 0.01. ns, not significant.

Supplementary Figure 37. The influence of read depths on RNA-seq performance. RNA-seq from lab28 was down-sampled into different depths (the number of fragments), including 150 Mb, 100 Mb, 100 Mb, 50 Mb, 30 Mb, and 10 Mb using seqtk (v.1.4)⁹. Then, genes were quantified using Ensembl annotation, STAR for alignment, and StringTie for gene quantification. Based on the four types of ground truth, the RNA-seq performance was assessed in multiple aspects: SNR, absolute and relative gene expression, and differentially expressed genes. MCC, Matthews Correlation Coefficient. RMSE, Root Mean Square Error.

Supplementary Figure 38. The impact of read length on the RNA-seq performance. RNA-seq data with read length of 150 bp from four laboratories were shortened into 100 bp, 75 bp, 50 bp, and 25 bp using Cutadapt (v.4.8)¹⁰. Based on four types of ground truth, the accuracy of absolute, relative gene expression, and identification of differentially expressed genes (DEGs) for different read lengths were examined.

4. - Regarding the presentation of analyses and results, while the authors' rich display of relatively unprocessed forms of evaluation results, many figures do not convey the point related to the context referred to in the main text. Instead, they look more like figures directly out of the quality check report. I think the publication should become more accessible to readers if, at least, the main figures clearly show the points of the context.

Reply: Thank you very much for the constructive suggestion.

According to your suggestion to make figures more accessible to the readers, we have checked all the figures and made necessary modifications to ensure that they clearly show the points of the context. The modified figures are as follows:

1) We modified the main **Figure 6** in the original manuscript (now labeled as **Figure 7**) to display the impact of each bioinformatics step on the accuracy of gene expression measurements, which shows the points of lines 528-550 in the revised manuscript.

Fig. 7 Accuracy of gene quantification pipelines

(a) Clustering of 28 quantification pipelines based on accuracy metrics (RMSE) derived from comparisons to four types of ground truth. **(b)** The color legend for c-g. The accuracy of all 28 quantification pipelines was assessed based on the **(c)** Quartet reference datasets, **(d)** Quartet TaqMan datasets, **(e)** MAQC TaqMan datasets, **(f)** built-in truth of mixing ratios in T1 and T2, and **(g)** built-in truth of ERCC spike-in ratios. The accuracy of quantification tools is compared under the same conditions of gene annotation and alignment tools. Box plots represent 13 benchmarked datasets.

2) We modified the main **Figure 7** in the original manuscript (now labeled as **Figure 8**) to display the impact of each bioinformatics step on the accuracy of DEG identification, which shows the points of lines 583-604 in the revised manuscript.

Fig. 8 Impact of each bioinformatics step on the accuracy of DEG identification

Using the benchmark datasets from 13 laboratories, the Matthews Correlation Coefficients (MCC) for 140 bioinformatics pipelines were measured based on the the

Quartet reference datasets, the Quartet TaqMan datasets, and the MAQC TaqMan datasets. The impact of each bioinformatics step on the accuracy of DEGs was assessed, including (a) different annotations, (b) different alignment tools, (c) different gene quantification tools, and (d) different DEG identification tools. The plot indicates average performance for 13 benchmark datasets when different annotations, alignment tools, quantification tools, and DEG identification tools are used. (e) The AUC was analyzed for five DEG identification tools using RNA-seq data from lab01, and the AUC values for other high-quality benchmark datasets are displayed in **Supplementary Fig. 57**. AUC, Area Under the receiver operating characteristic Curve. MCC, Matthews Correlation Coefficients.

3) We modified the **Supplementary Figure 29** in the original manuscript (now labeled as **Supplementary Figure 35**) to clearly show the influence of low-quality experiments on the RNA-seq performance, which demonstrates the influence of different alignment tools, corresponding to the points of lines 444-450 in the revised manuscript. The modified figure is as shown above in our response to your comment 3.

4) We moved the **Figure 6** in the original manuscript (now labeled as **Supplementary Figure 42**) into the Supplementary Materials, which demonstrates the influence of different alignment tools, corresponding to the points of lines 504-520 in the revised manuscript.

Supplementary Figure 42. Performance of different alignment schemes. (a) Distribution of mapping status of sequenced reads for six combinations of annotation and alignment tools. The 13 benchmark datasets corresponding to each sample are arranged in descending order based on the uniquely mapping rate. **(b)** Distribution of the number of reads with mismatch bases.

5) We modified the **Supplementary Figure 38** in original manuscript (now labeled as **Supplementary Figure 48**) to better display the performance ranking of 28 quantification pipelines, which corresponds to the points of lines 546-550 in the revised manuscript.

	Overall Rank	Rank	mean ± sd	Rank	mean ± sd	Rank	mean ± sd	Rank	mean ± sd	Rank	mean ± sd	RMSE
RefSeq_HISAT2_StringTie	1	9	0.314 ± 0.041	10	0.883 ± 0.106	2	1.137 ± 0.066	2	0.421 ± 0.084	2	0.825 ± 0.184	RefSeq_STAR_StringTie	2	10	0.315 ± 0.044	8	0.881 ± 0.113	3	1.139 ± 0.065	1	0.42 ± 0.085	4	0.84 ± 0.193	
Ensembl_Subread_featureCounts	3	1	0.289 ± 0.032	1	0.834 ± 0.087	8	1.159 ± 0.067	10	0.488 ± 0.097	9	0.901 ± 0.201	
Ensembl_STAR_featureCounts	4	2	0.294 ± 0.032	5	0.873 ± 0.101	9	1.163 ± 0.066	6	0.484 ± 0.099	8	0.898 ± 0.203	
Ensembl_HISAT2_StringTie	5	8	0.308 ± 0.041	14	0.905 ± 0.111	5	1.157 ± 0.066	3	0.428 ± 0.087	1	0.823 ± 0.183	
Ensembl_STAR_StringTie	6	11	0.316 ± 0.042	13	0.905 ± 0.109	6	1.159 ± 0.066	4	0.43 ± 0.085	3	0.839 ± 0.192	
Ensembl_STAR_HTSSeq	7	5	0.302 ± 0.033	3	0.873 ± 0.1	13	1.169 ± 0.066	7	0.485 ± 0.098	11	0.902 ± 0.204	
Ensembl_HISAT2_featureCounts	8	3	0.296 ± 0.033	7	0.877 ± 0.1	12	1.167 ± 0.068	12	0.49 ± 0.101	10	0.902 ± 0.2	
Ensembl_STAR_STAR	9	6	0.302 ± 0.033	4	0.873 ± 0.1	14	1.169 ± 0.066	8	0.485 ± 0.098	12	0.902 ± 0.204	
Ensembl_HISAT2_HTSSeq	10	7	0.304 ± 0.034	6	0.877 ± 0.099	15	1.17 ± 0.067	13	0.491 ± 0.1	7	0.897 ± 0.202	
Ensembl_Subread_HTSSeq	11	4	0.3 ± 0.033	2	0.849 ± 0.094	11	1.167 ± 0.066	14	0.494 ± 0.096	18	0.912 ± 0.205	
RefSeq_HISAT2_StringTie	12	20	0.38 ± 0.048	9	0.882 ± 0.112	19	1.19 ± 0.072	5	0.463 ± 0.081	6	0.849 ± 0.192	
Ensembl_Subread_StringTie	13	23	0.393 ± 0.039	12	0.892 ± 0.099	21	1.207 ± 0.068	9	0.488 ± 0.081	5	0.844 ± 0.191	
RefSeq_Salmon	14	22	0.388 ± 0.058	17	0.932 ± 0.103	1	1.123 ± 0.048	15	0.495 ± 0.09	22	0.976 ± 0.219	
RefSeq_Subread_featureCounts	15	19	0.377 ± 0.044	19	0.935 ± 0.124	4	1.151 ± 0.083	22	0.534 ± 0.107	16	0.91 ± 0.204	
RefSeq_HISAT2_featureCounts	16	18	0.373 ± 0.044	21	0.936 ± 0.122	7	1.159 ± 0.079	21	0.531 ± 0.106	17	0.911 ± 0.202	
RefSeq_Subread_featureCounts	17	12	0.333 ± 0.041	18	0.933 ± 0.097	16	1.173 ± 0.08	19	0.526 ± 0.103	19	0.914 ± 0.204	
RefSeq_Subread_HTSSeq	18	14	0.343 ± 0.041	20	0.936 ± 0.092	10	1.166 ± 0.079	20	0.529 ± 0.104	20	0.926 ± 0.207	
RefSeq_STAR_featureCounts	19	16	0.365 ± 0.036	24	0.947 ± 0.107	23	1.214 ± 0.067	16	0.514 ± 0.104	13	0.907 ± 0.205	
RefSeq_STAR_HTSSeq	20	17	0.369 ± 0.035	22	0.946 ± 0.109	22	1.207 ± 0.069	18	0.516 ± 0.105	14	0.91 ± 0.206	
RefSeq_kallisto	21	15	0.359 ± 0.048	11	0.892 ± 0.099	17	1.175 ± 0.094	25	0.657 ± 0.106	26	1.008 ± 0.217	
RefSeq_STAR_STAR	22	21	0.385 ± 0.033	23	0.946 ± 0.109	26	1.255 ± 0.064	17	0.514 ± 0.105	15	0.91 ± 0.206	
RefSeq_Sailfish	23	13	0.341 ± 0.043	15	0.916 ± 0.102	24	1.223 ± 0.1	27	0.708 ± 0.104	24	0.983 ± 0.225	
RefSeq_Bowtie_RSEM	24	27	0.494 ± 0.064	16	0.92 ± 0.128	25	1.223 ± 0.068	11	0.489 ± 0.114	27	1.09 ± 0.276	
Ensembl_Salmon	25	28	0.524 ± 0.073	26	1.025 ± 0.122	18	1.185 ± 0.071	24	0.575 ± 0.102	23	0.979 ± 0.22	
Ensembl_Bowtie_RSEM	26	25	0.447 ± 0.078	25	1.003 ± 0.137	20	1.192 ± 0.079	23	0.55 ± 0.119	28	1.09 ± 0.275	
Ensembl_Sailfish	27	24	0.445 ± 0.046	28	1.046 ± 0.125	28	1.342 ± 0.118	28	0.736 ± 0.103	21	0.976 ± 0.221	
Ensembl_kallisto	28	26	0.459 ± 0.046	27	1.046 ± 0.112	27	1.269 ± 0.109	26	0.694 ± 0.106	25	0.989 ± 0.216	
Ground truth												
			Quartet Reference datasets		Quartet TaqMan datasets		MAQC TaqMan datasets		Mixing ratios in T1 and T2		ERCC spike-in Ratios	

Supplementary Figure 48. The performance ranking of 28 quantification pipelines at relative expression levels. The four types of ground truth, arranged from left to right, include the Quartet reference datasets, the TaqMan dataset for Quartet and MAQC samples, the built-in truth regarding the recovery of mixed ratios in T1 and T2, and the built-in truth of ERCC spike-in ratios. The average Root Mean Square Error (RMSE) between relative gene expression and the ground truth was assessed across 13 benchmark datasets. Rankings were determined in ascending order based on RMSE, and the final ranking was calculated as the average of rankings derived from the five truths.

6) We modified the **Supplementary Figure 39** in original manuscript (now labeled as **Supplementary Figure 49**) to better display the comparison of six normalization methods, which corresponds to the points of lines 551-559 in the revised manuscript.

Supplementary Figure 49. Comparison of the SNR of RNA-seq data from different normalization methods. Read counts from 28 quantification pipelines were normalized using six normalization methods. Box plots for each method indicate the SNR for 13 benchmark datasets from 28 pipelines.

5. To further enhance the clarity and robustness of the study, the following minor points would need to be addressed:

5.1 - Line 127: Please clarify whether the technical replicates conducted by independent laboratories were performed in parallel or serially. This distinction is important because serial replication can primarily reflect between-lab variance, whereas parallel replication encloses both between-lab variances and batch effects together.

Reply: Thank you for your suggestion for clarity.

Each laboratory independently determined how to test technical replicates for RNA samples prepared in the same batch. Fifteen laboratories assigned all libraries to different flowcells or lanes for sequencing, referred to as "parallelly", introducing both between-lab variance and batch effects. Other laboratories sequenced all libraries within the same lane, referred to as "serially", reflecting only between-lab variance. To clarify this, we have added the description in the Results section as follows:

“Overall, RNA-seq data from 45 different laboratories reflected the inter-laboratory variations. Fifteen laboratories assigned all libraries to different flowcells or lanes for sequencing, introducing batch effects in RNA-seq data, while other laboratories sequenced them within the same lane, thereby without batch effects.” (Results, lines 137–141)

5.2 - Line 147: Read counts serve as a more direct measure than total base output in RNA-seq analysis and thus offer a more explicit representation of the data.

Reply: Thank you for the suggestion.

As you pointed out, read counts serve as a more direct quality control metric, reflecting the data size. We have modified **Supplementary Figure 1** (now labeled as **Supplementary Figure 9**) to present read counts and deleted the information regarding the number of bases. We also added an explanation of sequencing depth in the Results section, specifically referring to the read counts, as follows:

“We first assessed the sequencing quality properties (pre-alignment) for the Quartet and MAQC samples, including sequencing depth, base quality, GC content, and duplication rate (**Supplementary Table 3**). **The number of reads** ranged from 39.4 Mb to 418.8 Mb for Quartet samples and from 40.9 Mb to 424.2 Mb for MAQC samples across laboratories (**Supplementary Fig. 9**).” (Results, lines 266–271)

Supplementary Figure 9. Read counts of RNA-seq data. For all laboratories, the total number of reads for Quartet (up) and MAQC (down) samples was displayed, where higher sequencing depth is associated with greater inter-sample variations.

5.3 - Line 161: This explanation must be clarified, possibly stemming from misinterpreting the Illumina FAQ. The FAQ refers to mismatches resulting from

random hexamer priming rather than decreases in quality scores. Such mismatches are misincorporated bases in the DNA templates, whereas the quality score is a prediction made by the basecaller, subject to refinement only through post-analysis quality score calibration. Since this drop in quality scores also occurs in DNA sequencing, reverse transcription theories cannot adequately explain it. The actual cause is likely insufficient signal correction for factors such as phasing and crosstalk during signal intensity quantification and basecalling.

Reply: Thanks for your insightful comments.

As you mentioned, random hexamer priming could introduce mismatches, as stated in the Illumina FAQ, but this does not mean low quality scores of bases. Thus, we agree with you that random hexamer priming during reverse transcription cannot explain the low-quality scores of the first few sequencing bases. Following your insight, we referred to previous researches, which suggest that the main causes are the lower signal intensities for the first cycle and insufficient correction of base callers for factors such as phasing and cross-talk in the initial bases. The low signal intensities are attributable to dimming and bleaching caused by longer handling times before imaging of the first cycle (PMID: 19682367). Since phasing correction is generally applied iteratively starting from the first cycle, the correction of quality scores for the first few bases may not be complete (PMID: 19549630). To clarify this, we have added an explanation in the Results section as follows:

“Most laboratories showed a biased quality score distribution in the first 1–10 bases for the Quartet and MAQC samples (**Supplementary Figs. 12–13**), which was attributed to lower signal intensities for the first sequencing cycle and insufficient correction for factors such as phasing and cross-talk in the initial bases during base calling^{25, 26}.”
(Results, lines 280–284)

5.4- Line 163: Several RNA-seq library preparation kits, including the Illumina TruSeq kit, sequence the reverse strand first. Therefore, the analysis of quality scores should be separately conducted for kits that sequence in forward-reverse and reverse-forward orders to account for this variation accurately.

Reply: Thank you for pointing this out.

In line 163, "forward reads" and "reverse reads" refer to the two reads generated in paired-end sequencing, where "**forward reads**" refer to the reads **firstly** extended from the Reads 1 adapter, and "**reverse reads**" refer to the reads **secondly** extended from the Reads 2 adapter in the sequencing reaction. Thus, to clarify this and avoid misleading them as "**forward strand**" and "**reverse strand**", we have added the explanations of "forward reads" and "reverse reads" in the Results section and **Supplementary Figures 4–5** (now labeled as **Supplementary Figures 12–13**).

Indeed, 20 laboratories employed stranded-specific libraries, all of which were "reversed stranded", meaning that the reverse strand is sequenced first. In contrast, other laboratories prepared non-strand-specific libraries, where half of the reads were sequenced from the forward strand and the other half from the reverse strand. Theoretically, regardless of whether a kit sequences in forward-reverse and reverse-forward orders, the quality of forward reads is higher because they are sequenced first. This is particularly common on the Illumina sequencing platform, which is attributed to decreased cluster density and increased potential for sequencing errors after bridge PCR, along with reduced DNA polymerase activity before sequencing the reverse reads. We have added the details of the sequencing platforms in **Supplementary Figures 12–13** in the revised manuscript.

The modified contents are as follows:

"We also observed that the quality scores of forward reads (the first sequenced reads) were generally higher than those of reverse reads (the later sequenced reads), particularly in laboratories using the Illumina sequencing platform. This was attributed to the decreased cluster size and higher number of errors due to more amplification steps and reduced DNA polymerase activity before sequencing the reverse reads²⁷." (Results, lines 284–289)

Supplementary Figure 12. Base quality values of representative sequencing reads for the Quartet samples. The forward reads refer to the reads firstly extended from the Reads 1 adapter, and the reverse reads refer to the reads secondly extended from the Reads 2 adapter in the sequencing reaction. The base quality distribution in the first 1–10 bases was biased in most laboratories. The quality of reverse reads was generally lower than that of forward reads in most laboratories. Different colors represent different samples, with squares and circles representing forward and reverse reads, respectively.

Supplementary Figure 13. Base quality values of representative sequencing reads for the MAQC samples. Different colors represent different samples, with squares and circles representing forward and reverse reads, respectively.

5.5 - Line 172: The analysis should consider the potential strong influence of sequencing depth on duplication rates. Furthermore, since duplicated reads from noncoding RNAs introduce difficulties in accurate detection, a more precise duplication rate estimation could be achieved by restricting the analysis to reads mapped to coding genes.

Reply: Thank you for your valuable suggestions.

1) - Line 172: The analysis should consider the potential strong influence of sequencing depth on duplication rates.

As reported in previous studies, the sequencing depth is associated with the duplication rate (PMID: 30001700). We have included a correlation analysis between sequencing depth and duplication rate, revealing a higher number of reads in samples with a higher duplication rate. Additionally, we also observed higher duplication rates for samples using the Poly(A) selection method compared to the rRNA depletion method, which has been reported in a previous study (PMID: 25150835). We have added a description in the Results section and **Supplementary Figure 18** as follows:

“The average duplication rates of the sequencing reads varied significantly across laboratories, ranging from 4.2% to 73.4% for the Quartet samples and from 5.0% to 75.5% for the MAQC samples (**Supplementary Fig. 17**), with increased duplication rates correlating with higher sequencing depth (**Supplementary Fig. 18a**). Additionally, the mRNA enrichment method also influenced the duplication rates, with the Poly(A) selection method demonstrating higher duplication rates, similar to observations in other studies (**Supplementary Fig. 18b**)¹⁵.” (Results, lines 298–304)

Supplementary Figure 18. The correlation between duplication rates and sequencing depth. The duplication rate of sequencing reads was estimated based on FASTQ files using fastp (v.0.23.2)¹. (a) The duplication rates were categorized into four subgroups: < 10%, 10% – 20%, 10% – 20%, and > 30%. The Quartet and MAQC samples with higher duplication rates showed increased sequencing depth. (b) For the Quartet and MAQC samples, samples using the Poly(A) selection method for mRNA

enrichment demonstrated higher duplication rates compared to samples using the rRNA depletion method.

2) Furthermore, since duplicated reads from noncoding RNAs introduce difficulties in accurate detection, a more precise duplication rate estimation could be achieved by restricting the analysis to reads mapped to coding genes.

Following your suggestion, we have estimated the duplication rate of reads mapped to coding genes using the Picard CollectInsertSizeMetrics function. After excluding reads mapped to non-coding RNAs, we observed that duplication rates of reads mapped to coding genes significantly decreased compared to those of reads mapped to the entire genome. This observation is particularly prominent in samples employing the rRNA depletion method for mRNA enrichment, primarily attributable to its capture of a substantial amount of non-coding RNA. These results reflected higher duplication rates of reads mapped to non-coding RNAs, indicating difficulties in detecting non-coding RNAs. We have supplemented these results in Supplementary Table 3. We have also included **Supplementary Figure 19** to present the results and added a description in the Results section as follows:

“Noticeably, the rRNA depletion method, due to capturing a large amount of non-coding RNA, showed a significant decrease in duplication rates after removing reads aligned to non-coding regions, indicating difficulties in detecting non-coding RNAs (**Supplementary Fig. 19**).” (Results, lines 304–307)

“Post alignment duplication rates in the BAM files were calculated using Picard (v.3.1.1) CollectInsertSizeMetrics function⁹⁰.” (Materials and Methods, lines 1418–1420)

Supplementary Figure 19. The duplication rate of reads mapped to coding genes.

The duplication rate of reads mapped to coding genes and all reads mapped to the genome was estimated based on BAM files using the Picard CollectInsertSizeMetrics function (v.3.1.1)⁵. For both Quartet and MAQC samples, the Poly(A) selection method exhibited higher duplication rates of reads mapped to coding genes compared to those of reads mapped to the genome. The significance was tested using paired t-test. *** indicates a *p*-value < 0.001. ns, not significant.

5.6 - Line 203: While the text identifies contamination during library preparation as a source of error, it should also acknowledge potential contamination within the sequencer. This includes contamination of barcodes between physically adjacent reads on the flow cell and carry-over contamination from previous samples through the sequencer's fluidic channels.

Reply: Thank you for pointing this out.

As you rightly pointed out, contamination could also occur within the sequencer. For example, previous studies reported that contamination can occur due to errors during sequencing leading to an alternate barcode sequence being read (PMID: 24507442, PMID: 32321923). Carry-over contamination is also a potential source of contamination, such as amplicons carried over from the prior steps (PMID: 37098913, PMID: 26152304). To clarify this, we have added an explanation of the sources of contamination in the Results section as follows:

“This indicates potential cross-contamination, possibly due to sample, library, or barcode contamination during library preparation³⁴, or misallocation of barcodes and

carry-over contamination from previous samples during the sequencing process^{35-37.}” (Results, lines 357–360)

5.7 - Line 218: Please consider the significant impact of read depth on SNR values. Incorporating read depth into the analysis is crucial for accurately assessing SNR values.

Reply: Thank you for your suggestion.

In the section titled “Best practices for experimental designs” of the original manuscript, we have conducted a correlation analysis between read depth (total number of reads in exonic regions) and SNR values, and observed a low Pearson correlation coefficient of -0.286. According to your suggestion, to further validate the correlation between them, we have down-sampled RNA-seq data with a high read depth of 210 Mb into 150 Mb, 100 Mb, 100 Mb, 50 Mb, 30 Mb, and 10 Mb using seqtk (v.1.4). Then, we used Ensembl annotation, STAR for alignment, and StringTie for gene quantification. Similarly, we also observed a consistent SNR across different read depths.

Since we have discussed the impacts of read depth on data quality (SNR) in the section titled "Best practices for experimental designs", to keep the coherence of the manuscript, we have added the description to the same section. Additionally, by down-sampling RNA-seq data, we also examined the impact of read depth on the accuracy of gene expression measurements and DEG identification. The modified contents are as follows:

“To validate their impacts, we down-sampled RNA-seq data to different depths and drew similar conclusions. Specifically, lower read depths resulted in decreased accuracy of relative gene expression and DEG, while exerting a relatively minor overall impact on SNR and absolute expression levels. However, when sequencing depth decreased to extremely low levels, such as 10 Mb, SNR and accuracy of absolute expression detection also significantly declined (**Supplementary Fig. 37**).” (Results, lines 468–474)

“The seqtk (v.1.4)⁹⁵ was employed for down-sampling RNA-seq data to various sequencing depths.” (Materials and Methods, lines 1462–1463)

5.8 - Line 299: It should be clarified why the thresholds and cutoffs are different. Please specify whether these differences arise from decisions made by individual labs, variations in the handling of each library during analysis, or are inherent to the unique characteristics of each library, such as read depth.

Reply: Thanks for pointing this out.

Due to the diversity of experimental designs and methodologies across laboratories conducting RNA-seq, the thresholds are typically determined by each laboratory to balance sensitivity and specificity based on their research objectives during RNA-seq development. Thus, differences in thresholds for filtering low-expression genes and DEG identification arose from decisions made by individual laboratories. To clarify this, we have added an explanation in the Results section as follows:

“Additionally, due to the diversity in their experimental designs and methodologies, different thresholds for filtering low-expression genes and DEG identification were chosen differently by the laboratories, which led to variations in the number of DEGs and impacted the accuracy.” (Results, lines 255–258)

To further illustrate the impacts of different thresholds on the accuracy of DEGs, we added the following content:

“For example, lab45 filtered genes with FPKM < 1 in all replicates before differential expression analysis, leading to true-positive genes present in the reference datasets being filtered out. Lab26 employed a stringent threshold of p-value < 0.001 while lab07 only utilized Q-value < 0.05 without incorporating log₂FC for DEG identification, which resulted in either few or excessive DEGs and consequently lower accuracy.” (Results, lines 258–263)

5.9 - Line 307: To strengthen the conclusions' validity, verifying whether analogous outcomes are observed when employing a distinct analysis pipeline is desirable. This cross-checking would ensure that the findings are independent of the specificities of the initially used pipeline.

Reply: Thank you for your comments and suggestions.

We agree that conclusions drawn from a single analysis pipeline may not be universally applicable. According to the reviewer's suggestion, we applied RefSeq annotation and Salmon (v.1.10.1) for cross-checking to ensure that the findings are independent of the specificities of the pipeline. Salmon was run in the mapping-based mode, using optional parameters: `--gcBias`, `--seqBias`, and `--posBias` to correct sequence-specific biases, GC biases, and 5' or 3' positional bias.

Similar conclusions were drawn when using the alternative gene quantification pipeline. First, after applying the RefSeq-Salmon pipeline, the variations in SNR and accuracy metrics of gene expression measurements among laboratories decreased, as observed when using the Ensembl-STAR-StringTie pipeline. Some laboratories demonstrated a significant improvement in data quality and accuracy. This indicated that both fixed analysis pipelines can effectively mitigate the variations introduced from different bioinformatics pipelines. We have included **Supplementary Figure 30** (as shown above in our response to your comment 3) to present the results from RefSeq-Salmon and described the conclusions in the Results section as follows:

“Similar results were observed when an alternative gene quantification pipeline (for example, RefSeq and Salmon) was used for gene quantification (**Supplementary Fig. 30**). These findings indicated that the fixed pipeline effectively reduced variations introduced by various bioinformatics processes.” (Results, lines 378–381)

Second, after applying the RefSeq-Salmon pipeline, pvca analysis in absolute expression levels revealed sources of variations similar to those observed when using the Ensembl-STAR-StringTie pipeline, such as the mRNA enrichment method, strandedness, and library kits. Variations caused by experimental factors were also greater than biological differences among Quartet samples, yet smaller than biological differences among MAQC samples. The proportions of variations in relative expression levels also significantly decreased, indicating a similar conclusion that relative expression calculation could correct for the influence of experimental factors. We have modified **Supplementary Figures 31–32** (as shown above in our response to your comment 3) and added the necessary description in the Results section as follows:

“A total of 15 factors from the experimental process were considered (**Supplementary Table 4**), which introduced significantly greater variations than biological differences among the Quartet samples (85.1% vs. 5.8%), with mRNA enrichment methods and strandedness as the primary sources (**Fig. 4c**). Additionally, other factors, including library preparation kits, read lengths, the number of exonic reads, RNA inputs, and their interactions also contributed to more than 25% of the variations.” (Results, lines 389–395)

“Employing the alternative analysis pipeline (RefSeq and Salmon) also revealed similar sources of variations, with these introduced variations representing approximately 15-fold and 0.4-fold of biological differences in the Quartet and MAQC samples, respectively (**Supplementary Fig. 31**).” (Results, lines 399–402)

“In relative gene expression levels, the proportion of variation attributed to experimental factors decreased to below 20% for the Quartet and MAQC samples, respectively, which was observed when employing two different fixed analysis pipelines (**Supplementary Fig. 32**). This indicated that relative expression could effectively correct for the influence of experimental factors.” (Results, lines 403–407)

5.10 - Line 383–397: This discussion implies a high SNR as an absolute superiority in performance. However, when comparing methodologies like poly(A) selection and rRNA depletion, the direct comparison of SNR might not be equitable due to inherent differences, such as the inclusion of ncRNAs or the effect of poly(A) tail length on RNA quantification. A more nuanced examination of these factors would be preferred.

Reply: Thanks for your valuable comment and suggestion.

As you mentioned, the two mRNA enrichment methods captured different RNA for sequencing, with the rRNA depletion method including more ncRNAs for analysis. Poly(A) tail length of mRNAs also influences RNA quantification when using the Poly(A) selection method for mRNA enrichment, inevitably introducing bias toward mRNAs with long Poly(A) tails (PMID: 24582499). Here, SNR, as a sample-level quality control metric, similar to other basic quality control metrics (such as GC content), aims to assess expression data quality under different methodological differences in detectable ranges. Therefore, overall, we found that the SNR of the

Poly(A) selection method is higher than that of the rRNA depletion method, indicating that the former can better capture biological differences by detecting genes within its detectable range.

A more nuanced examination of these factors is also necessary, which enables to reveal more details about the impact of methodological differences. Following your suggestion, we calculated the SNR values for different gene types, including protein-coding genes, sncRNA, lncRNA, pseudogenes, and others. We then compared the SNR for each gene type between the Poly(A) selection and rRNA depletion methods. The results revealed that despite the influence of the Poly(A) tail length, the Poly(A) selection method consistently exhibited a higher SNR for protein-coding genes. In contrast, the rRNA depletion method exhibited a higher SNR for sncRNA. We have added **Supplementary Figure 36** (as shown above in our response to your comment 3) to display the results and added the description in the Results section as follows:

“For expression data quality, the Poly(A) selection method exhibited higher SNR values than the rRNA depletion method. Considering the differences in gene types captured by two methods, we further examined SNR values for different gene types and observed that the Poly(A) selection method primarily exhibited higher SNR values for protein-coding genes. Conversely, for other gene types, particularly sncRNA, the rRNA depletion method demonstrated significantly higher SNR, indicating a more accurate capture of biological differences in these RNAs (**Supplementary Fig. 36**).” (Results, lines 455–462)

5.11 - Line 405–443: The comparisons among different alignment and quantification software are as much about the chosen algorithmic options as the software choices. The text should detail the rationale behind the selection of specific options for each software to tailor the benchmarking to the study's objectives, ensuring readers understand the impact of these choices.

Reply: Thank you for your valuable comment and suggestion.

As you mentioned, the choice of algorithmic options is as important as the software selection. We included all RNA-seq analysis software used by the 45 laboratories to represent real-world scenarios. Additionally, we also included some popular alignment-

free quantification tools such as Sailfish and kallisto. For the algorithmic options of each software, we primarily utilized default parameters. Certain special options were also selected for specific analysis tasks or considering their potential to improve alignment or gene quantification as stated in the Reference Manual. For example, we ran STAR with the option `--outSAMUnmapped Within` to retain unmapped reads for assessing alignment performance, and ran Salmon with the options `--gcBias`, `--seqBias`, and `--posBias` to correct biases for more accurate quantification. We have added explanations of the selection of software and specific algorithmic options in the Materials and Methods section as follows:

*“RNA-seq analysis tools. We included all analysis tools used by the 45 laboratories to reflect real-world scenarios. Additionally, we also included three popular tools, including Subread for alignment⁷⁴, and two alignment-free quantification tools (Sailfish and Kallisto)^{75, 76}. The details of RNA-seq analysis tools, versions, and the command line are listed in **Supplementary Table 8**.*

Three alignment tools, STAR (v.2.7.10b)⁶⁶, HISAT2 (v.2.2.1)⁷⁷, and Subread (v.2.0.3)⁷⁴, were included. Eight gene or transcript quantification tools were included, consisting of genome-alignment quantification tools like featureCounts (v.2.0.3)⁷⁸, HTSeq (v.2.0.2)⁷⁹, StringTie (v.2.2.1)⁶⁷, and STAR, one transcriptome-alignment quantification tool, RSEM (v.1.3.1)⁸⁰ with Bowtie (v.1.3.1) for alignment⁸¹, as well as three alignment-free quantification tools, including kallisto (v.0.48.0)⁷⁵, Salmon (v.1.10.1)⁶⁸, and Sailfish (v.0.9.0)⁷⁶. According to the different quantification principles, these quantification tools can be divided into exon-level tools (featureCounts, HTSeq, and STAR) and transcript-level tools (Kallisto, Salmon, Sailfish, RSEM, and StringTie), which quantify gene expression by counting or estimating reads mapped to exons or transcripts, respectively. Five differential expression analysis tools, edgeR (v.3.42.4)⁸², limma (v.3.56.2)⁸³, DESeq2 (v.1.40.2)⁸⁴, DEGseq (v.1.54.0)⁸⁵, and EBSeq (v.1.40.0)⁸⁶, were included and compared.

Each tool was run with default parameters, and certain special options were also selected either for specific analysis tasks or considering their potential to improve alignment or gene quantification, as described in the reference manual. The specific parameter options are as follows:

- STAR was run with the “--twopassMode Basic” option for more accurate alignment, “--quantMode GeneCounts” option to obtain read counts, and “--outSAMunmapped Within” option to retain unmapped reads.
- Subread was run with --multimapping and -B to retain multi-mapped reads.
- featureCounts was run with optional arguments: -B to only count fragments that have both ends successfully aligned and -C to exclude the chimeric fragments.
- StringTie was run with parameter -e specifying to only estimate the expression levels of the reference transcripts.
- Salmon was run in the mapping-based mode, using optional parameters: --gcBias, --seqBias, and --posBias to correct sequence-specific biases, GC biases, and 5’ or 3’ positional bias.
- Kallisto was run using --bias to correct biases.” (Materials and Methods, lines 1381–1413)

5.12 - Line 422: In junction discovery or isoform detection, read length is as significant as sequencing depth. This needs to be discussed.

Reply: Thank you for your suggestion.

As you mentioned, both read length and sequencing depth influence junction discovery, thereby affecting the identification of isoforms and alternative splicing (PMID: 26100517). Following your suggestion, we have selected high-quality RNA-seq data with read length of 150 bp from four laboratories representing different sequencing depth levels, and shortened reads into 100 bp, 75 bp, 50 bp, and 25 bp using Cutadapt (v.4.8). Then, we used Ensembl annotation, STAR for alignment, and RSeQC for junction extraction from BAM files. We observed that longer read lengths could detect more junctions. The known junctions can still be detected when the read length is reduced to 25 bp, while novel junctions are almost undetectable.

Since we have discussed the impacts of read length and sequencing depth on RNA-seq performance in the section titled "Best practices for experimental designs", to keep the consistency of the manuscript, we have moved the content regarding their influence on junction discovery to the same section. The added **Supplementary Figure 41** and corresponding description is as follows:

“In addition to gene quantification aspects, certain experimental factors such as sequencing depth and read length, were considered to influence exon-exon junction detection, which implies an impact on isoform and alternative splicing identification^{40, 41}. We observed that even lower sequencing depth was sufficient to detect known junctions, and increasing the sequencing depth further facilitated the identification of novel junctions (**Supplementary Fig. 40**). The impact of read length was similar, as known junctions can still be detected when the read length was reduced to 25 bp, while novel junctions were almost undetectable (**Supplementary Fig. 41**).” (Results, lines 489–496)

“Cutadapt (v.4.8)⁹⁴ was used to shorten read lengths into 100 bp, 75 bp, 50 bp, and 25 bp.” (Materials and Methods, lines 1461–1462)

Supplementary Figure 41. The number of junctions detected with different read lengths. RNA-seq data with read length of 150 bp from four laboratories were shortened into 100 bp, 75 bp, 50 bp, and 25 bp using Cutadapt (v.4.8)¹⁰. The junctions were extracted from BAM files using RSeQC (v.5.0.1)¹¹, and could be classified into three types: known junctions, partially novel junctions, and completely novel junctions. In both (a) Quartet and (b) MAQC samples, longer read length led to the detection of more junctions. Error bars represent the standard deviation of all Quartet or MAQC samples.

5.13 - Line 482–501: The section implies that more detected DEGs indicate superior performance. However, considering the biological context of the Quartet dataset, the identification of over 5000 DEGs might be questionable. It's crucial to assess the biological relevance of these DEGs, considering factors like effect size and independent validations, to ensure the DEG analysis accurately reflects the biological phenomena under investigation.

Reply: Thank you for pointing this out.

We agree that the detected DEGs should accurately reflect the biological differences between samples. The identification of an excessive number of DEGs may be questionable, especially considering the small biological differences between the Quartet reference materials.

We also agree that the number of DEGs does not necessarily indicate the performance of DEG identification. In lines 482–501 of the original manuscript, we initially aimed to elucidate the differences in the number of detected DEGs across tools.

To avoid misleading, we have deleted the description of the number of DEGs in the Results section, and primarily elucidated the impact of different tools on the accuracy of DEG identification based on three types of reference datasets. The modified contents are as follows:

“After applying a series of threshold values to filter low-expression genes, we compared the optimal performance of five differential analysis tools with different choices of quantification pipelines, which contributed to 140 differential analysis pipelines (**Supplementary Fig. 33**). We assessed the DEG accuracy based on three reference datasets and investigated the influences of each bioinformatics step on the accuracy (**Figs. 8a–e**). edgeR and DESeq2 consistently outperformed other tools, with DEGSeq and limma slightly lower, and EBSeq being the lowest (**Fig. 8d**). As another accuracy measure, the area under the receiver operating characteristic curve (AUC) was compared across all differential expression analysis pipelines, which captured the statistical discrimination capability of the DEGs. Similarly, edgeR outperformed the other tools, and DESeq2 also exhibited relatively high AUC values (**Fig. 8e and Supplementary Fig. 57**).” (Results, lines 583–594)

REVIEWERS' COMMENTS

Reviewer #1 (Remarks to the Author):

Thank you for the revised manuscript. The authors have shown commendable diligence and effort in addressing the initial review comments, including conducting additional experiments and analyses. These efforts have significantly enhanced the manuscript, leading to a more thorough and robust presentation of the findings. The revisions have clearly improved the quality and depth of the work, making the manuscript well-rounded and ready for publication. I am satisfied with the current version and recommend its acceptance without further modifications.

Reviewer #2 (Remarks to the Author):

All my comments were addressed

Reviewer #3 (Remarks to the Author):

I have thoroughly reviewed the revised manuscript and the authors' detailed responses to the reviewers' comments. I am pleased to report that the authors have made significant improvements to the manuscript based on the feedback provided.

The authors have incorporated several additional quality metrics and filtering criteria and have validated their findings with further analyses, thus strengthening the conclusions drawn from their initial submission. They have also enhanced the clarity and visual representation of the text, addressing the concerns raised by reviewers to prevent potential misinterpretations.

While there are still inherent limitations related to the generalizability of the findings due to the intrinsic characteristics of the overall study design, the authors have thoughtfully discussed the applicability of their insights to more complex biological experiments and acknowledged the potential caveats. I see that these limitations cannot be readily overcome in the current framework with the reasonable efforts into a single research publication. Nonetheless, the authors' comprehensive analyses across various facets of RNA-seq-based transcriptome studies provide substantial value to readers and contribute significantly to the field.

In conclusion, I believe the revised manuscript satisfactorily addresses the reviewers' concerns and is now ready for publication. The thoroughness of the revisions and the enhancements significantly elevated the manuscript's quality and potential impact. I recommend its acceptance in its present form and encourage the authors to explore further discussions on best practices in RNA-seq analysis with more diverse datasets and study designs in future publications.

Title "Toward Best Practices in Identifying Subtle Differential Expression with RNA-seq: A Real-World Multi-Center Benchmarking Study Using the Quartet and MAQC Reference Materials"

Manuscript ID: NCOMMS-23-61052A

Point-by-Point Response to Reviewers

Reviewer #1 (Remarks to the Author):

Thank you for the revised manuscript. The authors have shown commendable diligence and effort in addressing the initial review comments, including conducting additional experiments and analyses. These efforts have significantly enhanced the manuscript, leading to a more thorough and robust presentation of the findings. The revisions have clearly improved the quality and depth of the work, making the manuscript well-rounded and ready for publication. I am satisfied with the current version and recommend its acceptance without further modifications.

Reply: We are glad that Reviewer #1 is satisfied with our revisions and also appreciate the reviewer' highly constructive comments and questions that significantly improve our manuscript.

Reviewer #2 (Remarks to the Author):

All my comments were addressed

Reply: We are glad we have satisfactorily answered all of Reviewer #2 comments.

Reviewer #3 (Remarks to the Author):

I have thoroughly reviewed the revised manuscript and the authors' detailed responses to the reviewers' comments. I am pleased to report that the authors have made significant improvements to the manuscript based on the feedback provided.

The authors have incorporated several additional quality metrics and filtering criteria and have validated their findings with further analyses, thus strengthening the conclusions drawn from their initial submission. They have also enhanced the clarity and visual representation of the text, addressing the concerns raised by reviewers to prevent potential misinterpretations.

While there are still inherent limitations related to the generalizability of the findings due to the intrinsic characteristics of the overall study design, the authors have thoughtfully discussed the applicability of their insights to more complex biological experiments and acknowledged the potential caveats. I see that these limitations cannot be readily overcome in the current framework with the reasonable efforts into a single research publication. Nonetheless, the authors' comprehensive analyses across various facets of RNA-seq-based transcriptome studies provide substantial value to readers and contribute significantly to the field.

In conclusion, I believe the revised manuscript satisfactorily addresses the reviewers' concerns and is now ready for publication. The thoroughness of the revisions and the enhancements significantly elevated the manuscript's quality and potential impact. I recommend its acceptance in its present form and encourage the authors to explore further discussions on best practices in RNA-seq analysis with more diverse datasets and study designs in future publications.

Reply: We are glad that Reviewer #3 is satisfied with our revisions. We also appreciate the reviewers' highly constructive comments, which have significantly enhanced our manuscript.

We are also very grateful to the reviewer for pointing out the limitations in our study design framework and appreciate the reviewer's understanding that it is difficult to consider all conditions of biological experiments in a single research publication. In future research, we will improve our study design with more diverse datasets to further advance the standardization of the application of RNA-seq.